# Formation of retinal direction-selective circuitry initiated by starburst amacrine cell homotypic contact

**Thomas A Ray[1,2], Suva Roy[1], Christopher Kozlowski[1,2], Jingjing Wang[1,2], Jon Cafaro[1], Samuel W Hulbert[1], Christopher V Wright[3], Greg D Field[1], Jeremy N Kay[1,2]***

[1]Department of Neurobiology, Duke University School of Medicine, Durham, United States; [2]Department of Ophthalmology, Duke University School of Medicine, Durham, United States; [3]Department of Cell and Developmental Biology, Vanderbilt University School of Medicine, Nashville, United States

**Abstract** A common strategy by which developing neurons locate their synaptic partners is through projections to circuit-specific neuropil sublayers. Once established, sublayers serve as a substrate for selective synapse formation, but how sublayers arise during neurodevelopment remains unknown. Here, we identify the earliest events that initiate formation of the direction-selective circuit in the inner plexiform layer of mouse retina. We demonstrate that radially migrating newborn starburst amacrine cells establish homotypic contacts on arrival at the inner retina. These contacts, mediated by the cell-surface protein MEGF10, trigger neuropil innervation resulting in generation of two sublayers comprising starburst-cell dendrites. This dendritic scaffold then recruits projections from circuit partners. Abolishing MEGF10-mediated contacts profoundly delays and ultimately disrupts sublayer formation, leading to broader direction tuning and weaker direction-selectivity in retinal ganglion cells. Our findings reveal a mechanism by which differentiating neurons transition from migratory to mature morphology, and highlight this mechanism's importance in forming circuit-specific sublayers.

DOI: https://doi.org/10.7554/eLife.34241.001

*For correspondence:
jeremy.kay@duke.edu

**Competing interests:** The authors declare that no competing interests exist.

## Introduction

In the developing nervous system, neurons form selective synapses to generate circuits comprised of cell-type-specific connections. This selectivity is important for circuit function because it ensures connectivity between neurons specialized for particular information-processing tasks. Despite its importance, basic questions about selective synapse formation remain unanswered. For example, we do not know how cell types fated to form synapses coordinate their growth to establish contact with each other. This is a significant cell biological challenge, because the neurons that comprise a single circuit are often born at disparate times and physical locations.

In many tissues, notably the insect and vertebrate visual systems, synaptic specificity is facilitated by laminar specificity, the phenomenon whereby circuit partners project their axons and dendrites to narrow strata within a laminated neuropil (*Sanes and Zipursky, 2010*). The inner plexiform layer (IPL) of the vertebrate retina comprises at least 10 distinct sublayers built from the axons and dendrites of different amacrine, bipolar, and retinal ganglion cell (RGC) types (*Baier, 2013*). By projecting to the same IPL sublayer, circuit partners can be assured of encountering each other. The developmental events that create sublayers and guide circuit partners to converge upon them are therefore essential for establishment of retinal circuitry. At later developmental stages, when rudimentary IPL sublayers have already formed, neurons rely on molecular cues localized to those sublayers for guidance

**eLife digest** Our experience of the world relies on circuits spanning the sense organs and the brain that process information received through our senses. These circuits are made up of many different types of nerve cells that form connections with each other while the brain is developing. For these circuits to be set up properly, nerve cells have to be selective about how they connect with each other. However, researchers know little about how exactly nerve cells form the right connections, or about which genes and proteins are involved.

One of the better understood circuits in the body is known as the 'direction-selective circuit'. Found in the retina at the back of the eye of all backboned animals, this circuit's task is to detect the direction that objects are moving. In the case of mice, scientists have identified all of the cells that make up the circuit, and know how they are all supposed to be connected together. This is a useful starting point for researchers to look in more detail at how nerve cells make the right connections during development to set up a working circuit.

Ray et al. looked at how the direction-selective circuit forms in the retinas of young mice by genetically engineering cells to carry fluorescent proteins, or staining them with chemicals. This allowed the cells to be examined under a microscope at different points in their development. It turns out that one type of cell, known as the 'starburst amacrine cell' because of its firework-like shape, coordinates the formation of the whole direction-selective circuit. First, starburst cells branch out and touch each other. Next, they build a scaffold for the circuit with their branch-like extensions. Finally, other cell types follow this scaffold to form connections and complete the circuit.

Ray et al. identified a protein called MEGF10 on the surface of starburst cells that tells the cells when they have made contact with each other. When starburst cells had MEGF10 taken away, or were prevented from contacting each other, they did not build a scaffold properly, and the circuit was less effective at detecting movement.

It is possible that cells in other brain circuits use a similar method to form connections. Understanding more about how nerve cells form circuits will help researchers to work out what goes wrong in developmental disorders that affect vision, memory and learning. This knowledge would be helpful for designing new treatments for these conditions.

DOI: https://doi.org/10.7554/eLife.34241.002

to the appropriate IPL strata (*Duan et al., 2014*; *Matsuoka et al., 2011*; *Sun et al., 2013*; *Yamagata and Sanes, 2008*; *Visser et al., 2015*). However, a crucial question remains unresolved: How do sublayers form in the first place? Understanding the mechanisms that initiate creation of sublayers will provide significant insight into the earliest step in circuit formation.

To learn how members of a single circuit create layers and converge upon them to achieve synapse specificity, we studied the direction-selective (DS) circuit of mouse retina (*Figure 1A*). This circuit reports the direction of image motion to the brain through the spiking activity of distinct DS ganglion cell (DSGC) types that are tuned to prefer stimuli moving in particular directions (*Demb, 2007*; *Vaney et al., 2012*). The DS circuit comprises a limited number of well-described cell types amenable to genetic marking and manipulation (*Kay et al., 2011*; *Huberman et al., 2009*; *Duan et al., 2014*): (1) DSGCs; (2) GABAergic/cholinergic interneurons called starburst amacrine cells (SACs); and (3) four subtypes of glutamatergic bipolar cells (*Chen et al., 2014*; *Duan et al., 2014*; *Greene et al., 2016*; *Kim et al., 2014*). These DS-circuit cell types project to two IPL sublayers, ON and OFF, named for the light response profiles of the neurons that project to them. ON-OFF DSGCs (ooDSGCs) send dendrites to both sublayers, while SACs and bipolar cells project to one or the other, depending on their subtype (*Figure 1A*). Several molecular perturbations have been described that influence ON vs. OFF laminar targeting in the mouse DS circuit (*Sun et al., 2013*; *Duan et al., 2014*), but in these cases, IPL sublayers still form in the right place; errors are limited to choosing the wrong DS sublayer. Thus, neither the establishment of the DS circuit sublayers nor their positioning in the appropriate IPL region depends on molecules that have been studied to date.

Here, we seek to understand the earliest events leading to formation of the DS circuit IPL sublayers. Two lines of evidence suggest that SACs may take the lead in assembling this circuit. First,

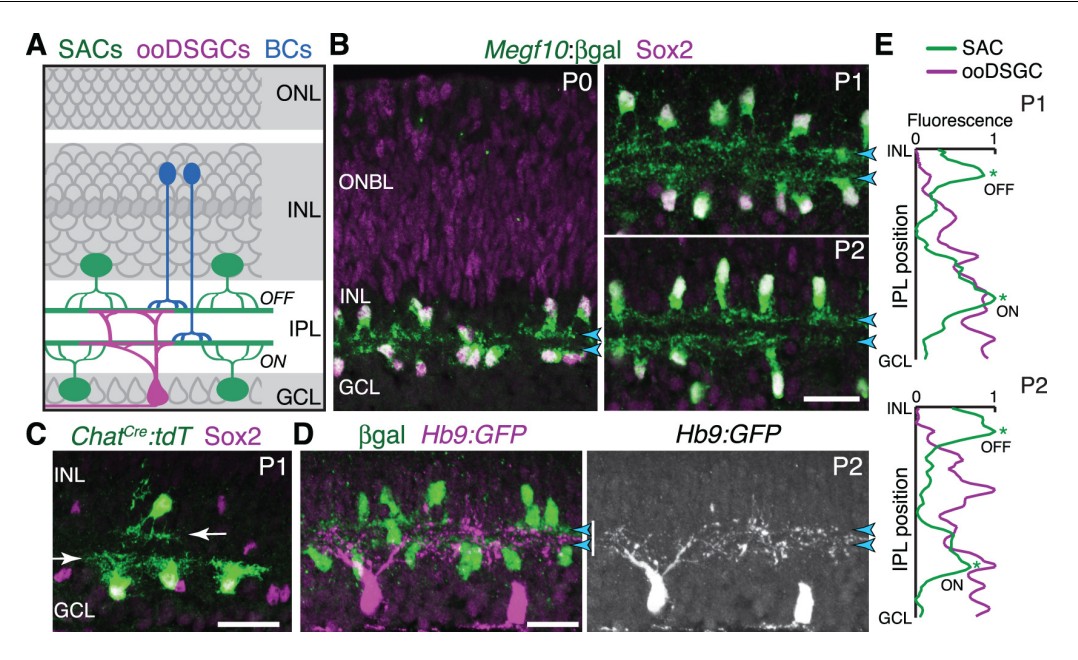

**Figure 1.** Initial formation of DS circuit IPL sublayers. (**A**) Schematic of mature direction-selective (DS) circuit and its cell types, depicted in cross-section. SACs (green) and bipolar cells (blue) project to one of two IPL sublayers (*OFF, ON*). OFF SACs reside in inner nuclear layer (INL); ON SACs reside in ganglion cell layer (GCL). ooDSGCs (purple) send dendrites to both DS circuit sublayers. ONL, outer nuclear layer. (**B**) SAC IPL sublayer formation assessed in *Megf10^{lacZ}* mice. All SACs are double-positive for anti-Sox2 (purple) and anti-βgal (green). Progenitors in outer neuroblast layer (ONBL) also express Sox2. SAC IPL sublayers (arrowheads) begin to appear by P0, and are fully apparent by P1. (**C**) Sparse labeling of neonatal SACs in *Chat^{Cre}* mice. Individual SACs have laminar-specific projections by P1 (arrows). tdT, tdTomato. (**D,E**) ooDSGCs (labeled by Hb9-GFP) project diffusely in the IPL at P1-P2, whereas SAC arbors are stratified (arrowheads). (**D**) retinal cross-sections. Vertical white bar denotes IPL width. E: Fluorescence intensity plots of SAC and ooDSGC dendrite staining across IPL, from representative images (P2 image in D; P1 image in *Figure 1—figure supplement 2*). ON and OFF strata (asterisks) are clear for SACs but not for ooDSGC dendrites. Scale bars: 25 μm.

DOI: https://doi.org/10.7554/eLife.34241.003

The following figure supplements are available for figure 1:

**Figure supplement 1.** Characterization of SAC markers in neonatal retina.

DOI: https://doi.org/10.7554/eLife.34241.004

**Figure supplement 2.** ooDSGC stratification in neonatal retina.

DOI: https://doi.org/10.7554/eLife.34241.005

SACs are among the first cells to stratify the IPL: Even though other neurons innervate it contemporaneously, SACs are precocious in restricting their arbors into sublayers (*Stacy and Wong, 2003*; *Kay and Sanes, 2013*). Second, in mutant mice that entirely lack RGCs or bipolar cells, SAC IPL projections are largely normal, indicating SACs can form sublayers in the absence of their circuit partners (*Moshiri et al., 2008*; *Green et al., 2003*). Thus, we set out to test the hypothesis that SACs orchestrate assembly of the DS circuit sublayers. We find evidence supporting this hypothesis, and we identify a surprising cellular mechanism initiating SAC lamination: Rather than immediately innervating the IPL, newborn SACs first produce a transient homotypic arbor network outside the IPL. These early homotypic contacts serve as a cue promoting SAC dendrite development and circuit integration upon conclusion of their radial migration to the inner retina. When deprived of homotypic contacts, SAC IPL innervation – and consequent sublayer formation – is impaired. We identify the SAC cell-surface protein MEGF10 as the molecular mediator of IPL innervation upon homotypic contact. In the absence of MEGF10, SACs persist in growing arbors outside the IPL, delaying IPL innervation. This in turn delays formation of the DS circuit sublayers and leads to SAC sublaminar

targeting errors that persist to adulthood. We further show that impaired SAC sublayer formation has consequences for laminar targeting of their circuit partners: While partnering remains intact, lamination is disrupted, leading to spatial inhomogeneity in the DS circuit network. Finally, we show that these MEGF10-dependent anatomical changes both broaden and weaken direction tuning across the population of ooDSGCs. These results demonstrate that SACs orchestrate DS circuit assembly, first by initiating sublayer formation via homotypic contact, and then by using their laminated dendrites as a scaffold that guides projections of their circuit partners.

## Results

### Timing of DS circuit IPL sublayer formation

To explore how the DS circuit creates its IPL sublayers, we began by determining when the sublayers first emerge in mouse. This analysis focused on SACs and ooDSGCs because bipolar cells develop later (*Morgan et al., 2006*). Previous estimates of layer emergence vary widely (*Stacy and Wong, 2003*; *Sun et al., 2013*) due to the lack of adequate markers to study dendrite development in neonatal SACs. We therefore assembled a suite of mouse lines and antibody markers for this purpose, enabling anatomical studies of the full SAC population as well as individual cells (*Figure 1B–C*; *Figure 1—figure supplement 1*; *Figure 2—figure supplement 1*). These markers revealed that SAC dendrites form two continuous well-defined laminae by P1 (*Figure 1B,E*). Some dendrites were stratified already at P0, even though the P0 IPL neuropil is less than one-cell diameter wide (*Figure 1B*; *Figure 1—figure supplement 1*). Further supporting this timeline, individual P1 SACs made lamina-specific projections (*Figure 1C*): 96% of OFF SACs in the inner nuclear layer (INL), and 99% of ON SACs in the ganglion cell layer (GCL), stratified within the expected IPL sublayer (n = 49/51 OFF; 78/79 ON; four mice). By contrast, ooDSGCs projected rudimentary and unstratified dendrites at P1 (n = 18 cells, three mice, none were stratified; *Figure 1E*; *Figure 1—figure supplement 2*; also see *Peng et al., 2017*). Even at P2, only 30% of ooDSGCs co-fasciculated with SAC arbors; the rest projected diffusely within the IPL (n = 23 cells, two mice; *Figure 1D,E*; *Figure 1—figure supplement 2*). These results indicate that SACs form IPL sublayers at P0-P1, and are joined later by their synaptic partners.

### Early SAC projections target neighboring SAC somata

To gain insight into how SACs form their sublayers, we next investigated the cell-cell interactions that immediately precede SAC dendrite stratification. Because SACs stratify early – before any other cell type investigated to date (*Figure 1*; *Kay and Sanes, 2013*; *Stacy and Wong, 2003*) – they are unlikely to form strata by following pre-existing laminar cues. Instead, we hypothesized that SACs create their sublayers by engaging in homotypic interactions. To test this idea, we examined embryonic retina to determine if and when SACs establish homotypic contact. SACs exit the cell cycle at the apical retinal surface and migrate radially through the outer neuroblast layer (ONBL). They next arrive at the inner neuroblast layer (INBL), where postmitotic neurons reside (*Hinds and Hinds, 1978*); *Figure 2A,B*). Then they begin to innervate the nascent IPL, which begins to appear in some retinal regions at E16 (*Figure 2A*). To reveal SAC morphology throughout these steps, the early SAC marker *Isl1* (*Galli-Resta et al., 1997*) was used to drive Cre-dependent expression a membrane-targeted GFP (mGFP) reporter (*Isl1^{mG}* mice). We also examined the orientation of SAC dendrite projections using antibodies to internexin, a marker of SAC primary dendrites (*Figure 2—figure supplement 1*). Staining was performed at E16, when SACs at all stages of their early development could be discerned (*Figure 2A–D*).

Since mature SACs contact each other in the IPL, we expected that the onset of SAC homotypic contact would occur around the time of their earliest IPL projections. Surprisingly, however, this analysis revealed that SACs begin to contact each other within the INBL cell body layer upon the conclusion of their radial migration. Migrating SACs rarely interacted, but on arrival at the INBL, SAC arbors were observed touching the soma or primary dendrite of neighboring SACs (*Figure 2A–D*). The majority of INBL SACs engaged in these soma-layer contacts, such that a GFP+ arbor network connected them (*Figure 2G*). Analysis of primary dendrite orientation indicated that soma-layer contacts likely arose due to projections targeted within this layer: Unlike mature SACs, which exclusively project their primary dendrites toward the IPL, many E16 SACs projected tangentially through the

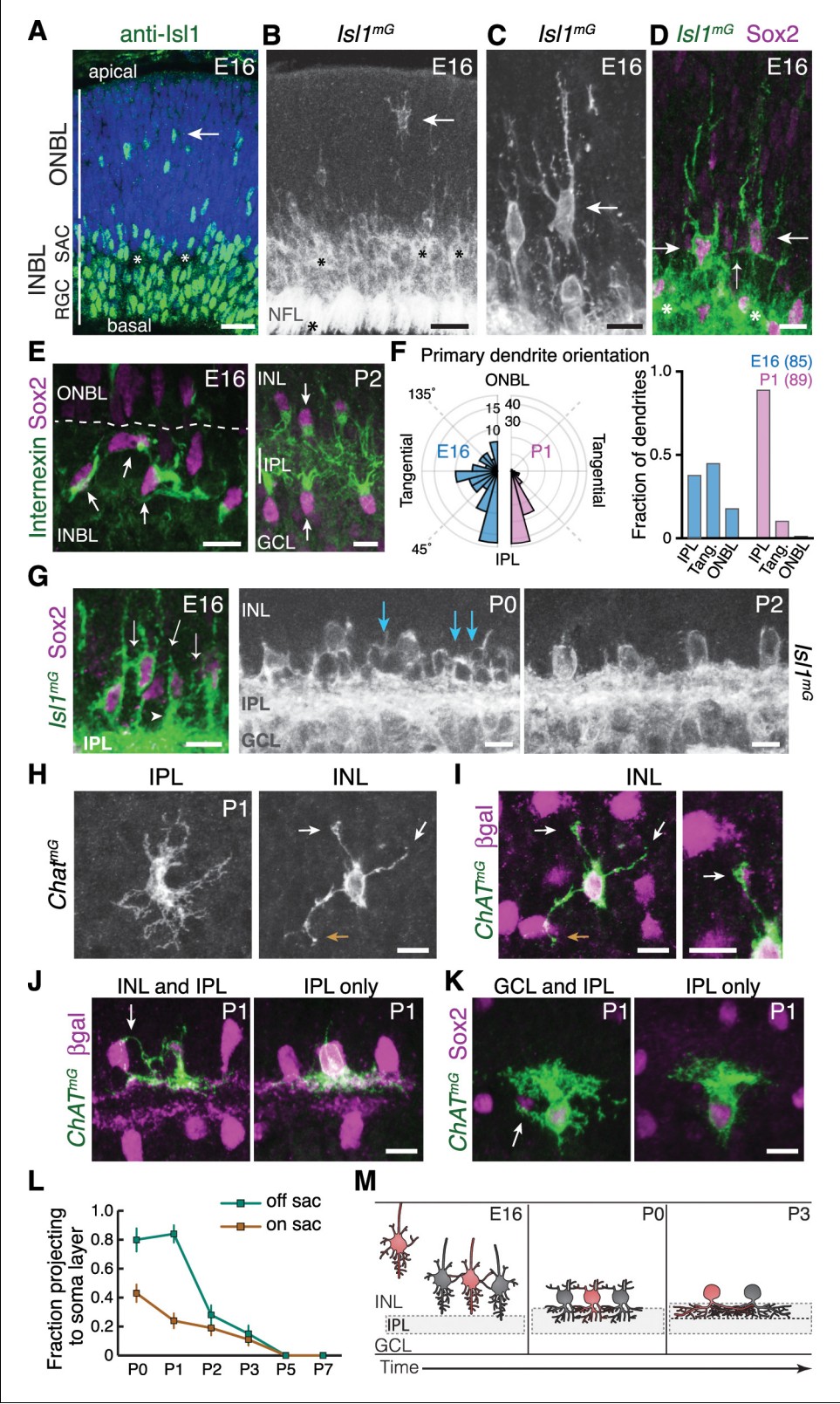

**Figure 2.** Newborn SACs contact each other via a network of soma layer arbors. (**A,B**) Isl1 labels SACs and RGCs in embryonic retina. A, immunostaining; B, mGFP driven by *Isl1^Cre* (*Isl1^mG*). Arrows, newborn SACs migrating apico-basally through ONBL. INBL SACs and RGCs predominantly reside in indicated regions. IPL neuropil (asterisks) exists in discontinuous patches at this age. NFL, nerve fiber layer containing RGC axons. Blue, nuclear

*Figure 2 continued on next page*

*Figure 2 continued*

counterstain. (**B,C**) Migrating SACs in ONBL (arrows) have multipolar morphology. They are far from other SACs and do not contact them. (**D**) Morphology of $Sox2^+Isl1^+$ SACs (large arrows) upon arrival at INBL. SACs contact each other outside the IPL (small arrow, connecting arbor). Their migratory morphology and distance from IPL (asterisks) indicate they have not yet innervated IPL (also see *Figure 2—figure supplement 2*). (**E**) Internexin immunostaining reveals polarization of SAC primary dendrites. SACs project toward IPL at P2. E16 INBL SACs often project tangentially within INBL, towards neighboring SAC somata (arrows). (**F**) Quantification of primary dendrite orientation. Left, polar histogram (raw counts) of primary dendrite angles (absolute values). 0˚ was defined as perpendicular towards IPL. Note that E16 and P1 are displayed on different scales; each plot is scaled to size of largest bin, aiding legibility in E16 plot where bin sizes are more evenly distributed. Right, fraction of dendrites oriented towards IPL, ONBL, or tangential quadrants of the polar plot (denoted by dashed lines, labels). Sample size of scored dendrites is indicated. (**G**) A network of arbors (arrows) connects somata of INBL SACs at E16. The network remains prominent in INL at P0 but is mostly gone by P2. Arrowhead, IPL-directed projection. (**H,I**) An individual P1 OFF SAC labeled by $Chat^{mG}$, imaged *en face* to show its arbor morphology at IPL and INL levels. Full SAC population is revealed using $Megf10$:βgal. INL arbors (**I**) make selective contacts with SAC neighbors (purple): $GFP^+$ arbor tips terminate on SAC somata (orange arrow) or SAC arbors (white arrows). Right panel (**I**) Higher magnification view of touching arbors. Images are Z-projections of confocal stacks encompassing each arbor's volume (H, 2.0 μm; I, 3.5 μm). Projections are shown for illustration but all contacts were verified across stack volume – see *Figure 2—figure supplement 3* for details. (**J,K**) Individual P1 OFF (**K**) and ON (**L**) SACs labeled by $Chat^{mG}$ (green) in cross-section. Purple, full SAC population. Some SACs are bi-laminar with arbors that contact neighboring somata (arrows, left panels); others project only to IPL (right panels). (**L**) Frequency of soma layer projections across development, determined from single $Chat^{mG}$ cells as in J,K. Error bars, standard error. Sample sizes, see Methods. (**M**) Schematic of newborn SAC morphology based on B-L. Soma-layer homotypic contacts are established upon completion of migration and are mostly eliminated by P3. Scale bars: 25 μm (**A,B**); 10 μm (all others).

DOI: https://doi.org/10.7554/eLife.34241.006

The following figure supplements are available for figure 2:

**Figure supplement 1.** Characterization of internexin as a primary dendrite marker of developing SACs.

DOI: https://doi.org/10.7554/eLife.34241.007

**Figure supplement 2.** Soma-layer SAC arbors across development.

DOI: https://doi.org/10.7554/eLife.34241.008

**Figure supplement 3.** Homotypic specificity of soma-layer SAC projections.

DOI: https://doi.org/10.7554/eLife.34241.009

INBL – that is, toward neighboring somata (*Figure 2E,F*). We even noted cases where SACs appeared to project directly towards each other (*Figure 2E*). These observations suggest that post-migratory SACs initiate contact with each other by generating an arbor network in the INBL cell body layer.

Many E16 SACs also innervate the nascent IPL, raising the question of whether the soma- or IPL-layer projection establishes the first homotypic contact. We concluded that soma-layer SAC contact precedes IPL innervation, for three reasons. First, soma contacts were found in retinal regions where the IPL had not yet emerged (*Figure 2—figure supplement 2*). Second, soma contacts were observed among cells that still showed migratory morphological features, such as apical and/or basal processes (*Deans et al., 2011*; *Hinds and Hinds, 1978*), and did not yet project into the IPL (*Figure 2D*; *Figure 2—figure supplement 2*). Third, SAC dendrite polarization in the tangential plane was highly transient: By P1, the vast majority of SAC primary dendrites were oriented toward the IPL (*Figure 2E,F*). These three observations suggest that INBL SACs transiently seek out homotypic soma contact before shifting to target the IPL.

We next sought to determine how long the soma-layer SAC arbor network persists. To this end, we examined SAC anatomy at early postnatal ages using $Isl1^{mG}$ and $Chat^{mG}$ (*Figure 1—figure supplement 1*) mice. At P0-1, although SAC arbors within the soma layers no longer express internexin (*Figure 2—figure supplement 1*), the arbor network remained remarkably prominent (*Figure 2G*). Most OFF SACs assumed a bi-laminar morphology, with one set of arbors in the IPL and another set targeting neighboring SACs in the INL (*Figure 2H–J,L*; *Figure 2—figure supplements 2–3*). INL contacts were highly SAC-selective: 88.8% of branches terminated homotypically (*n* = 122 arbor tips from 22 cells), significantly greater than the contact rate expected by chance (*Figure 2—figure*

supplement 3). By P2-3, however, this dense INL network was mostly gone (*Figure 2G,L*; *Figure 2— figure supplement 2*). ON SACs also made soma layer projections between P0 and P3 that contacted neighboring SAC somata (*Figure 2K,L*; *Figure 2—figure supplement 3*). Together, these observations demonstrate that both ON and OFF SACs make transient soma-layer homotypic contacts that arise prior to IPL dendrite elaboration, and are disassembled at P2-3 after SAC sublayers have formed (*Figure 2M*).

## Homotypic contact is required for SAC IPL innervation and dendrite lamination

SAC homotypic contacts arise at a time when they could serve as a cue for IPL innervation and sublayer formation. To test this idea, we developed a genetic strategy to prevent SACs from contacting each other in vivo. *Ptf1a* encodes a transcription factor required for progenitor cells to assume an amacrine fate (*Fujitani et al., 2006*; *Nakhai et al., 2007*; *Figure 3—figure supplement 1*). We crossed conditional *Ptf1a^flox^* mutant mice (*Krah et al., 2015*) to a Cre line (*Six3-Cre*; *Furuta et al., 2000*) that drives widespread recombination in central retina but spares some progenitors from Cre activity in peripheral retina (*Figure 3A*; *Figure 3—figure supplement 1*). In *Six3-Cre; Ptf1a^flox/flox^* mice (abbreviated Ptf1a-cKO), only these spared Cre⁻ progenitors were capable of giving rise to SACs, indicating that any SACs produced in these mutants are wild-type at the *Ptf1a* locus (*Figure 3C*). Therefore, the Ptf1a-cKO mutant creates a situation where otherwise-normal SACs are present at significantly lower density than in wild-type retina (*Figure 3B,C*). In P1-2 mutants, some SACs were effectively segregated from their neighbors – these were termed 'solitary' SACs – while others had neighbors sufficiently nearby that they touched (*Figure 3B–F*; *Figure 3—figure supplement 2*).

Comparing solitary to touching SACs in Ptf1a-cKO retinas revealed a role for homotypic contacts in promoting IPL innervation and sublayer formation. At P1-2, touching SACs projected normally to the IPL, similar to SACs from *Ptf1a⁺* littermates (*Figure 3D,E,G*). This suggests that any changes in retinal cell type composition caused by loss of *Ptf1a* (*Figure 3—figure supplement 1*) are not by themselves sufficient to perturb SAC sublayer formation. By contrast, solitary SACs largely failed to innervate the IPL (*Figure 3F,G*). This was not caused by abnormal migration: Solitary SACs were properly positioned at the IPL border, but sent only rudimentary arbors into it (*Figure 3F*; *Figure 3— figure supplement 2*). Solitary SACs were also more likely to project processes into the soma layers (*Figure 3G*), and when they did so, the projections were typically more elaborate than those observed in wild-type retina (*Figure 3D,F*; *Figure 3—figure supplement 2*). Thus, solitary SACs overgrew arbors directed toward neighboring somata instead of growing IPL dendrites. Both types of projection errors were also seen at P15, indicating that early errors persist to retinal maturity (*Figure 3—figure supplement 2*). Misprojecting SACs were still closely apposed to numerous other amacrine cells, and their arbors were intermingled in the IPL, strongly suggesting that generic amacrine interactions are not sufficient to ensure normal dendrite targeting (*Figure 3—figure supplement 2*). Instead, homotypic interactions are specifically required for IPL innervation and sublayer formation.

## Requirement for MEGF10 in SAC IPL innervation and sublayer formation

To understand how SACs initiate IPL innervation upon homotypic contact, we next sought to identify the molecular cues that SACs use to recognize that contact has occurred. The cell-surface protein MEGF10 (*Figure 4A*) is a strong candidate to mediate homotypic recognition in this context, for four reasons. First, it is selectively expressed by SACs during the perinatal period (*Figure 1B*; *Figure 1—figure supplement 1*). Second, the onset of its expression coincides with onset of SAC homotypic contact at the conclusion of radial migration (*Figure 4B*). Third, MEGF10 protein is present on soma-layer SAC arbors, making it available to transduce signals arising on these arbors (*Figure 4C*). Finally, MEGF10 mediates SAC-SAC interactions in a separate context – during formation of the orderly 'mosaic' among SAC cell bodies across the retina (*Kay et al., 2012*). Thus, we tested whether MEGF10 also mediates SAC-SAC recognition to initiate IPL innervation. If so, SACs from mice lacking *Megf10* gene function should have phenotypes similar to solitary Ptf1a-cKO SACs – that is, reduced IPL innervation and increased arborization in cell body layers.

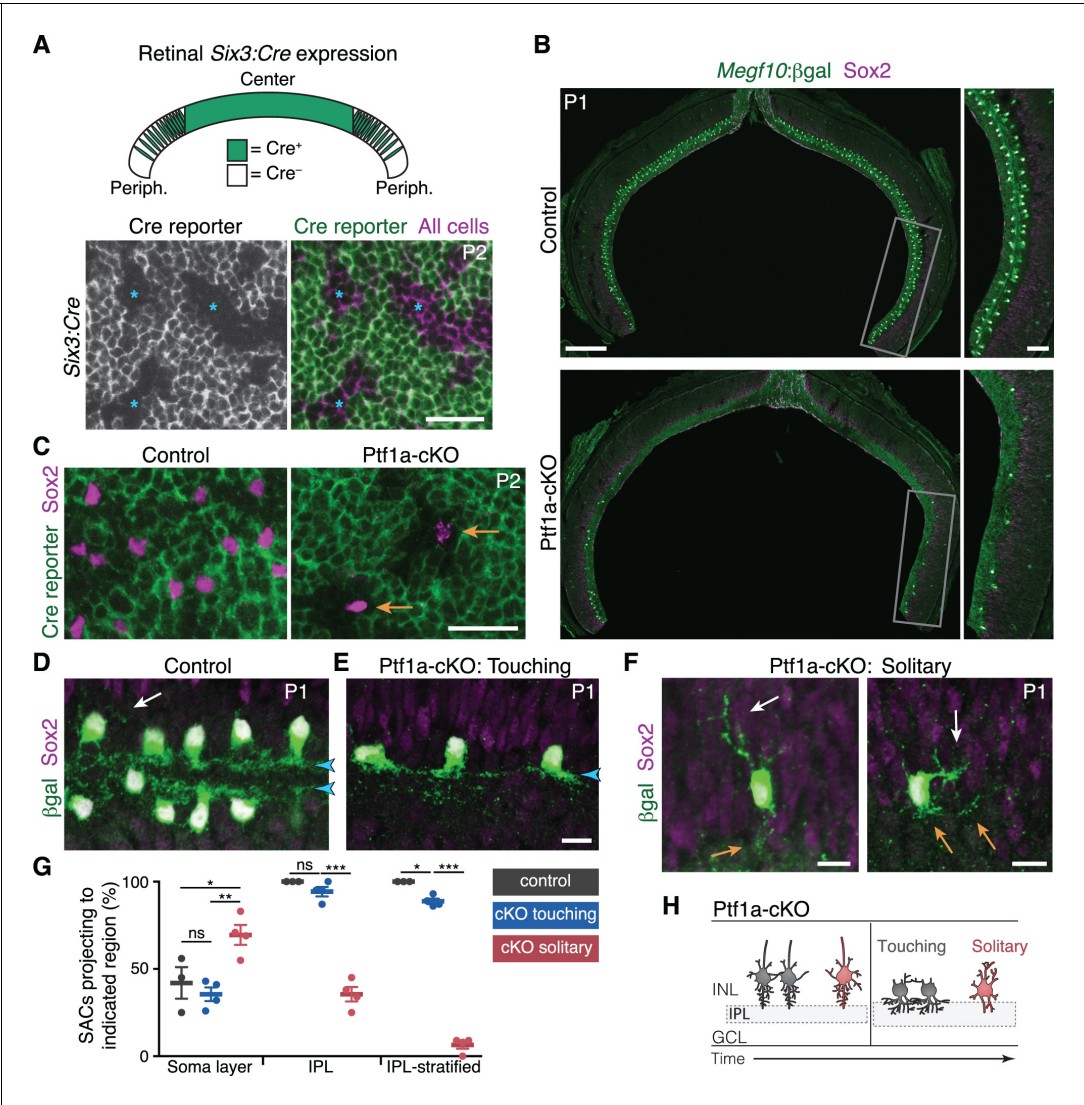

**Figure 3.** SAC homotypic contact is required for IPL sublayer formation. (A) Top: Schematic illustrating *Six3-Cre* expression pattern in retinal cross-section. Bottom: *En-face* view of *Six3-Cre* recombination in peripheral retina, revealed using GFP Cre reporter. Asterisks, Cre⁻ regions. (B) Reduced SAC density in Ptf1a-cKO retina. SACs (labeled by Sox2 and *Megf10^{lacZ}*) are completely eliminated from Ptf1a-cKO central retina; some remain in peripheral retina (boxed regions, right panels). Top, littermate control (*Ptf1a^{+/+}*). (C) *En-face* view of SACs in peripheral retina of Ptf1a-cKO and littermate control. Green, GFP Cre reporter. Control SACs were either Cre⁺ or Cre⁻. Mutant SACs were Cre⁻ (arrows), indicating that they derive only from cell lineages that maintain *Ptf1a* function. (D–F) SAC IPL laminar targeting in Ptf1a-cKO (E,F) and littermate control (D). Ptf1a-cKO SACs close enough to touch (E) form IPL strata (blue arrowheads), similar to control SACs (D). Solitary SACs (F) are not polarized toward IPL; they have extensive INL-directed arbors (white arrows) and rudimentary IPL-directed arbors (orange arrows). Some solitary SACs entirely fail to innervate IPL (F, left cell) and resemble migrating E16 SACs (*Figure 2B,C*); others innervate IPL with minimally branched, non-stratified arbors (F, right cell). (G) Quantification of SAC dendrite phenotypes at P1-2. Left, frequency of soma layer innervation. *p=0.0350; **p=0.0081; ns, p=0.7516. Center, frequency of IPL innervation failure (e.g. F, left). ***p=4.0×10⁻⁷; ns, p=0.3723. Right, frequency of cells that send arbors into IPL but fail to stratify (e.g. F, right). *p=0.0110; ***p<1.0×10⁻⁷. Dots, individual animals. Error bars, S.E.M. p-values, Tukey's post-hoc test. Sample sizes, see Methods. (H) Summary of Ptf1a-cKO SAC phenotype. Touching SACs are similar to wild-type SACs (*Figure 2M*); they are able to innervate the IPL and form sublayers. Solitary SACs remain multipolar, similar to migrating SACs, and fail to innervate the IPL. Soma layer arbors are more elaborate than in wild-type or touching SACs. Scale bars: 25 μm (A,C); 200 μm (B, left), 50 μm (B, right), 10 μm (D–F).

DOI: https://doi.org/10.7554/eLife.34241.010

The following figure supplements are available for figure 3:

**Figure supplement 1.** Retinal cell types in Ptf1a-cKO mutants.

DOI: https://doi.org/10.7554/eLife.34241.011

**Figure supplement 2.** SAC anatomy in Ptf1a-cKO mutants.

DOI: https://doi.org/10.7554/eLife.34241.012

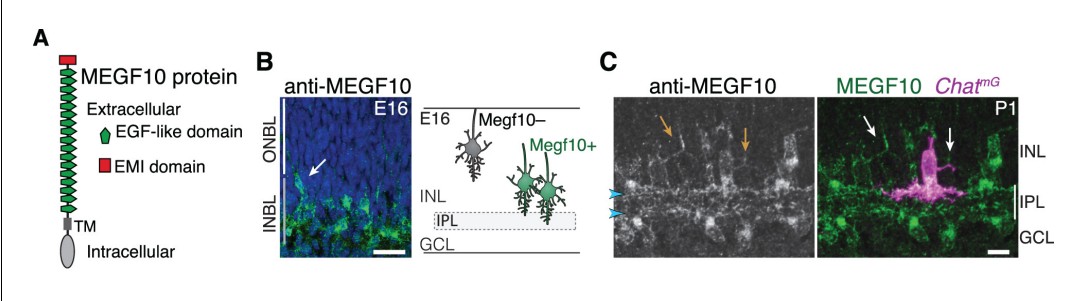

**Figure 4.** MEGF10 is expressed by SACs during early homotypic contact. (**A**) Schematic of MEGF10 protein. TM, transmembrane domain. (**B**) Left, MEGF10 immunostaining at E16 reveals onset of protein expression at conclusion of radial migration. INBL SACs express MEGF10, but SACs migrating through ONBL do not. Arrow, INBL SAC with migratory morphology suggesting it is newly arrived. Right: Schematic illustrating timing of *Megf10* expression onset in SACs (also see *Kay et al., 2012*). (**C**) SAC arbors in the INL (arrows) express MEGF10 protein. IPL dendrites are also labeled (arrowheads). Scale bars: 25 μm (**B**); 10 μm (**C**).
DOI: https://doi.org/10.7554/eLife.34241.013

To test this prediction, we examined SAC anatomy in *Megf10* null mutants (*Kay et al., 2012*) and littermate controls at P0-1, when sublayers are first forming. We found a striking effect on sublayer formation: Both ON and OFF strata were absent or severely disrupted in mutants (*Figure 5A*). The cause of sublayer absence was investigated using pan-SAC labeling (*Figure 5A,B*) and single-cell analysis (*Figure 5C*; *Figure 6D*). These studies revealed a severe deficit in IPL dendrite arborization: Most *Megf10*$^{-/-}$ SACs made only rudimentary, unstratified IPL projections at P0-1 (*n* = 1/15 OFF SACs were stratified). Other amacrine cell types showed normal dendritic morphology in *Megf10* mutants (*Figure 5—figure supplement 1*), indicating that the phenotype was specific to SACs. Loss of IPL innervation was not due to aberrant SAC radial migration, because, at P0, mutant SACs had reached the inner retina in normal numbers (wild-type, 2600 ± 287 SACs/mm$^2$; mutant, 3153 ± 145 SACs/mm$^2$; p=0.144, 2-tailed *t*-test; *n* = 3 each group), and were positioned adjacent to the IPL, similar to littermate controls (*Figure 5A*). Furthermore, most mutant SACs sent at least some arbors into the IPL at P0-1 (*Figure 5A,C*; *Figure 6D*), suggesting that they migrated to a location from which IPL innervation was feasible. However, the mutant SAC arbors that reached the IPL appeared undifferentiated, with a lack of space-filling branches (*Figure 5A,C*). As a result, not only did their arbors enclose a significantly smaller IPL territory, but they also failed to sample as much of their enclosed territory as control SACs (*Figure 5C*; also compare to control cell in *Figure 2H*). By P3 some ON SAC IPL innervation was evident, but OFF SAC arbors remained largely confined to the soma layer; those that did reach the IPL remained undifferentiated (*Figure 5B*; *Figure 6A,D*). These observations indicate that deletion of MEGF10 causes an IPL innervation phenotype strongly reminiscent of Ptf1a-cKO solitary SACs: Both manipulations profoundly impair SAC dendrite arborization within the IPL, preventing timely sublayer formation.

In contrast to their underinnervation of the IPL, *Megf10* mutant SACs arborized exuberantly in the soma layers (*Figure 6A*). Both ON and OFF SACs were affected (*Figure 6D,E*; *Figure 6—figure supplement 1*), but the OFF SAC phenotype was particularly striking: Starting at P1, the mutant INL network became much more elaborate than the control network of any age (*Figure 6A,C*). INL arbor density increased in mutants from P0 to P1 and remained high at P3; by contrast, control SACs largely eliminated their INL projections over the same period (*Figure 5A,B*; *Figure 6A,E*). To understand how mutant SACs generate a denser and more persistent soma-layer network, we assessed single SAC morphology (*Figure 6A,D*). From this analysis, we determined that one reason for the denser mutant network, particularly at P2-3, was that a larger number of mutant cells projected to the soma layers (*Figure 6E*). However, this reason was not sufficient to explain the denser mutant INL network at P1 (*Figure 6C*), because at that age the number of cells projecting to the INL was similar in mutants and littermate controls (*Figure 6E*). Therefore, to account for this increase in INL arbor density, we surmised that individual mutant SAC must, on average, overinnervate INL. Supporting this conclusion, we found that mutant SACs frequently had more extensive INL arbors than littermate control SACs (*Figure 6B*). Further, mutant SACs continued to grow primary dendrites tangentially within soma layers at P1, when the vast majority of control SACs only targeted the IPL

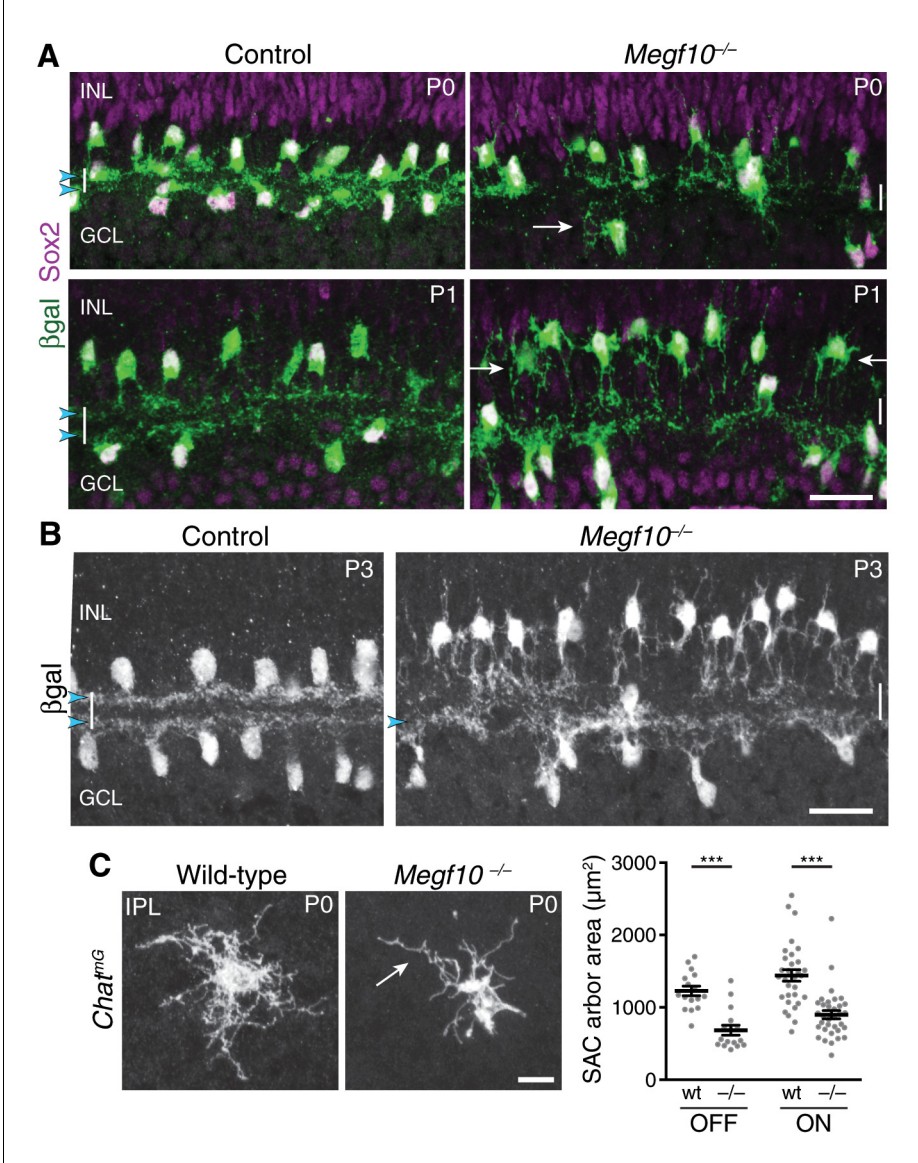

**Figure 5.** *Megf10* is required for initial formation of SAC IPL sublayers. (**A**) SAC sublayers are absent from P0-1 *Megf10* mutant IPL. Antibodies to Sox2 and βgal reveal SACs in retinal cross-sections. Littermate control, *Megf10^lacZ/+*. Vertical white lines denote IPL location. Arrowheads, SAC IPL strata. Arrows, exuberant arbor growth in mutant INL and GCL. Note that mutant somata abut the IPL at P0, indicating their radial migration was similar to controls. By P1 OFF somata have moved apically. (**B**) At P3, SAC IPL sublayers remain disrupted in *Megf10* mutants. Mutant OFF SACs mostly fail to ramify arbors in IPL; instead they arborize in INL (also see *Figure 6*). INL projections are absent from controls. Some ON SACs are stratified in mutants (arrowhead) but have not yet formed a continuous restricted sublayer as is seen in controls. (**C**) Individual SACs show IPL innervation deficits in *Megf10* mutants. *En-face* whole-mount view of single P0 OFF SAC IPL arbors. Mutant SAC IPL dendrites appear undifferentiated, with less branching (arrow). Their arbors cover smaller retinal territories than SACs from wild-type (wt) littermate controls (quantified at right, mean ± s. e. m.). Images are Z-projections of slices encompassing full IPL arbor volume. ***$p$(on)<$1.0 \times 10^{-7}$, $p$(off)=$9.38 \times 10^{-5}$; one-way ANOVA/Tukey's post-hoc test. Sample size, see Methods. Scale bars: 25 µm (**A,B**); 10 µm (**C**).

DOI: https://doi.org/10.7554/eLife.34241.014

The following figure supplement is available for figure 5:

**Figure supplement 1.** *Gad1-GFP⁺* amacrine cells show normal dendrite projections in *Megf10* mutants.

DOI: https://doi.org/10.7554/eLife.34241.015

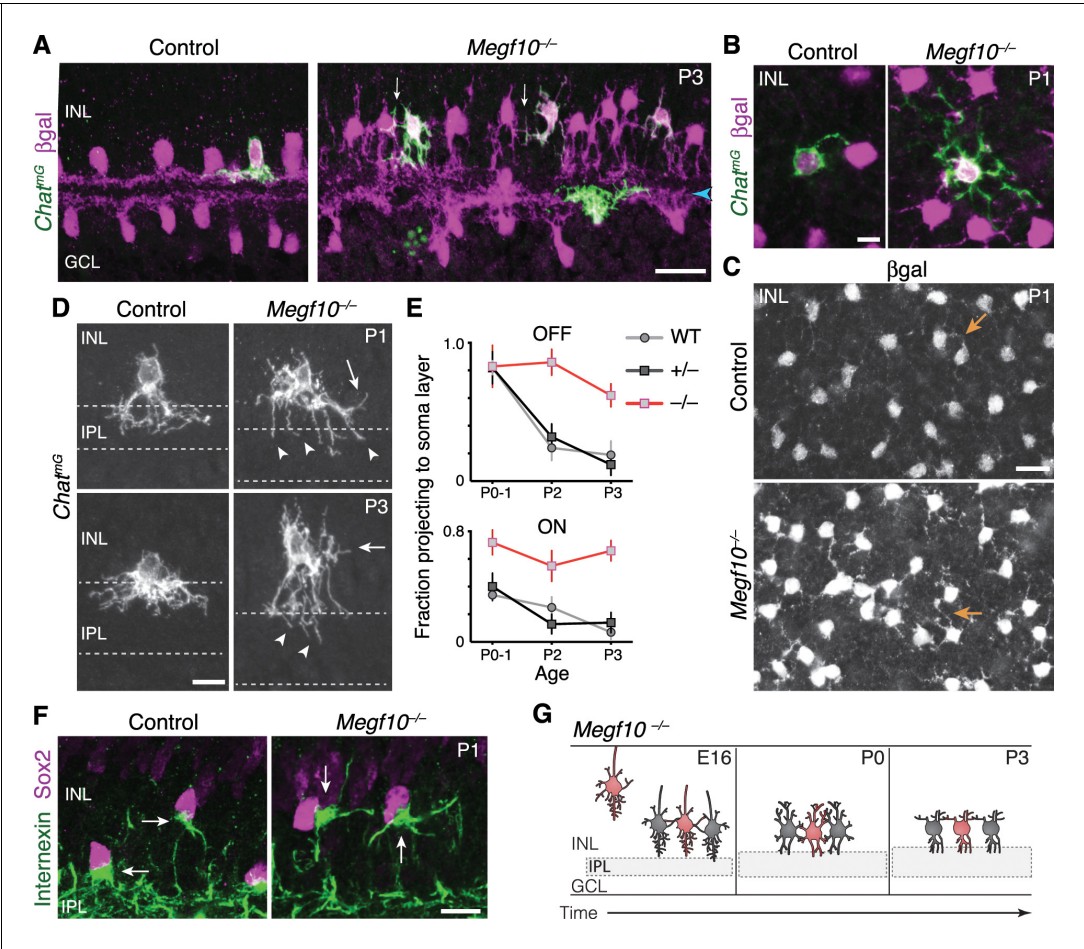

**Figure 6.** Exuberant soma-layer innervation by *Megf10* mutant SACs. (**A,B**) Individual SACs seen in cross-section (**A**) and whole-mount *en face* view (**B**) of *Chat^{mG}* tissue at indicated ages. *Megf10* mutant SACs have more extensive branches in INL than littermate *Megf10^{+/−}* SACs. Images in B are Z-projections of confocal slices encompassing each cell's INL arbors. (**C**) En *face* images of βgal staining to show complete SAC INL network at P1. INL projections (arrows) are present at this age in controls (top); however, they are much more prominent in mutants (bottom), consistent with single-cell anatomy (**B**). Images are Z-projections of confocal slices encompassing volume of 2.0 μm (bottom) or 2.4 μm (top). (**D**) Higher-magnification view of individual OFF SACs labeled as in A. Genotypes and ages are indicated. Arrows, arbors in INL. Dashed lines denote borders of IPL. Mutant IPL projections (arrowheads) fail to arborize or stratify. See *Figure 6—figure supplement 1* for ON SACs. (**E**) Frequency of soma layer projections across development in mutants (–/–) and littermate controls (+/–), determined from single *Chat^{mG}* cells as in D. Wild-type (WT) data replotted from *Figure 2L* to show that +/– controls resemble WT. Error bars, standard error. Sample size, see Materials and methods. (**F**) Internexin immunoreactivity reveals orientation of SAC primary dendrites (arrows) at P1. Right: Example of mutant SACs projecting primary dendrites in tangential plane, within soma layer. Control primary dendrites were almost exclusively oriented towards IPL (left). (**G**) Summary of *Megf10^{−/−}* phenotype. After initial contact at E16, mutant SACs do not immediately innervate the IPL, instead overgrowing arbors in cell body layers. This leads to delayed sublayer formation and persistent soma-layer projections at P3. Scale bars: 25 μm (**A,C**); 10 μm (**B,D,F**).

DOI: https://doi.org/10.7554/eLife.34241.016

The following figure supplement is available for figure 6:

**Figure supplement 1.** ON SACs also make exuberant soma-layer projections in *Megf10* mutants.

DOI: https://doi.org/10.7554/eLife.34241.017

(*Figure 6F*; also see *Figure 2F*). These observations indicate that mutant SACs continue to expand their soma layer arbor network at P1. Thus, as with solitary Ptf1a-cKO SACs, soma layer projections were both more frequent and more exuberant for *Megf10^{−/−}* SACs.

Together, these data suggest that MEGF10 governs a developmental transition from soma-layer to IPL-layer dendrite growth (*Figure 6G*): Whereas control SACs have only a brief period of soma-layer growth, switching to IPL ramification around P0, *Megf10* mutant SACs do not make this transition and instead persist in soma-layer innervation. As a result of this failed transition, many individual

mutant SACs ramify extensively in the INL but underinnervate the IPL, causing the dendrite targeting phenotypes that were observed at the population level (*Figure 6A–C*). We conclude that, because MEGF10 regulates IPL innervation in this way, MEGF10 is required for initial formation of SAC IPL sublayers.

## SAC dendrite targeting requires transcellular MEGF10 signaling

Given the similar phenotypes of *Megf10* mutant and solitary Ptf1a-cKO SACs, we hypothesized that MEGF10 is the molecular cue that triggers IPL innervation upon SAC-SAC contact. A key prediction of this model is that SACs should require MEGF10 signals from their neighbors to target their dendrites properly. To test this prediction, we generated a conditional *Megf10flox* allele and used it to create a situation where *Megf10+* SACs were surrounded by *Megf10−* mutant cells. This was accomplished via the same *Six3-Cre* strategy that we employed in our Ptf1a-cKO studies (*Figure 3A–C*). In central retina of *Six3-Cre; Megf10flox/lacZ* (Six3-Megf10-cKO) animals, the vast majority of cells expressed a Cre-dependent GFP reporter, indicating that they lacked *Megf10* function (*Figure 7A*). Accordingly, SACs projected exuberantly to the INL and sublayer formation was disrupted, as in null mutants (*Figure 7B*; *Figure 7—figure supplement 1*).

In peripheral retina, some SACs escaped Cre activity, leading to absence of the GFP reporter and continued MEGF10 protein expression (*Figure 7A,B*; *Figure 7—figure supplement 1*). Our model predicts that these cells should have mutant dendrite phenotypes despite retaining MEGF10. To test this prediction, we imaged βgal-stained OFF SACs from Six3-Megf10-cKO and littermate control mice at P2. This age was chosen because wild-type and null mutant mice showed a large difference in SAC INL projection frequency (*Figure 6E*). In littermate controls, we found that βgal+ SACs rarely projected to the INL (*Figure 7C,D*); therefore, they behaved like control SACs from earlier experiments (*Figure 6E*). By contrast, *Megf10+* SACs surrounded by mutant SACs in Six3-Megf10-cKO retina showed a high rate of INL projections, nearly identical to their *Megf10−* neighbors (*Figure 7B,D*; *Figure 7—figure supplement 1*). Thus, when *Megf10+* SACs are deprived of MEGF10 signal from adjacent SACs, they make exuberant soma-layer projections. This finding implicates MEGF10 as a transcellular signal that controls SAC dendrite targeting (*Figure 7K*).

Next, we investigated how SACs receive this MEGF10 signal from their neighbors. Given that MEGF10 can function as a receptor in other contexts (*Chung et al., 2013*; *Kay et al., 2012*), we speculated that MEGF10 might act as its own receptor. In support of this idea, co-immunoprecipitation experiments using intracellularly truncated *Megf10* constructs showed that MEGF10 can interact with itself through its extracellular domain (*Figure 7I,J*; *Figure 7—figure supplement 2*). Thus, MEGF10 appears biochemically capable of acting as both ligand and receptor.

If MEGF10 is indeed a receptor in this context, SACs should require it to detect contact with MEGF10-expressing homotypic neighbors. To test this prediction, we asked whether removal of *Megf10* from a single SAC, during the period of soma-layer homotypic contact, would impair its IPL innervation despite normal MEGF10 expression by surrounding cells. We used *ChatCre* to achieve sparse recombination in SACs of neonatal mice, as in the anatomy experiments described above (*Figure 2H–K*; *Figure 6D*). In Chat-Megf10-cKO animals, MEGF10 immunostaining was used to identify SACs that lost MEGF10 protein prior to P3 – that is, during the period when soma-layer arbors are present (*Figure 7F,G*). MEGF10− cells constituted a small minority of SACs at P3, meaning that they were generally surrounded by MEGF10+ neighbors (*Figure 7—figure supplement 1*). In this context, MEGF10− SACs produced more exuberant soma-layer arbors than neighboring MEGF10+ cells, while sending only minimal arbors into the IPL (*Figure 7E–H*). Thus, single MEGF10− SACs had phenotypes similar to SACs from mice entirely lacking *Megf10* (*Figure 7G,H*; compare to *Figure 6D*). By contrast, adjacent MEGF10+ cells in the same Chat-Megf10-cKO retinas were indistinguishable from littermate control SACs (*Figure 7E,F,H*). Therefore, when *Megf10* is lost during dendro-somatic contact (but not after; see below), SACs make projection errors typical of neurons deprived of homotypic interactions, and they do so even if their neighbors express MEGF10 and are developing normally. Together, these experiments support the conclusion that MEGF10 is a receptor through which SACs detect each other to terminate soma-layer growth and initiate IPL innervation (*Figure 7K*).

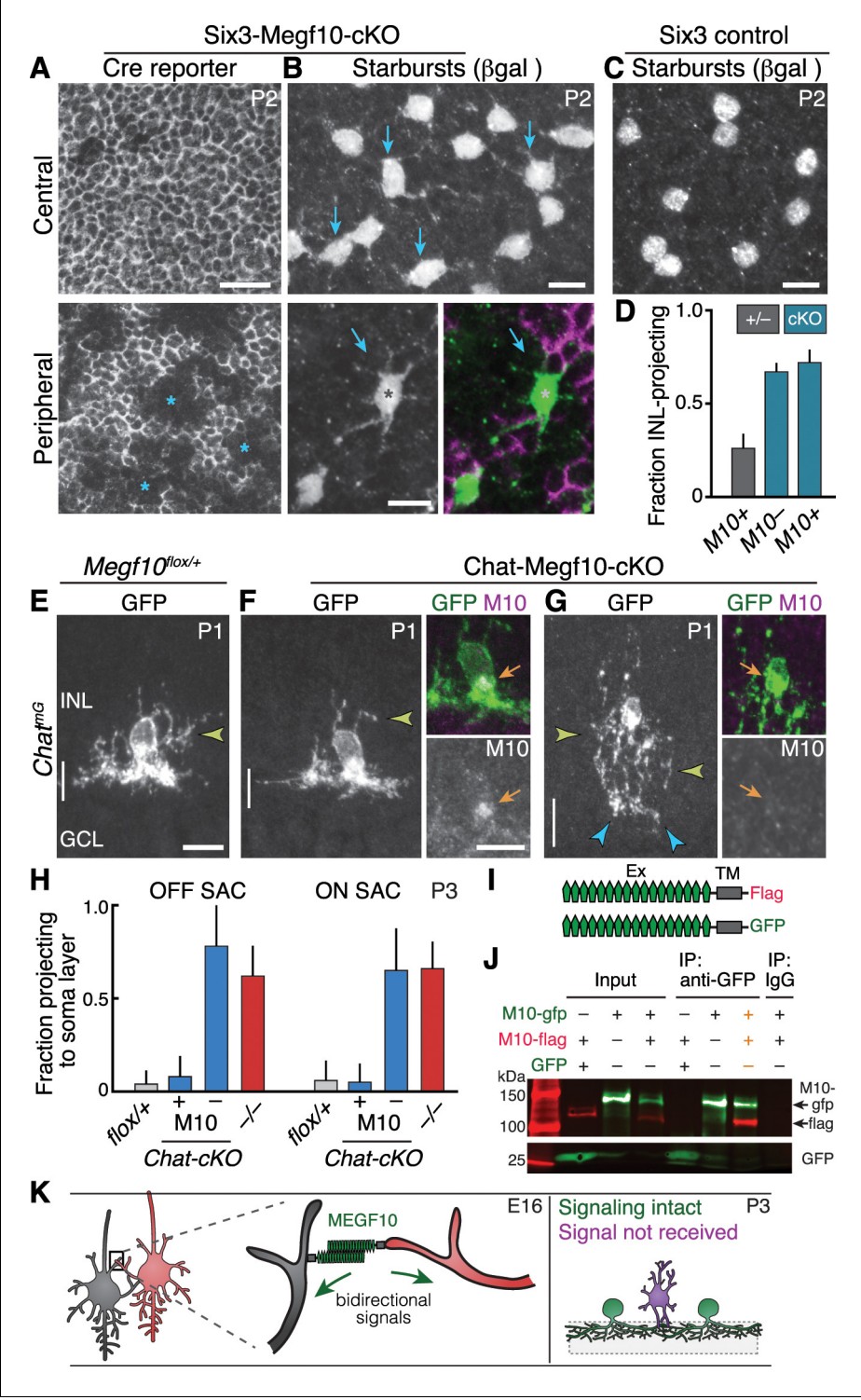

**Figure 7.** *Megf10* mediates transcellular SAC signals for dendrite development. (**A,B**) *En-face* images of INL in Six3-Megf10-cKO retinas stained for GFP Cre reporter (**A**) and βgal SAC marker (**B**). Reporter expression indicates loss of MEGF10 (see *Figure 7—figure supplement 1*). In central retina (top row), most SAC are mutant, and project extensive INL dendrites (B, arrows; compare to C). In peripheral retina (bottom row), some cells escape Cre (asterisks) and retain MEGF10 but still make ectopic INL projections. Purple, Cre reporter; green, βgal. (**C**) Littermate control retina imaged as in B; SACs rarely project INL dendrites at P2. (**D**) Quantification of P2 INL projection phenotypes illustrated in A-C. Six3-Megf10-cKO (*cKO*) SACs that escape Cre (M10+) make projection
*Figure 7 continued on next page*

*Figure 7 continued*

errors at similar rate as surrounding mutant cells from the same tissue (M10⁻). (**E–G**) Chat-Megf10-cKO phenotype. Morphology of single SACs, revealed by *Chat^mG* in cross-sections. Anti-MEGF10 (M10) distinguished two classes of *cKO* SACs (orange arrows): Those that express MEGF10 (**F**) are anatomically similar to littermate control SACs (**E**). Those lacking MEGF10 (**G**) arborize extensively in INL (yellow arrowheads) but minimally in IPL (blue arrowheads). Vertical line, IPL. (**H**) SAC soma-layer projection frequency at P3. Sparse M10 deletion (blue, –) phenocopied germline null (red). Chat-Megf10-cKO cells that retained M10 (blue, +) resembled controls (*flox/+*). (**I**) Schematic of MEGF10 proteins used for co-immunoprecipitation (IP). Intracellular domain was deleted (ΔICD) and replaced with epitope tags (Flag or GFP). Ex, extracellular; TM, transmembrane. (**J**) Co-IP from lysates of HEK 293 T cells transfected with indicated constructs (**I**). Western blot with antibodies to GFP (green) and Flag (red). IP with anti-GFP, but not rabbit IgG control, pulled down both MEGF10-ΔICD constructs (2^nd lane from right, orange text). IP with anti-Flag gave similar result (*Figure 7—figure supplement 2*). GFP alone did not co-IP with M10-Flag. Ladder molecular weights (kDa) at left. Full blots in *Figure 7—figure supplement 2*. (**K**) Model of MEGF10 function in early SAC dendrite development. Left, soma-layer contact between neighboring SACs initiates MEGF10-mediated signaling in each cell. This signal inhibits soma-layer dendrite growth and promotes arborization in IPL (see green cell, right panel). Purple, phenotype of SACs that fail to receive MEGF10 signals, either because neighbors do not have MEGF10, or because the cell itself lacks MEGF10 as a receptor. These SACs project exuberant soma-layer arbors and fail to ramify dendrites in IPL. Error bars, 95% confidence interval. Sample sizes, see Methods. Scale bars: 25 μm (**A**), 10 μm (**B–G**).

DOI: https://doi.org/10.7554/eLife.34241.018

The following figure supplements are available for figure 7:

**Figure supplement 1.** *Megf10* cell autonomy: Characterization of conditional mutant mice.
DOI: https://doi.org/10.7554/eLife.34241.019
**Figure supplement 2.** MEGF10 co-immunoprecipitation experiments.
DOI: https://doi.org/10.7554/eLife.34241.020

## SAC errors persist to adulthood in *Megf10* mutants

We next asked whether neonatal MEGF10-mediated interactions influence the anatomy of SAC IPL sublayers at maturity. We found that SAC sublayers eventually formed (by P5; *Figure 8H*), and were present in the mature *Megf10^-/-* retina, but they were marred by numerous errors. Sporadically, and at apparently arbitrary retinal locations, two kinds of local laminar disruptions were apparent. First, there were discontinuities in the ON and OFF strata, such that mutant SACs did not completely innervate their sublaminae (*Figure 8A–C*). These discontinuities diminished retinal coverage within each mutant sublayer by ~15% (OFF decrease, 15.0 ± 0.9%; ON decrease, 13.7 ± 4.0%; mean ±SD; n = 9 fields of view/2 mice per genotype). Innervation gaps were not observed for other amacrine cells, indicating that SACs were selectively affected (*Figure 8—figure supplement 1*). Examination of single SACs revealed that while dendritic patterning substantially recovered between P1 and adulthood, SAC arbor territories remained significantly smaller in mutants (*Figure 8D*). These phenotypes suggest that mutant SACs never fully made up for their initial IPL innervation deficit, thereby contributing to gaps in the dendritic plexus.

The second type of SAC error in mature *Megf10^-/-* IPL was dendrite mistargeting to ectopic IPL strata (*Figure 8A,B,E*). Both ON and OFF SACs were affected; in each case, ectopic arbors were mostly found in IPL regions inappropriately close to the soma layers (*Figure 8A,B*). *En-face* images of mutant IPL revealed that ectopic OFF arbors formed a patchy but extensive fascicle network connecting many of the cells (*Figure 8E,F*; 78.5 ± 3.5% of SACs participated in the network, mean ±95% CI). This IPL network was morphologically similar to the ectopic INL network observed in mutants at earlier ages (*Figure 6C*), raising the possibility that the early network gives rise to the adult network by shifting location from the INL to the IPL. Supporting this view, we found that a soma layer-to-IPL transition occurs at P5, when mutant SACs began projecting to ectopic IPL locations in addition to the soma layers (*Figure 8G,H*; *Figure 8—figure supplement 1*). This transition occurred without a significant change in the number of mutant SACs projecting into the ectopic network (*Figure 8F*; *Figure 8—figure supplement 1*), suggesting that the same cells continued to participate in the network but simply altered their anatomy to target the IPL. Thus, early exuberant soma-layer projections appear to give rise to adult IPL ectopias, starting between P3 and P5.

Together, these two adult mutant phenotypes demonstrate that DS circuit sublayer formation is delayed and imperfect in the absence of MEGF10. While other mechanisms appear to partially

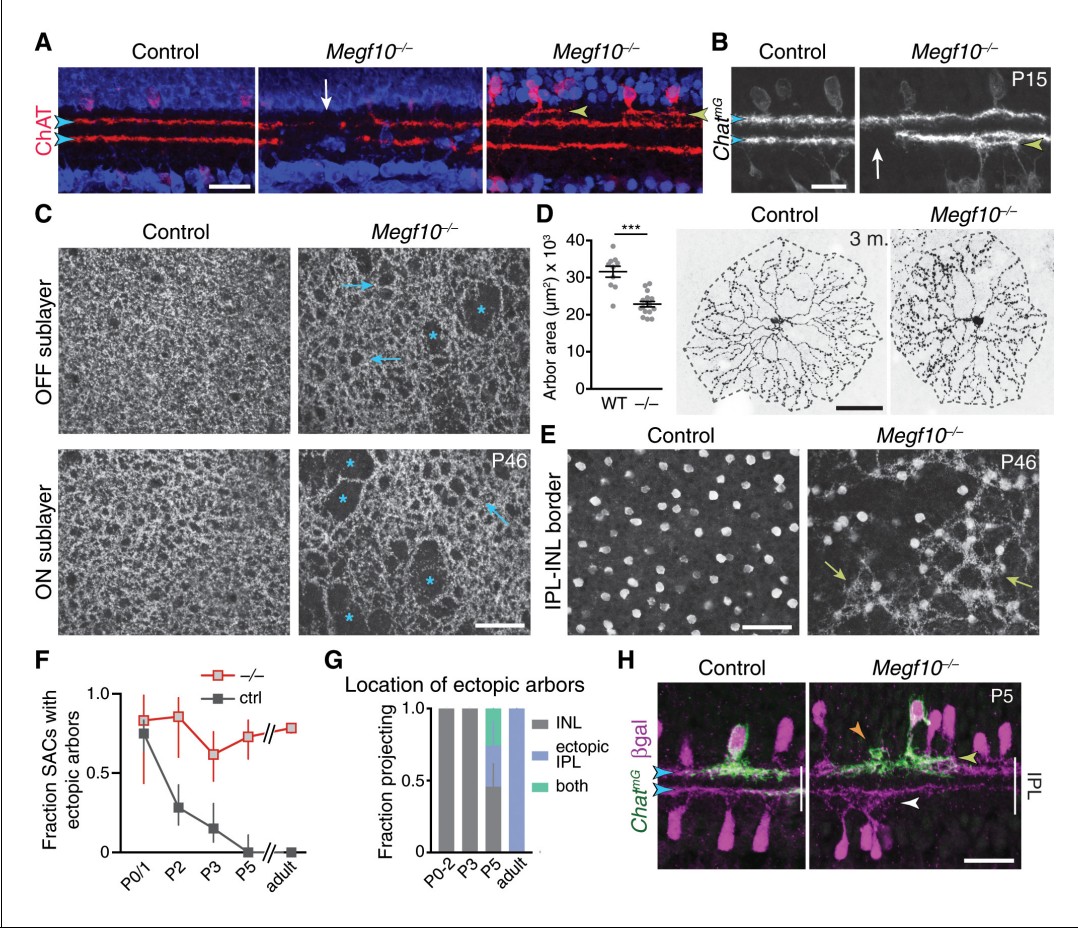

**Figure 8.** SAC IPL errors persist to maturity in *Megf10* mutants. (**A,B**) SAC IPL phenotype in mature (two-week-old) retina, cross-section view. Blue, soma counterstain. Control IPL has two continuous SAC dendrite bands (blue arrowheads). Mutant IPL has sporadic ectopic SAC arbors (yellow arrowheads) or laminar gaps (white arrows). (**C**) *En-face* views of SAC dendrites, stained with anti-ChAT, in adult retinal whole-mounts. The same fields of view are shown at two different Z-stack planes, corresponding to OFF and ON SAC sublayers. SAC dendrite plexus is uniform in littermate controls, but has holes (arrows) and large gaps (asterisks) in mutants. Note that errors are not spatially correlated between OFF and ON sublayers. Images are Z-projections of 5 slices encompassing 2.0 μm in Z. (**D**) Single SAC labeling in adult (3 month old) mice, via *Chat^Cre*-dependent viral fluorescent protein expression. *Megf10^−/−* SACs have relatively normal morphology but are significantly smaller than wild-type (WT) control cells (***p=4.6×10^{−6}, two-tailed *t*-test). Sample size, see Materials and methods. (**E**) *En-face* images reveal extent of ectopic mutant SAC dendrite network. Same fields of view as C, but at different Z-planes depicting OFF IPL (at IPL-INL border). In mutant but not control, SAC dendrite fascicles (arrows) are evident at this IPL level. Images are Z-projections of 3 (left) or 2 (right) slices spaced 0.4 μm in Z. (**F,G**) Dendrite targeting of individual OFF SACs assessed across development. Fraction of mutant SACs projecting into ectopic network does not change over time (**F**). However, location of ectopic arbors shifts from INL to IPL (**G**), starting at P5, when cells projected to either or both locations (**G,H**). In F, P0-3 data are replotted from *Figure 6E*, with both control groups combined. Sample sizes, see Materials and methods. (**H**) Example P5 SACs from *Megf10* mutant and littermate control. A mutant SAC projects to three different locations: (1) correct IPL sublayer (blue arrowhead); (2) ectopic IPL sublayer (yellow arrowhead); (3) ectopic INL arbor aggregate (orange arrowhead). ON SACs also make ectopic IPL projections (white arrowhead). Control cells are monostratified in IPL (left). Note that SAC sublayers have formed by P5 in mutants. Error bars, 95% confidence intervals. Scale bars: 50 μm (**C–E**); 25 μm (**A,B,H**).

DOI: https://doi.org/10.7554/eLife.34241.021

The following figure supplement is available for figure 8:

**Figure supplement 1.** SAC phenotypes in *Megf10* mutants at P5 and at maturity.
DOI: https://doi.org/10.7554/eLife.34241.022

compensate for MEGF10 in generating the sublayers, such mechanisms are not sufficient to prevent persistence of innervation gaps and laminar targeting errors. Thus, MEGF10 is essential for normal formation of the mature SAC IPL projection.

Next, we sought to directly test the idea that MEGF10 is required early – at the time of initial SAC homotypic contact – to ensure normal SAC IPL lamination at maturity. To this end, we used *Megf10^flox^* mice to delete MEGF10 at different times. Deletion prior to the onset of homotypic contact, using the *Six3-Cre* line, fully phenocopied *Megf10^−/−^* adult IPL errors (*Figure 9A*), suggesting a requirement for MEGF10 at the time of contact. To remove MEGF10 from SACs that had already established homotypic contact, we used *Chat^Cre^*. In this line, the number of SACs expressing Cre gradually increases over the first postnatal days to encompass the full SAC population (*Xu et al., 2016*). Therefore, Chat-Megf10-cKO mice can be used both for early, sparse MEGF10 deletion (*Figure 7F–H*) and for later, broad MEGF10 deletion. MEGF10 immunostaining revealed that this late, broad deletion occurs between P3 and P5 (*Figure 7—figure supplement 1*), such that MEGF10 expression is largely preserved during the period when homotypic soma-layer contacts exist (*Figure 2L*), but is eliminated shortly thereafter. In this *Chat^Cre^*-mediated deletion regime, SAC

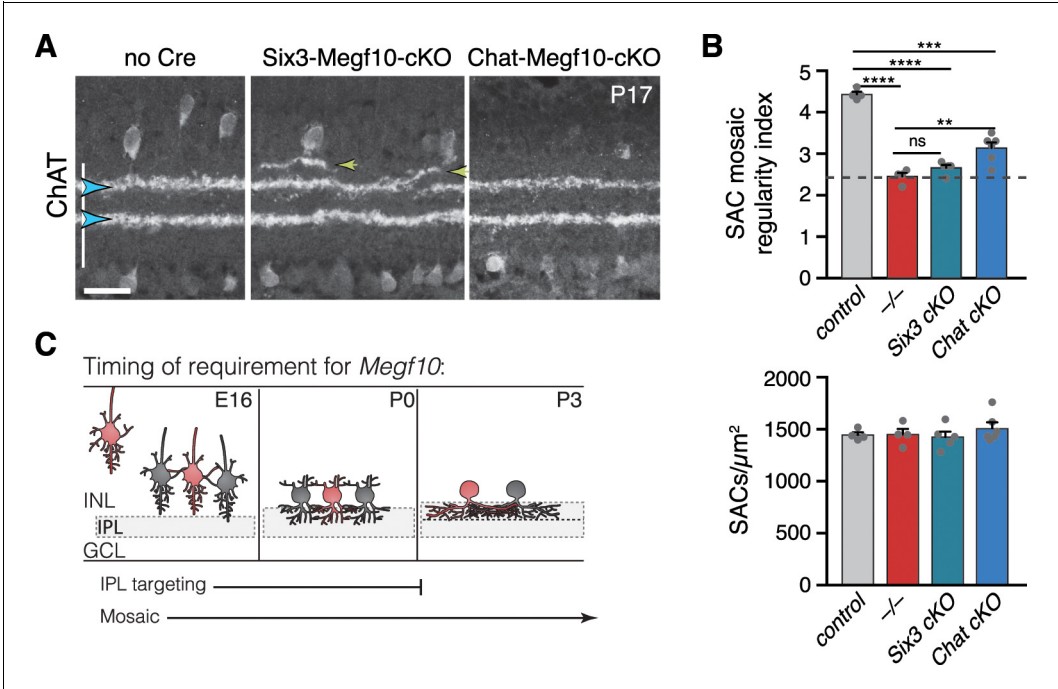

**Figure 9.** Temporal requirements for MEGF10 in SAC IPL stratification and mosaic spacing. (**A**) SAC IPL errors (yellow arrowheads) induced by early deletion of *Megf10* in Six3-Megf10-cKO mice, but not deletion between P3-5 in Chat-Megf10-cKO mice (see *Figure 7—figure supplement 1* for MEGF10 expression in these strains). Blue arrowheads, SAC sublayers. (**B**) Mosaic spacing phenotype measured at P17 using Voronoi domain regularity index (top graph). Dashed line, index for simulated random SAC arrays. In both *Six3* and *Chat* conditional mutants, SAC positioning is less regular than in controls (*Chat^Cre^; Megf10^flox/+^*). *Megf10^−/−^* and simulation data from *Kay et al. (2012)*. ns, p=0.6438; **p=0.0023; ***p=2.1×10^−6^; ****p<1.0×10^−6^ (one-way ANOVA/Tukey's post-hoc test). Bottom graph, regularity effects are not due to changes in SAC cell density across genotypes. One-way ANOVA, F (3, 15)=0.6063; p=0.6210. (**C**) Summary of results from all conditional *Megf10* mutant studies (A, B; *Figure 7*). Loss of MEGF10 while soma-layer arbors are present (i.e. prior to P3) disrupts both SAC mosaic patterning and IPL laminar targeting; this conclusion is based on phenotypes of: (1) germline mutant; (2) Six3-Megf10-cKO mutant; (3) occasional cells in Chat-Megf10-cKO mice that lost MEGF10 prior to P3 (*Figure 7F–H*). When MEGF10 is lost after P3, as is the case for most Chat-Megf10-cKO SACs (*Figure 7—figure supplement 1*), only mosaic is disrupted (**A**, **B**). Thus, MEGF10 acts at distinct, albeit partially overlapping times, to influence these two distinct SAC developmental events. Error bars, S.E.M. Scale bar, 25 μm.

DOI: https://doi.org/10.7554/eLife.34241.023

The following figure supplement is available for figure 9:

**Figure supplement 1.** Severity of *Megf10^−/−^* SAC mosaic phenotype does not correlate with IPL targeting error rate.

DOI: https://doi.org/10.7554/eLife.34241.024

laminar targeting and gap errors were exceedingly rare (*Figure 9A*). These experiments therefore define a time window for MEGF10 function (*Figure 9C*): Adult IPL targeting phenotypes require absence of MEGF10 during the soma-layer projection phase of SAC development – that is, prior to P3. Any additional activity of MEGF10 after P3 is dispensable for the adult IPL phenotype. These findings strongly support a model whereby the functions of MEGF10 during early homotypic contact – that is, promoting IPL innervation and terminating soma-layer arbor growth – are necessary for development of normal SAC IPL innervation at maturity.

## Mosaic spacing errors do not account for SAC IPL phenotype in *Megf10* mutants

In addition to laminar targeting errors, *Megf10* mutants also show disruptions in the mosaic spacing of SAC cell bodies across the retina: Instead of a regular, uniform distribution, mutant SAC positioning is random (*Kay et al., 2012*). We considered the possibility that SAC IPL errors might arise due to MEGF10 effects on soma spacing. Two lines of evidence suggest that this is not the case. First, the two phenotypes were not well correlated at the individual SAC level: Regardless of the severity of their mosaic spacing defects, SACs made IPL targeting errors at a constant rate (*Figure 9—figure supplement 1*). This finding suggests that disturbed cell positioning does not influence the probability of making an IPL error. Second, using our *Megf10^flox^* allele, we were able to dissociate the IPL and mosaic phenotypes: Deletion of MEGF10 after P3 in Chat-Megf10-cKO mice caused mosaic patterning deficits, but IPL projections were largely normal (*Figure 9A,B*). This finding demonstrates that IPL laminar perturbations are not an inevitable consequence of altered soma positioning. Altogether, these experiments support the notion that altered SAC position makes at best a minor contribution to IPL phenotypes; instead, delayed IPL innervation and exuberant soma-layer arborization are likely the major sources of perturbed SAC projections at maturity.

## SAC IPL errors induce laminar targeting errors by their DS circuit partners

Next, we asked whether MEGF10, and its effects on SAC sublayer formation, are important for assembly of the broader DS circuit. To this end, we tested the impact of SAC IPL stratification errors on laminar targeting by their circuit partners. First, we examined ooDSGC IPL projections using the *Hblx9-GFP* (referred to as Hb9-GFP; *Figure 10*) and *Drd4-GFP* (*Figure 10—figure supplement 1*) transgenic lines, which label ooDSGC subtypes with different preferred directions (*Trenholm et al., 2011*; *Huberman et al., 2009*). In littermate control mice (*n* = 9), ooDSGC dendrites were tightly and selectively associated with SAC arbors, as shown previously (*Vaney and Pow, 2000*). This association was maintained in *Megf10* mutants: Both normal and ectopic SAC IPL arbors reliably recruited ectopic ooDSGC projections (*Figure 10A–C*; *Figure 10—figure supplement 1*; *n* = 240 ectopias from five mutants,>97% contained ooDSGC arbors). Further, when SAC gaps were present in the mutant IPL, ooDSGC dendrites typically grew around the gap edges and failed to enter them (*Figure 10D*; *Figure 10—figure supplement 1*; *n* = 325 gaps from five mutants, >95% devoid of ooDSGC arbors). Thus, SACs provide both permissive cues required for ooDSGC IPL innervation, and also attractive cues sufficient to recruit ooDSGCs to the wrong IPL sublayer.

Next we determined the impact of altered SAC lamination on the axons of bipolar cells that participate in the DS circuit (*Figure 11A*). We examined the four cell types (BC2, BC3a, BC5, and BC7) that make extensive monosynaptic connections with SACs and ooDSGCs (*Duan et al., 2014*; *Ding et al., 2016*; Greene*Greene et al., 2016*; Kim*Kim et al., 2014*; *Chen et al., 2014*). Bipolar axons were marked with type-specific antibodies and mouse lines reported previously (*Wässle et al., 2009*; *Duan et al., 2014*), as well as a novel transgenic marker of BC5 (*Gjd2-GFP*; *Figure 11—figure supplement 1*). In wild-type retina, DS-circuit bipolar cells arborized in close contact with SAC dendrites; however, unlike ooDSGCs, they remained adjacent to SACs rather than overlapping them (*Figure 11A–D*; *Figure 11—figure supplement 1*). This arrangement was preserved in *Megf10* mutants: Axons of all four bipolar cell types were recruited to ectopic IPL locations by mistargeted SAC arbors, where they stratified adjacent to SACs (*Figure 11B–D,F*; *Figure 11—figure supplement 1*). For example, BC5 and BC7 terminals always sandwiched SAC arbors, regardless of their IPL location – even when doing so required formation of a supernumerary BC axon field between the normal and ectopic SAC sublayers (*Figure 11C,D*). To quantify the mistargeting effect,

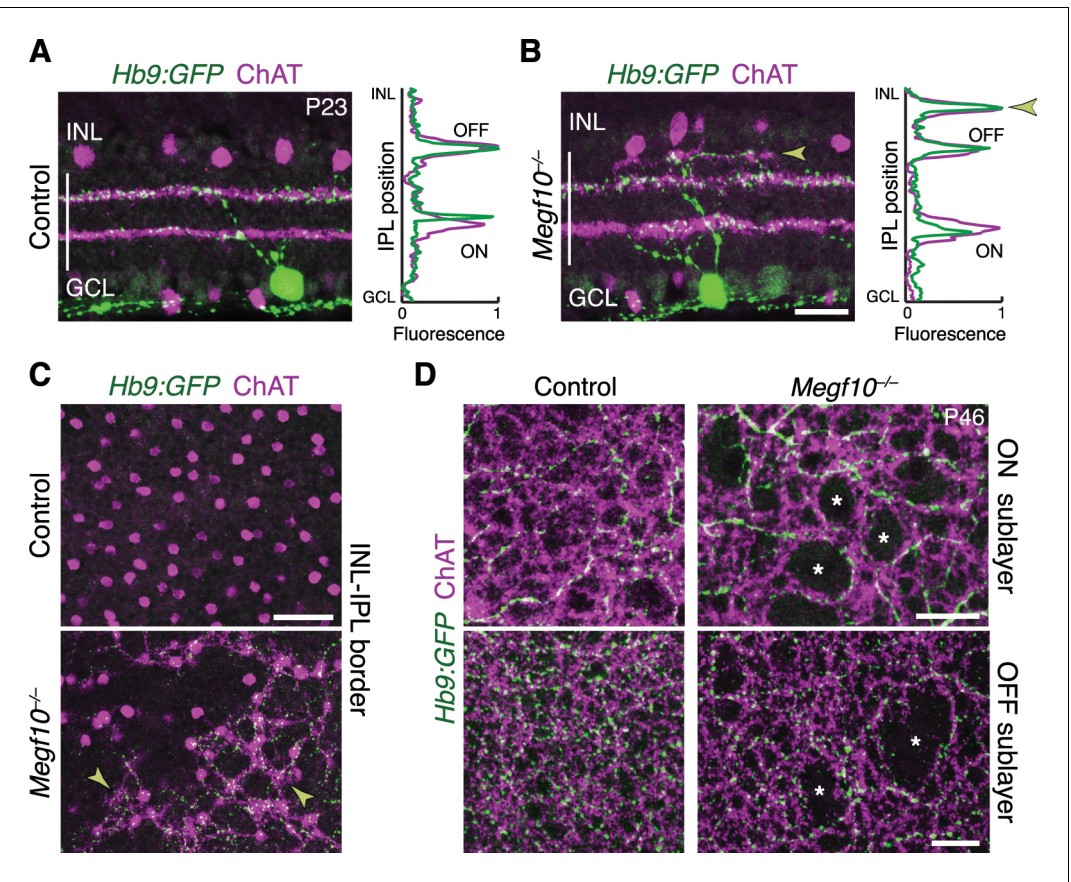

**Figure 10.** SACs guide IPL sublayer choice by ooDSGCs. (A,B) SACs (ChAT, purple) and ooDSGCs (Hb9-GFP, green) labeled in cross-sections. Right panels, fluorescence profile plots across IPL showing position of ON and OFF sublayers. In both controls (A) and *Megf10* mutants (B) ooDSGC dendrites strictly co-localize with SAC arbors. Arrowhead (B), ectopic sublayer. (C) *En-face* view of OFF IPL near INL border. Same fields of view as *Figure 8E*. In *Megf10* mutants (bottom), the ectopic SAC network is extensively innervated by ooDSGC dendrites (arrowheads). Control retina (top) lacks DS circuit arborization at this IPL level. (D) *En-face* view of ON (top) and OFF (bottom) SAC IPL sublayers. In *Megf10* mutants, ooDSGC dendrites (green) fail to enter IPL regions (asterisks) that are not innervated by SACs (purple). Images in C,D are Z projections of confocal slices encompassing ≥1.2 μm (C) or 2–4 μm (D). All scale bars: 25 μm. Also see *Figure 10—figure supplement 1* for phenotype of *Drd4-GFP* ooDSGCs.
DOI: https://doi.org/10.7554/eLife.34241.025
The following figure supplement is available for figure 10:

**Figure supplement 1.** IPL innervation by ooDSGCs in *Megf10* mutants.
DOI: https://doi.org/10.7554/eLife.34241.026

we measured the position of BC5 and BC7 terminals adjacent to ON SAC ectopias. Their arbors were pushed farther apart by SAC arbor clumps (*Figure 11C–E*), which shifted BC7 terminals significantly toward the GCL by ~4 μm (69 ± 0.8% of IPL depth in control regions to 74 ± 1.9% in affected regions; mean ± S.E.M.; n = 21 control, n = 6 affected; 2-tailed *t*-test, p=0.0024). These observations indicate that DS-circuit bipolar cells, like ooDSGCs, respond to SAC attractive cues. However, in contrast to ooDSGCs, bipolar cell projections were minimally affected by SAC IPL gaps. While BC5 and BC7 terminals were slightly mispositioned in the absence of SAC arbors – they were closer together – innervation of gap regions was otherwise normal (*Figure 11C–F*). Thus, DS-circuit bipolar axons either do not require SAC-derived signals for IPL innervation, or the relevant signals are capable of acting over larger distances than the typical SAC IPL gap size (35–45 μm maximum diameter). Altogether, these analyses of DS circuit anatomy in *Megf10* mutants support the notion that early-stratifying SACs form a scaffold that directs IPL laminar targeting of their circuit partners using multiple guidance strategies.

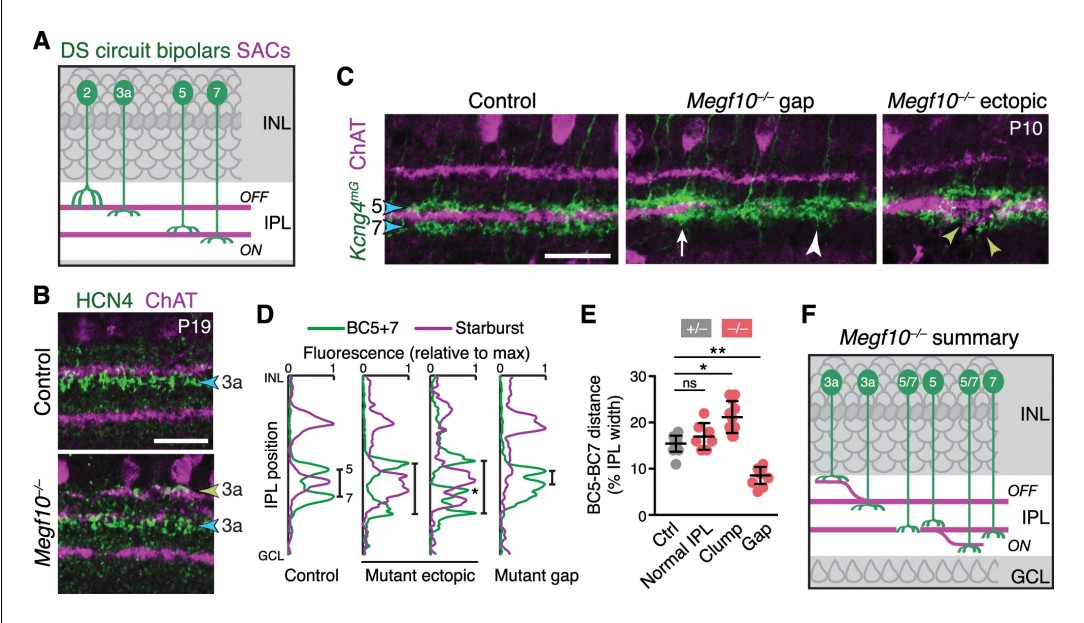

**Figure 11.** SACs guide IPL sublaminar targeting by DS circuit bipolar cells. (**A**) Illustration depicting anatomy of four bipolar cell types known to participate in DS circuit. BC3a, BC5, and BC7 arborize axons in narrow sublayers directly adjacent to SAC strata. BC2 projects more broadly through OFF IPL, overlapping with SAC sublayer (see **Figure 11—figure supplement 1**). (**B**) BC3a IPL projections assessed with anti-HCN4. Blue arrowhead, normal BC3a stratum. Yellow arrowhead, ectopic stratum containing SAC dendrites (purple) and BC3a axons (green). (**C,D**) BC5 and BC7 IPL projections (blue arrowheads), labeled in $Kcng4^{mG}$ mice. C, images; D, representative fluorescence plots of $Kcng4^{mG}$ (green) and ChAT (purple) across IPL. In littermate controls, or normal regions of mutant IPL (C, arrow), BC5 and BC7 arborize in sublayers immediately adjacent to ON SAC layer, but do not enter it. In $Megf10$ mutants, ectopic SAC arbors displace BC5+7 terminals to new IPL locations, where they remain adjacent to SACs but non-overlapping (C, yellow arrowheads; D, center plots). Asterisk (**D**) ectopic BC arbors between normal and ectopic SAC strata. BC5/7 arbors can innervate gaps in the SAC stratum (C, center panel); in these cases, their terminals are abnormally close together (C, white arrowhead; D, right plot). Vertical bars in D: distance between BC5/7 strata. (**E**) Quantification of BC5-BC7 distance in normal IPL and in presence of SAC innervation gaps or ectopic arbor clumps. *p=0.0219; **p=0.0012; ns, p=0.3965 (Tukey's post-hoc test). Sample sizes, see Methods. Error bars, S.E.M. (**F**) Summary of $Megf10^{-/-}$ bipolar cell phenotypes. BC3a, BC5, and BC7 are illustrated here (see **Figure 11—figure supplement 1** for BC2). Each of these cell types can either make errors (left cell in each pair) or project normally (right cell). All three cell types show recruitment to ectopic IPL locations. BC5 and BC7 terminals innervating SAC gaps colonize the sublayer normally occupied by SACs. Error bars, 25 μm.

DOI: https://doi.org/10.7554/eLife.34241.027

The following figure supplement is available for figure 11:

**Figure supplement 1.** IPL innervation by DS circuit bipolar cells in $Megf10$ mutants.

DOI: https://doi.org/10.7554/eLife.34241.028

## Early SAC homotypic interactions impact DS circuit function

Finally, we investigated the extent to which developmental events controlled by MEGF10 affect DS circuit function. We sought to determine whether the anatomical perturbations caused by loss of MEGF10 – that is, SAC laminar targeting and mosaic spacing errors – alter direction coding by ooDSGCs. To do this, we recorded from wild-type and $Megf10^{-/-}$ retinas on a large-scale multielectrode array (**Field et al., 2007**; **Yu et al., 2017**). ooDSGCs were identified based on their responses to drifting gratings and moving bars (see Materials and methods), which unambiguously distinguished them from other recorded RGCs (**Figure 12A**). Because MEGF10 is not expressed in the adult DS circuit (**Kay et al., 2012**), we could be confident that any mutant physiological phenotypes reflect anatomical changes that arose during development.

These experiments revealed that ooDSGCs with robust direction selectivity were present in both wild-type and $Megf10^{-/-}$ retinas (**Figure 12A,B**), and constituted a similar fraction of the RGC population in both strains (wild-type: 80/609, 13.1%; mutant: 74/551, 13.4%). Furthermore, loss of $Megf10$ did not alter the organization of ooDSGC preferred directions along cardinal axes (**Oyster and Barlow, 1967**), or the fraction of ooDSGCs preferring each direction (**Figure 12—figure supplement 1**). These results are consistent with the observation that mutant SACs remain

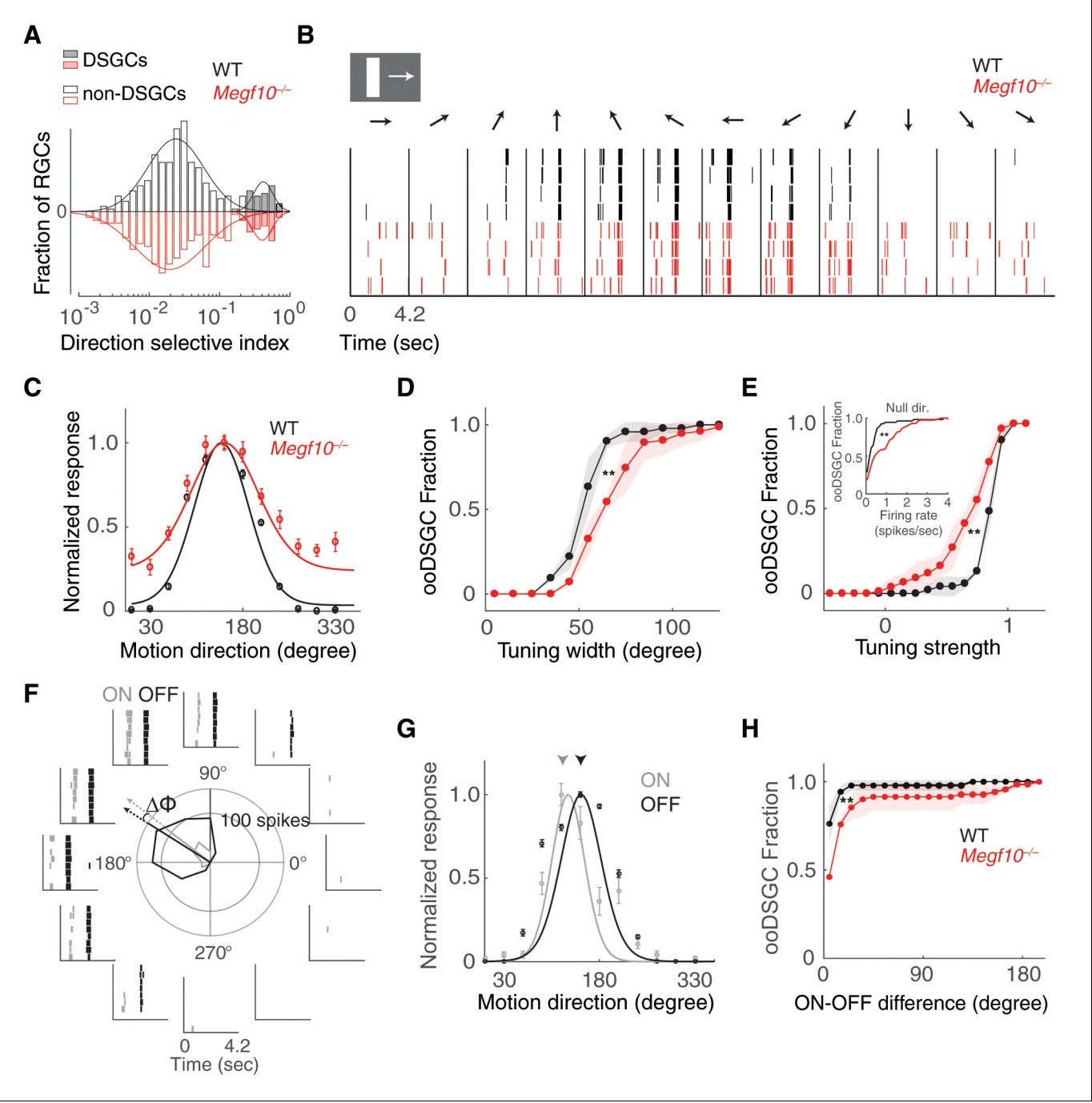

**Figure 12.** Broader and weaker direction tuning of ooDSGCs in *Megf10* mutants. (**A**) Histograms of RGC direction selectivity indices, measured on a multielectrode array, for wild-type (WT, black) and *Megf10⁻/⁻* (red) retinas. Bimodal histograms fit with two-Gaussian mixture model distinguished DSGCs (filled bars) from non-DSGCs (open bars). (**B**) Spike rasters from representative WT and *Megf10⁻/⁻* posterior-preferring ooDSGCs in response to a bright bar moving along 12 directions (arrows). (**C**) Direction tuning curves from cells in B normalized to the maximum response (line: von Mises fit). Non-zero values at tails of mutant curve reflect increase in null-direction spikes (B, left- and right-most bins). (**D,E**) Cumulative distribution of tuning widths (**D**) and tuning strengths (**E**) for all ooDSGCs recorded from two retinas of each genotype (WT *n* = 80 cells; *Megf10⁻/⁻ n* = 74 cells). Mutant ooDSGC population is tuned more broadly (D, right shift of red curve) and more weakly (E, left shift of red curve) than WT. Similar results were obtained when ON and OFF responses were considered separately (not shown). Mutant ooDSGCs also exhibit higher firing rate to null direction motion (E, inset). \*\*p=0.005 (**D**), p=0.003 (**E**), paired KS-test. (**F**) Rasters and polar plot of a representative WT ooDSGC, highlighting preferred directions of ON (gray) and OFF (black) responses (arrows). Δφ, angular difference between preferred directions of ON and OFF responses. (**G**) ON and OFF direction tuning curves for cell in F (line, von Mises fit). ON and OFF preferred directions (arrowheads) are well aligned in WT retina. (**H**) Cumulative distribution across all ooDSGCs of ON-OFF preferred direction difference (Δφ). Same cells as in D,E. Rightward shift of mutant curve indicates larger ON-OFF

*Figure 12 continued on next page*

*Figure 12 continued*

misalignment. **p=0.004, paired KS test. For all panels, background light level was photopic ($10^4$ P*/M-cone/sec; contrast of moving bar was 60%). Error bars/bands, S.E.M. Also see *Figure 12—figure supplement 1*.

DOI: https://doi.org/10.7554/eLife.34241.029

The following figure supplement is available for figure 12:

**Figure supplement 1.** Contrast-dependence of direction-tuning phenotypes in *Megf10*$^{-/-}$ ooDSGCs.

DOI: https://doi.org/10.7554/eLife.34241.030

paired with ooDSGC dendrites and bipolar cell axons even when normal lamination and arbor spacing are disrupted. They indicate that the qualitative functional properties of the circuit are still present.

However, a more careful examination of DS tuning properties in *Megf10*$^{-/-}$ retinas revealed clear quantitative differences in ooDSGC responses. Moving bars were used to measure the width and strength of direction tuning for each identified ooDSGC across the populations recorded on the electrode array (*Figure 12C*). Tuning width was measured as the circular standard deviation of the tuning curve, while tuning strength was measured as the normalized response difference to motion in the preferred and null directions (see Materials and methods). These experiments revealed systematic shifts toward broader (*Figure 12D*) and weaker (*Figure 12E*) direction tuning across the population of ooDSGCs in *Megf10* mutant retinas. This was mainly due to higher null direction spiking among ooDSGCs in mutants (*Figure 12B,C,E*). Furthermore, these effects on tuning width and strength persisted across a broad range of stimulus contrasts (*Figure 12—figure supplement 1*). These results demonstrate that disruption of MEGF10-dependent developmental patterning degrades the precision and strength of ooDSGC direction tuning. They further suggest that perturbations to the anatomical regularly of the circuit across space (e.g. laminar uniformity and SAC spacing) may effectively introduce noise in the DS circuit that broadens and weakens direction tuning (see Discussion).

This idea led us to consider additional functional properties of ooDSGCs that might depend on the spatial regularity of the DS circuit, and therefore might be perturbed in *Megf10* mutants. One such property is the generation of symmetric DS responses to stimuli that are darker or brighter than the background (*Figure 12F,G*). This ON-OFF symmetry allows the DS response to be largely insensitive to contrast reversals (*Amthor and Grzywacz, 1993*); it arises because ooDSGCs receive highly symmetric SAC inputs in both ON and OFF sublayers (*Figure 1A*). In *Megf10* mutants, ON-OFF anatomical symmetry is disturbed, because ON and OFF SAC errors are not spatially correlated (*Figure 8A–C*). We hypothesized that this might lead to disparities in the direction tuning of individual cells' ON and OFF responses. Indeed, *Megf10*$^{-/-}$ ooDSGCs exhibited greater separation (i.e. less coherence) between their ON and OFF preferred directions than wild-type ooDSGCs, across a broad range of contrasts (*Figure 12H*; *Figure 12—figure supplement 1*). These results support the idea that MEGF10 serves to establish a highly uniform and regular network of SAC dendrites (via controlling both the precise timing of INL lamination and through regularizing inter-SAC spacing), the net effect of which is to allow greater precision and coherence in the direction tuning of ooDSGCs.

## Discussion

Neural circuits typically consist of multiple cell types born at different places and times, raising the question of how circuit partners manage to converge at a common site for selective synapse formation. Here, we describe a developmental strategy that the retinal DS circuit uses to solve this problem. We show that SACs coordinate amongst themselves to assemble a dendritic scaffold that subsequently recruits projections from their DS circuit partners. By identifying for the first time a genetic manipulation – loss of *Megf10* – that causes SACs to misproject outside their two typical IPL layers, we uncover mechanisms by which SACs assemble this dendritic scaffold. Further, we use *Megf10* mutants to examine the effects on DS circuit anatomy and function when the SAC scaffold is disrupted. We find that MEGF10 establishes DS circuit spatial homogeneity across the retina, both by controlling IPL innervation patterns and by positioning SAC cell bodies. In *Megf10* mutants, disruptions in circuit homogeneity occur with minimal effects on radial SAC dendrite anatomy or synaptic partnering, making the phenotype unique among DS circuit developmental mutants. Finally, we

find that this abnormal spatial pattern degrades DS circuit function by broadening the range of directions to which ooDSGCs will respond, and by weakening overall direction selectivity. These results provide new insight into general strategies for circuit development, as well as the specific mechanisms that ensure functional assembly of the DS circuit.

## Homotypic recognition as a mechanism regulating dendrite differentiation

During radial migration, newborn central nervous system neurons have a multipolar morphology, but on arrival at their final position within the tissue they become highly polarized (*Nadarajah et al., 2001*; *Tabata and Nakajima, 2003*; *Cooper, 2014*; *Chow et al., 2015*; *Krol et al., 2016*; *Hinds and Hinds, 1978*). This morphological change enables elaboration of dendrites and integration into local circuitry. If dendrite differentiation begins early, migration is impaired (*Hoshiba et al., 2016*), suggesting that the transition from migratory to mature morphology must be highly regulated to ensure that neurons only differentiate once they arrive at their final position. The extracellular cues that signal arrival are poorly understood in most nervous system regions.

Here, we show that SACs use homotypic recognition, mediated by MEGF10, to initiate IPL-directed dendrite morphogenesis. When deprived of homotypic neighbors or MEGF10, SACs at the IPL retain a multipolar morphology (compare *Figure 2C* to *Figures 3F* and *6A*) instead of polarizing arbors toward the IPL. This indicates that the transition from migratory to mature morphology is impaired in the absence of SAC homotypic recognition. We show that migrating SACs first establish homotypic contact upon arrival at the inner retina. At this stage, they are still multipolar (*Figure 2D*), but they orient primary dendrites tangentially within the INBL to ultimately contact their SAC neighbors. These contacts occur prior to IPL innervation, and are required for it to occur in a timely manner. SACs lacking neighbors or the molecular means to detect them (i.e. MEGF10) appear to persist in this multipolar soma-layer-targeting phase, causing over-innervation of the INL/GCL and delaying IPL innervation (*Figure 6G*). Thus, establishment of homotypic contact is a key checkpoint for the progression of SAC dendrite differentiation and IPL sublayer morphogenesis.

We propose that the function of this checkpoint is to ensure that SACs elaborate dendrites only when they have arrived adjacent to the IPL. The presence of other SACs that have already completed their migration is a reliable indicator of arrival in the proper location. Because soma-layer SAC contacts appear earliest, and because MEGF10 selectively influences IPL innervation during the period when they exist, we favor the notion that the key homotypic interactions occur through these arbors. However, we cannot exclude that IPL-based interactions also play a role. INL-directed arbors resembling those we describe can be discerned in many developing zebrafish amacrine cells (*Godinho et al., 2005*; *Chow et al., 2015*), raising the possibility that this mechanism applies across species and across other amacrine cell types. Because most neurons require a way to control when and where they differentiate, we anticipate that this homotypic contact strategy, or variations upon it, may have important roles in the differentiation of other CNS neurons at the completion of their radial migration.

## MEGF10 as the signal mediating SAC homotypic recognition

We conclude that MEGF10 is the molecule responsible for homotypic recognition during SAC IPL innervation. Four key results support this conclusion. First, MEGF10 is expressed at the right time and place to assume this role: It is expressed selectively in SACs (*Figure 1*), upon conclusion of their radial migration, and in the soma-layer arbors that we propose mediate recognition (*Figure 4*). Second, *Megf10* null mutant SACs phenocopy the dendrite polarization errors seen in solitary Ptf1a-cKO SACs, suggesting that homotypic recognition requires *Megf10*. Third, co-immunoprecipitation experiments indicate that MEGF10 interacts with itself via its extracellular domain, suggesting it could act as both ligand and receptor. While this biochemical interaction may take place in the *cis* configuration, the fourth line of evidence indicates that MEGF10 interacts in trans as well: Using a conditional-null *Megf10* allele in vivo, we show that MEGF10 is required on the cell that sends homotypic signals as well as the cell receiving those signals. Loss of MEGF10 on either side leads to dendritic phenotypes resembling solitary SACs and *Megf10* null mutants. Together, these data are consistent with a model whereby SAC-SAC contact initiates a transcellular MEGF10 homophilic

interaction, in which MEGF10 serves as both receptor and ligand to trigger the switch from migratory to mature morphology (see model, *Figure 7K*).

This homophilic model of MEGF10 function is consistent with its role during establishment of mosaic cell body patterning (*Kay et al., 2012*). In that context, MEGF10 acts as ligand and receptor to mediate cell-cell repulsion, thereby spacing SAC somata evenly across the retina. Here we discover a second MEGF10 function in SAC IPL innervation. Because the two SAC phenotypes have different underlying cell biology (soma movement vs. dendrite dynamics), and separable temporal requirements for MEGF10 function (*Figure 9*), it seems unlikely that they reflect disruption of a single biological event. Instead, MEGF10 appears to act at distinct, albeit partially overlapping times, to control different aspects of SAC development, each of which are regulated by contact with homotypic neighbors (see model, *Figure 9C*).

## Formation of SAC IPL sublayers

Our results shed light on the mechanisms controlling SAC dendrite lamination. While repulsion mediated by Sema6a and PlexinA2 prevents OFF SACs from straying to the ON sublayer (*Sun et al., 2013*), molecules required for formation of the SAC sublayers have not been identified. We show that SACs deprived of homotypic neighbors or MEGF10 initially fail to form IPL sublayers, and when they eventually do so, their strata are riddled with errors. Both the lack of sublayers at early stages and the dendritic mistargeting to inappropriate sublayers at maturity are novel SAC phenotypes; they implicate MEGF10 as a key player in forming SAC IPL sublayer-specific projections.

It is generally assumed that sublayer formation has two basic molecular requirements: 1) Attractive/adhesive molecules that mediate co-fasciculation of stratified arbors; and 2) repulsive cues that prevent straying of arbors into other sublayers (*Lefebvre et al., 2015*; *Sanes and Yamagata, 2009*). Our MEGF10 studies suggest an additional, earlier requirement for cell-cell interactions that occur prior to neuropil innervation. The purpose of this surprisingly early SAC-SAC interaction, we propose, is to ensure that SACs grow dendrites at the right time and place to co-fasciculate with their SAC neighbors. The molecular basis of this homotypic co-fasciculation – clearly another essential player in sublayer formation – remains to be determined. MEGF10 is probably not involved; the co-fasciculation system appears intact in *Megf10* mutants given that sublayers do eventually form. Perhaps this system is part of the mechanism that compensates for loss of MEGF10 to ultimately generate the sublayers.

When IPL arborization is delayed by loss of *Megf10*, two SAC errors ensue. First, SACs generate mistargeted dendritic material that appears to persist as ectopic IPL sublayers. Second, SACs never completely innervate their sublayers, resulting in fragmented IPL strata. These two errors are caused by delays rather than an ongoing requirement for MEGF10 during later stages of arbor growth, as shown by conditional mutant experiments (*Figure 9*). Thus, our findings support the idea that timing is critical to the sequential lamination of the IPL: When SAC dendrites arrive in the IPL too late, they encounter a different cellular and molecular milieu that may not support the proper development of their arbors. In this view, the normal role of MEGF10 in DS circuit assembly is to instigate SAC dendrite outgrowth at the crucial time when laminar self-assembly can occur.

SACs may face an additional obstacle to overcoming their delayed IPL innervation in *Megf10* mutants: abnormal soma positioning. While mosaic spacing errors do not account for the *Megf10* mutant ectopic IPL phenotype, we cannot exclude the possibility that the placement of IPL arbor gaps might be at least partly explained by soma position. If SACs are struggling to make up for their delayed IPL innervation, it is plausible that increasing the distance between SACs (as happens sporadically due to random positioning) might further hinder the development of complete retinal coverage.

## SACs as a scaffold for DS circuit assembly

Because of their early stratification, SAC dendrites have been proposed to act as a scaffold that guides assembly of the DS circuit (*Stacy and Wong, 2003*). A key prediction of this model is that laminar targeting of later-stratifying cell types should depend on the existence of this scaffold. We show using a SAC-specific manipulation – removal of *Megf10* – that disruption of SAC stratification causes their bipolar and ooDSGC circuit partners to make corresponding projection errors. Based on the kinds of errors we observed, SACs appear to provide attractive, permissive, and possibly even

repulsive arbor sorting cues to influence the laminar positioning of their circuit partners. This work thus constitutes the first critical test of the scaffolding model, and provides strong support for it. We find that SACs use homotypic interactions to initiate formation of their circuit sublayers, and then heterotypic interactions to recruit circuit partners to join them. SACs might achieve their scaffolding functions directly, by providing guidance cues to their partners; or they may do so indirectly, by patterning the IPL projections of an intermediary cell type that in turn guides later-arriving projections. Direct scaffolding may be mediated in part by Cadherins 8 and 9, which regulate interactions between SAC dendrites and DS circuit bipolar cell axons (*Duan et al., 2014*). Molecular mediators of ooDSGC-SAC dendrite interactions remain to be identified.

Evidence that the SAC scaffold can be repulsive – or at least can exclude bipolar arbors from certain IPL regions – came from our observations of BC axon anatomy. In wild-type retina, we were surprised to note how completely the BC3a, BC5, and BC7 axon terminals were excluded from the SAC territory – they contacted it but did not enter (*Figure 11B–D*; *Figure 11—figure supplement 1*). This behavior stands in stark contrast to the behavior of ooDSGC dendrites, which completely overlapped SACs (*Figure 10A–B*; *Vaney and Pow, 2000*). Moreover, in *Megf10* mutants, the laminar distance between BC5 and BC7 terminals was reduced in the absence of SAC arbors, and increased in the presence of SAC ectopias, further suggesting the existence of local SAC-BC repulsion. The finding that SACs exclude bipolar circuit partners from their sublayers appears at first counterintuitive. But given that no bipolar cell type is exclusively devoted to the DS circuit (*Wässle et al., 2009*; *Greene et al., 2016*; *Kim et al., 2014*), a mechanism must exist to ensure that they can also contact non-DS partners. We speculate that SACs initially recruit their bipolar partners using long-range attractive cues, and then use contact-repulsion (or an equivalent arbor sorting mechanism) to displace bipolar arbors such that they remain in contact with the SAC layers but also innervate adjacent layers. This model is consistent with bipolar arbor phenotypes in *Megf10* mutants, but will require further study.

## Role of MEGF10 in the functional assembly of DS circuitry

We found that impairment of SAC interactions in the perinatal retina causes permanent functional DS circuit deficits. In *Megf10* mutants, direction tuning of ooDSGCs becomes broader and weaker, and their ON/OFF preferred directions are less aligned. Direction tuning is degraded in large part because mutant ooDSGCs have aberrant spiking responses to null-direction stimuli. This suggests that impaired null-direction inhibition – which arises from SACs – is a key contributor to the phenotype. Broader ooDSGC tuning curves have been shown, in modeling studies, to degrade population-level coding of directional information, and the ability of downstream neurons to extract such information (*Fiscella et al., 2015*). Thus, the physiological phenotypes we identified are likely sufficient to impair the ability of mutant retina to appropriately relay visual information.

Dysfunctional DS circuit physiology in *Megf10* mutants is almost certainly a consequence of its effects on development, because neurons do not express MEGF10 beyond the second postnatal week (*Kay et al., 2012*). Further, even though MEGF10 is expressed by Müller glia in adulthood, we have been unable to detect any changes in Müller glia anatomy or interactions with DS circuit synapses upon loss of *Megf10* function (*Wang et al., 2017*; J.W. and J.N.K., unpublished observations). We therefore conclude that anatomical changes to the DS circuit arising during development are responsible for circuit dysfunction.

The fundamental change to DS circuit anatomy in *Megf10* mutants is altered distribution of arbors and synapses, unlike other manipulations which simply serve to destroy SAC radial morphology or disrupt synaptic partnering among DS circuit cells (*Sun et al., 2013*; *Duan et al., 2014*; *Kostadinov and Sanes, 2015*; *Peng et al., 2017*). In *Megf10* mutants, the combined effect of mosaic spacing defects and IPL laminar targeting errors is to disturb the regularity of SAC IPL innervation. As a result, some parts of the visual map become over-innervated (e.g. *Figure 10C*) while others are uninnervated (*Figure 10D*). In turn, ooDSGCs are recruited to the over-innervated regions and excluded from uninnervated gaps, likely causing sporadic local inhomogeneity in synapse density across visual space. According to some models of DS, which posit that the total amount of SAC inhibition is the key factor underlying DS responsiveness, these relatively small-scale changes would be considered unlikely to change circuit function (*Taylor and Vaney, 2002*; *Demb, 2007*). A more recent alternate view is that the fine spatial arrangement of glutamatergic inputs to SACs, and the synaptic balance of SAC and bipolar input onto ooDSGC dendrites, are both important for DS

responses (*Ding et al., 2016*; *Vlasits et al., 2016*; *Poleg-Polsky and Diamond, 2016*; *Sethuramanujam et al., 2016*; *Sethuramanujam et al., 2017*). The finding that *Megf10* mutants have DS tuning phenotypes suggests that local synaptic arrangements are indeed important for the DS computation. More broadly, this finding shows that the developmental mechanisms we describe here are important for enabling circuit function, raising the possibility that other circuits throughout the retina and CNS may use similar developmental mechanisms to establish their functional connectivity.

# Materials and methods

**Key resources table**

| Reagent type | Designation | Source or reference | Identifier | Additional information |
|---|---|---|---|---|
| Antibody | Megf10: rabbit, 1:1000 | *Kay et al. (2012)* | | |
| Antibody | Sox2: rabbit, 1:500 | Abcam | ab97959 | |
| Antibody | Sox2: goat, 1:500 | Santa Cruz | sc-17320 | |
| Antibody | ChAT: goat, 1:400 | EMD Millipore | AB144P | |
| Antibody | Beta Galactosidase: rabbit, 1:5000 | other | | Antibody was a gift of J.R. Sanes, Harvard |
| Antibody | GFP: chicken, 1:1000 | Life Technologies | A10262 | |
| Antibody | GFP (Co-IP): rabbit, 1:1000 | Thermo Fisher Scientific | A-6455 | |
| Antibody | AP-2a: mouse, 1:200 | Developmental Studies Hybridoma Bank | 3B5 | |
| Antibody | RBPMS: guinea pig, 1:2000 | other | | Antibody was a gift of N. Brecha, UCLA |
| Antibody | Chx10: sheep, 1:300 | Exalpha | X1180P | |
| Antibody | Chx10: goat, 1:500 | Santa Cruz | sc-21690 | |
| Antibody | GAD65: rabbit, 1:1000 | Millipore | AB1511 | |
| Antibody | FLAG: mouse, 1:500 | Sigma Aldrich | F-1804 | |
| Antibody | VGLUT3 guinea pig | synaptic systems | 135 204 | |
| Antibody | Synaptotagmin-2 (Syt2), mouse, 1:250 | Zebrafish International Resource Center | ZDB-ATB-081002–25 | |
| Antibody | Isl1: mouse, 1:25 | Developmental Studies Hybridoma Bank | 39.4D5 | |
| Antibody | Internexin: rabbit, 1:1000 | EMD Millipore | AB5354 | |
| Antibody | Normal Rabbit IgG | Cell Signaling Technology | 2729S | |
| Antibody | Normal Mouse IgG | Cell Signaling Technology | 5415S | |
| Antibody | Alexa Fluor 488 AffiniPure Donkey Anti-Chicken: 1:1000 | Jackson Immuno Research | 703-545-155 | |
| Antibody | Alexa Fluor 488 AffiniPure Donkey Anti-rabbit: 1:1000 | Jackson Immuno Research | 711-545-152 | |

*Continued on next page*

*Continued*

| Reagent type | Designation | Source or reference | Identifier | Additional information |
|---|---|---|---|---|
| Antibody | Alexa Fluor 488 AffiniPure Donkey Anti-goat: 1:1000 | Jackson Immuno Research | 705-545-147 | |
| Antibody | Alexa Fluor 488 AffiniPure Donkey Anti-mouse: 1:1000 | Jackson Immuno Research | 706-605-148 | |
| Antibody | Alexa Fluor 647 AffiniPure Donkey Anti-rabbit: 1:1000 | Jackson Immuno Research | 705-605-147 | |
| Antibody | Cy3-AffiniPure Donkey Anti-rabbit: 1:1000 | Jackson Immuno Research | 715-165-151 | |
| Antibody | Cy3-AffiniPure Donkey Anti-Guinea Pig: 1:1000 | Jackson Immuno Research | 706-165-148 | |
| Antibody | Cy3-AffiniPure Donkey Anti-Goat: 1:1000 | Jackson Immuno Research | 705-165-147 | |
| Antibody | IRDye 680RD Donkey anti-Mouse IgG (H + L): 1:1000 | Li-Cor Biosciences | 925–68072 | |
| Antibody | IRDye 800CW Donkey anti-Rabbit IgG (H + L): 1:1000 | Li-Cor Biosciences | 925–32213 | |
| Biological sample (AAV) | AAV9.hEF1a.lox.TagBFP. lox.eYFP.lox.WPRE. hGH-InvBYF(Harvard) | Penn Vector Core | AV-9-PV2453 | |
| Biological sample (AAV) | AAV9.hEF1a.lox.mCherry. lox.mTFP1.lox.WPRE.hGH-InvCheTF(Harvard) | Penn Vector Core | AV-9-PV2454 | |
| Chemical compound | Fetal Bovine Serum | Life Technologies | 16250–078 | |
| Chemical compound | Opti-MEM I Reduced Serum Medium | Thermo Fisher Scientific | 31985070 | |
| Chemical compound | Polyethylenimine (PEI), Linear (MW 25,000) | VWR/Polysiciences | 23966–2 | |
| Chemical compound | 16% Paraformaldehyde | Electron Microscopy Sciences | 15710 | |
| Chemical compound | Normal Donkey Serum | Jackson ImmunoResearch | 017-000-121 | |
| Other | Immun-Blot Low Fluorescence PVDF membrane | Bio-Rad | 1620264 | |
| Chemical compound | Fluoromount G | SouthernBiotech | 0100–01 | |
| Chemical compound | Hoechst 33258 | Invitrogen | H21491 | |
| Chemical compound | Isothesia: Isoflurane | Henry Schein | 11695–6776 | |
| Chemical compound | Tissue Freezing Medium | VWR | 15148–031 | |
| Chemical compound | Acrylamide/Bis solution | Bio-Rad | 161–0158 | |
| Chemical compound | 4x Laemmli Sample Buffer | Bio-Rad | 1610747 | |
| Chemical compound | Immun-Blot Low Fluorescence PVDF membrane | Bio-Rad | 1620264 | |
| Chemical compound | Odyssey Blocking Buffer | Li-Cor Biosciences | 927–40000 | |
| Chemical compound | Dynabeads Protein G for Immunoprecipitation | Thermo Fisher Scientific | 10003D | |

*Continued on next page*

*Continued*

| Reagent type | Designation | Source or reference | Identifier | Additional information |
|---|---|---|---|---|
| Chemical compound | cOmplete, Mini, EDTA-free Protease Inhibitor Cocktail Tablets | Roche | 4693159001 | |
| commercial assay or kit | Bio-Rad DC Protein Assay Kit | Bio-Rad | 5000112 | |
| Strain (mus musculus) | Mouse: *Megf10*$^{LacZ}$ | **Kay et al. (2012)** | *Megf10*$^{tm1b(KOMP)Jrs}$ | |
| Strain (mus musculus) | Mouse: *Megf10*$^{flox}$ | this study | *Megf10*$^{tm1c}$ | see Methods, 'Animals' section |
| Strain (mus musculus) | Mouse: Ptf1a-cKO | **Krah et al., 2015** | *Ptf1a*$^{tm3Cvw}$ | |
| Strain (mus musculus) | Mouse: *Isl1*$^{Cre}$ | Jackson Labs 024242 | *Isl1*$^{tm1(cre)Sev/J}$ | |
| Strain (mus musculus) | Mouse: Hb9-GFP | Jackson Labs 005029 | B6.Cg-*Tg (Hlxb9-GFP)1Tmj/J* | |
| Strain (mus musculus) | Mouse: *ChAT*$^{Cre}$ | Jackson Labs 006410 | *Chat*$^{tm2(cre)Lowl}$ | |
| Strain (mus musculus) | Mouse: Six3-Cre | Jackson Labs 019755 | Tg(Six3-cre)69Frty | |
| Strain (mus musculus) | Mouse: *Kcng4*$^{Cre}$ | Jackson Labs 029414 | *Kcng4*$^{tm1.1(cre)Jrs}$ | |
| Strain (mus musculus) | Mouse: Drd4-GFP | **Huberman et al. (2009)** | Tg(Drd4-EGFP)W18Gsat | |
| Strain (mus musculus) | Mouse: Gjd2-GFP | MMRRC | Tg(Gjd2-EGFP)JM16Gsat/Mmucd; RRID:MMRRC_030611-UCD | |
| Strain (mus musculus) | Mouse: *Rosa26*$^{mTmG}$ | Jackson Labs 007676 | Gt(ROSA) 26Sor$^{tm4(ACTB-tdTomato,-EGFP)Luo}$ | |
| Strain (mus musculus) | Mouse: *Rosa26*$^{fGFP}$ | **Rawlins et al. (2009)** | Gt(ROSA)26Sor$^{tm1(CAG-EGFP)Blh}$ | |
| Strain (mus musculus) | Mouse: *Rosa26*$^{Ai14}$ | Jackson Labs 007914 | B6.Cg-*Gt(ROSA) 26Sor*$^{tm14(CAG-tdTomato)Hze/J}$ | |
| Strain (mus musculus) | Mouse: *ACTB:FLPe* | Jackson Labs 003800 | B6;SJL-*Tg(ACTFLPe) 9205Dym/J* | |
| Strain (mus musculus) | Mouse: Gad1-GFP | Jackson Labs 007677 | Tg(Gad1-EGFP)G42Zjh | |
| Strain (mus musculus) | Mouse: C57Bl6/J | Jackson Labs 000664 | C57BL/6J | |
| Cell line (human) | HEK293T | ATCC | 293T (ATCC CRL-3216) | Cell line was authenticated by ATCC at time of purchase |
| Recombinant DNA reagent | CMV-M10-FLAG | this paper | | see Methods, 'Generation of MEGF10-ΔICD constructs' section |
| Recombinant DNA reagent | CMV-M10-GFP | **Kay et al. (2012)** | | |
| Recombinant DNA reagent | pCMV-MEGF10-ΔICD-GFP | **Kay et al. (2012)** | | |

*Continued on next page*

*Continued*

| Reagent type | Designation | Source or reference | Identifier | Additional information |
|---|---|---|---|---|
| Recombinant DNA reagent | MEGF10-ΔICD-Flag | this paper | | see Methods, 'Generation of MEGF10-ΔICD Constructs' section |
| Recombinant DNA reagent | pAAV-EF1a-Brainbow-tagBFP-EYFP-WPRE | Addgene | 45185 | |
| Recombinant DNA reagent | pAAV-EF1a-Brainbow-mTFP1-Cherry-WPRE | Addgene | 45816 | |
| Software | Fiji/ImageJ | *Schindelin et al. (2012)* | | |
| Software | Prism | GraphPad | | |
| Software | NIS Elements | Nikon Instruments | | |
| Software | Custom JAVA scripts for spike sorting | *Yu et al., 2017* | | |
| Software | Custom MATLAB scripts for data analysis | this paper | https://github.com/Field-Lab/megf10-dstuning | see Methods, 'Multielectrode Array Recordings'; 'Quantification and Statistical Analysis' sections |
| Software | Matlab | Mathworks, Natick, MA | | |
| Software | Image Studio™ | LI-COR Biosciences | | |
| Software | Photoshop | Adobe | | |
| Sequence-based reagent | M10flagNotl_Rev | IDT | ATAGCGGCCGCtta CTTGTCGTC ATCGTCTTTGT AGTCttcactg ctgctgctgctgctg | |
| Sequence-based reagent | M10flag_Fwd | IDT | GGTACATGCCT GTGCGAAGCA | |
| Sequence-based reagent | Cyto9_flag_Rev1 | IDT | 5'ATAGCGGC CGCttaCTTGT CGTCATCGTCT TTGTAGTC TTCCTTCCTCT TCTGCTTGTGT | |

## Animals

All animal experiments were reviewed and approved by the Institutional Animal Care and Use Committee of Duke University. The animals were maintained under a 12 hr light-dark cycle with *ad lib* access to food and water. Retinas from adult (4–8 weeks old) $Megf10^{-/-}$ mutant mice and wild-type control mice with same genetic background were used for experiments performed on the multielectrode array (MEA). Animals were dark-adapted overnight prior to the experiment.

For this study, the following transgenic and mutant mouse lines were used: (1) $Megf10^{tm1b(KOMP)Jrs}$ (**Kay et al., 2012**), referred to as $Megf10^-$ or $Megf10^{lacZ}$; (2) $Ptf1a^{tm3Cvw}$ (**Krah et al., 2015**), referred to as $Ptf1a^{flox}$ or (when crossed to Cre mice) Ptf1a-cKO; (3) $Isl1^{tm(cre)Sev}$ (**Yang et al., 2006**), referred to as $Isl1^{Cre}$; (4) $Tg(Hlxb9-GFP)1Tmj/J$ (**Trenholm et al., 2011**), referred to as Hb9-GFP; (5) $Chat^{tm2(cre)Lowl}$ (**Rossi et al., 2011**), referred to as $Chat^{Cre}$; (6) $Tg(Six3-cre)69Frty$ (**Furuta et al., 2000**) referred to as Six3-Cre; (7) $Kcng4^{tm1.1(cre)Jrs}$ (**Duan et al., 2014**) referred to as $Kcng4^{Cre}$; (8) $Tg(Drd4-EGFP)W18Gsat$ (**Huberman et al., 2009**), referred to as Drd4-GFP; (9) $Tg(Gjd2-EGFP)JM16Gsat$, referred to as Gjd2-GFP; (10) $Tg(Gad1-EGFP)G42Zjh$, referred to as Gad1-GFP. Two Cre reporter strains were used that express membrane-targeted green fluorescent protein (mGFP) upon Cre recombination: (1) $Gt(ROSA)26Sor^{tm4(ACTB-tdTomato,-EGFP)Luo}$, also known as mT/mG (**Muzumdar et al., 2007**); (2) $Rosa26^{fGFP}$ (**Rawlins et al., 2009**). An additional Cre reporter strain was used that expresses tdTomato fluorescent protein upon Cre recombination: $Gt(ROSA)26Sor^{tm14}$

*(CAG-tdTomato)Hze* (*Madisen et al., 2010*). See Key Resources table for repository stock numbers where applicable.

To produce *Megf10^flox* mice, *Megf10^tm1a(KOMP)Jrs* mice (*Kay et al., 2012*) were crossed to germ-line Cre strain *B6;SJL-Tg(ACTFLPe)9205Dym/J*, thereby generating a functional allele (also known as *Megf10^tm1c*) in which exon four was flanked by loxP sites.

## Cell culture

HEK293T cells were obtained from, validated by, and mycoplasma tested by ATCC. The cells were cultured in Dulbecco's Modified Eagle's Medium (DMEM) with 10% bovine growth serum, 4.5 g/L D-glucose, 2.0 mM L-glutamine, 1% Penicillin/Streptomycin in 10 cm cell culture dishes. Cells were passaged every 2–3 days to reach confluence. Before splitting, culture media were removed and Dulbecco's phosphate-buffered saline (D-PBS) was used to rinse cell layers as well as removing residual serum. Cells were detached from dish with 4 ml of 0.05% Trypsin and incubated at 37°C until cell layer is dispersed (about 5 min). Equal volume of complete culture media was added to the dish to inhibit protease activity. The suspension was centrifuged at 200 x g for 5 min. Supernatant was aspirated and the cells were suspended with appropriate amount of media and plated (1:4-1:8). Cells used for experiments were passaged no more than 10 times. Cell stocks were stored as 2 million cells per vial in complete culture media with 10% DMSO in liquid nitrogen.

## Identification of DS circuit cell types using antibody and transgenic markers

### SAC markers in mature retina

Antibodies to choline acetyltransferase (ChAT) were used as a SAC marker in mice older than P5. This antibody stains SAC somata and their dendrites in the IPL (e.g. *Figure 8A*).

### SAC markers in embryonic and neonatal retina

Antibodies to ChAT and vesicular acetylcholine transporter, typically used as SAC markers in the mature retina, do not stain reliably in the embryonic and neonatal (P0-P3) mouse retina, precluding their use as markers during one of the key time periods of this study. We therefore characterized several other SAC markers that we found to be suitable for definitive SAC identification and their anatomical characterization in the E16-P3 period:

The *Megf10^lacZ* allele (*Kay et al., 2012*) drives strong, selective β-galactosidase (βgal) expression in all SACs starting at embryonic day (E)17 (*Figure 1B*; *Figure 1—figure supplement 1*; data not shown). Horizontal cells are also labeled. Expression is strong enough to allow characterization of SAC dendrite anatomy at these early stages. Antibodies to *Megf10* yield a similar staining pattern (*Figure 1—figure supplement 1*; *Figure 4B,C*), but staining of fine dendritic arbors was brighter with anti-βgal staining of *Megf10^lacZ* mice, so this approach was used for most of our anatomical experiments analyzing the full SAC population at or before P3. In some such experiments, a *Megf11^lacZ* allele (*Kay et al., 2012*) was also present; this allele drives βgal expression in essentially the same pattern as *Megf10^lacZ* and therefore contributed to signal brightness. The presence of this allele had no apparent effect on SAC anatomy, in either wild-type or *Megf10* mutant background.

Antibodies to Sox2 (*Whitney et al., 2014*) strongly label all SAC nuclei in the INL and GCL, starting at embryonic stages (*Figure 1—figure supplement 1*; *Figure 2D,E,G*). Progenitor cells in the ONBL are also labeled. This marker was typically used in conjunction with *Megf10^lacZ* to provide definitive identification of SACs as βgal⁺Sox2⁺ cells.

Antibodies to internexin label SAC intermediate filaments, which localize in a polarized manner to the primary dendrite(s) and the side of the cell body from which they emerge (*Figure 2—figure supplement 1*). Primary dendrites were defined as any first-order dendrite branch, that is those arising directly from the cell body. Internexin is a selective marker of SAC in perinatal mouse retina, as previously shown in tree shrew (*Knabe et al., 2007*). RGC axons are also labeled (*Figure 2—figure supplement 1*).

Antibodies to Isl1 (*Figure 2A*) label all SAC nuclei, starting at cell cycle exit (*Galli-Resta et al., 1997*). A large subset of RGCs are also labeled. The *Isl1^Cre* knock-in mouse (*Yang et al., 2006*) faithfully recapitulated this expression pattern (*Figure 2A,B*) and was used to study SAC anatomy at embryonic stages (see below for further details).

## SAC single-cell labeling

To assess the single-cell morphology of individual SACs during early postnatal development, the $Chat^{Cre}$ line was used. In contrast to mature retina (e.g. *Figure 8B*), in which all SACs were labeled, $Chat^{Cre}$ expression was rare and sporadic in early postnatal retina (*Figure 1C*; *Figure 1—figure supplement 1*), as reported previously (*Xu et al., 2016*). Therefore, when crossed with Cre reporter mice to make $Chat^{mG}$ animals, the full anatomy of individual SACs was clearly delineated (e.g. *Figure 2H–K*). We did not typically observe Cre recombination in non-SAC cell types; nevertheless, we always co-stained with another SAC marker, either Sox2 or *Megf10*:βgal, to confirm the SAC identity of the cells that were analyzed.

## ooDSGC markers

Two mouse lines were used, each of which labels distinct types of ooDSGCs. *Hblx9-GFP* (referred to as Hb9-GFP throughout the manuscript) labels the superior subtype of ooDSGC, while *Drd4-GFP* labels the posterior subtype of ooDSGC (*Trenholm et al., 2011*; *Huberman et al., 2009*).

## DS-circuit bipolar cell markers

Four types of bipolar cells have been shown to make monosynaptic connections with SACs and/or ooDSGCs: Types BC2, BC3a, BC5, and BC7 (*Duan et al., 2014*; *Ding et al., 2016*; *Greene et al., 2016*; *Kim et al., 2014*; *Chen et al., 2014*). OFF bipolar cells BC2 and BC3a were labeled, respectively, by antibodies to Syt2 and HCN4 (*Wässle et al., 2009*).

ON bipolar cells BC5 and BC7 were marked with $Kcng4^{Cre}$ (*Duan et al., 2014*) crossed to mGFP Cre reporter mice (denoted $Kcng4^{mG}$). Labeling of BC7 was more prominent with the Rosa26 locus mGFP Cre reporter line that we used, compared to the cytosolic GFP reporter driven by Thy1 that was used by *Duan et al. (2014)*.

*Gjd2-GFP* was also used to label BC5 bipolar cells (*Figure 11—figure supplement 1*). In adult retina, GFP was strongly expressed by a bipolar cell type that ramified in a laminar location typical of BC5 (Sidney Kuo, University of Washington, personal communication). We confirmed this expression pattern; weak expression in amacrine cells was also noted (*Figure 11—figure supplement 1*). At earlier developmental stages the amacrine cell staining was much stronger and filled many amacrine processes throughout the IPL, precluding use of this line for developmental studies of bipolar axons (M. Stogsdill and J.N.K, unpublished observations).

## Immunohistochemistry

### Retinal cross sections

Mice were anesthetized by isoflurane or cryoanesthesia (neonates only) followed by decapitation. Eyes were enucleated, washed in PBS, and fixed in PBS containing 4% formaldehyde (pH 7.5) for 1.5 hr at 4°C. After fixation, eyes were washed 3X with PBS and stored in PBS containing 0.02% sodium azide at 4°C until further processing. Retinas were dissected from the eyecup, cryoprotected by equilibration in PBS containing 30% sucrose, then embedded in Tissue Freezing Medium and frozen by submersion in 2-methylbutane chilled by dry ice. Tissue sections were cut on a cryostat to 20 μm and mounted on Superfrost Plus slides. Slides were dried on a slide warmer for 1 hr then stored at −80°C or used immediately.

For antibody labeling, slides were washed for 5 min with gentle agitation in PBS to remove embedding medium and blocked for 1 hr in PBS + 0.3% Triton X-100 (PBS-Tx) containing 3–5% normal donkey serum. Primary antibodies were diluted in blocking buffer, added to slides, then incubated overnight at 4° C. Slides were washed with PBS 3X for 10 min followed by incubation with secondary antibody diluted in PBS-Tx for 1–2 hr at RT. Slides were washed again with PBS 3X for 10 min then coverslipped using Fluoromount G.

### Retinal whole-mounts

Tissue was processed as above up to the point of dissection from the eyecup. After dissection from eyecup, retinas were washed in PBS then blocked for 3 hr with agitation at 4° C in blocking buffer (constituted as described above). Primary antibodies were diluted in blocking buffer, added to retinas, and incubated for 5–7 days with gentle agitation at 4°C. Retinas were washed 3X with PBS over the course of 2 hr with gentle agitation. Secondary antibody was diluted in PBS containing 0.3%

Triton X-100 and was added to retinas followed by incubation overnight at 4° C with gentle agitation. Retinas were washed again 3X in PBS over the course of 2 hr with gentle agitation. For mounting on slides, four radial incisions separated by 90° were made centripetally, approximately 1/3 the radius of the retina. Retinas were flattened on nitrocellulose paper photoreceptor side down and coverslipped with Fluoromount G.

## Image acquisition and processing

Sections and whole-mounts were imaged on a Nikon A1 or an Olympus FV300 confocal microscope. Image Z-stacks (Z-resolution 0.4–0.5 μm for whole-mount images; 0.8–1.0 μm for cross-sections) were imported to Fiji (*Schindelin et al., 2012*), de-noised by median-filtering (0.5–2.0 pixel radius), and projected to a single plane. The portion of the stack selected for maximum-intensity projection was determined by the Z-volume of the structure to be depicted in the final image. Except where noted, data analysis and quantification was only performed using original stacks, not Z-projections. Color channels were assembled, and minor adjustments to brightness and contrast were made, in Adobe Photoshop. When images were to be compared, equivalent adjustments were performed on all images in the experiment. The width of the IPL is marked in many of the figures; this was determined by one of the following methods: (1) counterstaining with Hoechst to label all cell nuclei; (2) tdTomato fluorescence from unrecombined cells in *mT/mG* mice, which fills the IPL; (3) immunofluorescence against GAD65, which also fills the IPL; (4) autofluorescence signal intensity differences between soma layers and IPL.

## Analysis of SAC anatomy in embryonic retina

To study SAC anatomy during embryonic stages, *Isl1$^{Cre}$* was crossed to *lox-stop-lox-mGFP* Cre reporter mice (*mT/mG* or *Rosa26$^{GFPf}$*; see Key Reagents) to generate *Isl1$^{mG}$* animals. Timed-pregnant dams were sacrificed at E16 and eyes collected from embryos (*n* = 11 mice from three litters). Tissue was processed as described for postnatal eyes, except fixation time was 60 min. Cross-sections were stained with anti-GFP to reveal the morphology of *Isl1$^{mG}$*-expressing neurons, as well as Sox2 to distinguish *Isl1$^{mG}$*-positive SACs from RGCs. (All cells shown in *Figure 2B–G* were confirmed to be SACs by Sox2 co-labeling.) In combination with these markers, anti-internexin staining was used to assess orientation of primary dendrites. Location and/or presence of the IPL was determined using Hoechst nuclear staining, which revealed cell body-free neuropil regions, and/or by *Isl1$^{mG}$* labeling of neuronal processes, which filled these neuropil regions (*Figure 2—figure supplement 2*). We assessed anatomy of mGFP$^+$ migrating SACs in the ONBL, as well as SACs in the INBL that were concluding their migration. Morphology of ON SACs in the GCL could not be discerned due to *Isl1* expression by RGCs (*Figure 2A,B*), but because displaced amacrine cells pause at the INL-IPL border before crossing to the GCL (*Chow et al., 2015*), the population of cells available to analyze might have included both ON and OFF SACs.

To measure the orientation of primary dendrites at E16 and P1, the angle ROI function in ImageJ was used. This function outputs an angle degree measurement (absolute value) between two line segments. The first line segment of the angle was drawn to follow the trajectory of the internexin$^+$ primary dendrite; the endpoint was at the cell body. The second line segment of the angle was a plumb line to the IPL (i.e. it was drawn to intersect the IPL at ~90°). As such, dendrites oriented exactly toward the IPL were assigned an angle of 0°. At E16 the IPL was occasionally not present yet; in this case the second line segment was a plumb line to the inner limiting membrane. In cases where the internexin$^+$ dendrite curved, we traced the initial trajectory of the dendrite as it emerged from the cell body. Dendrites were classified as projecting (1) towards the IPL; (2) toward the ONBL; or 3) tangentially, according to the angle scheme delineated in *Figure 2—figure supplement 1E*. Image stacks were randomly selected for analysis from a larger library of images; within each selected stack every SAC was traced. *Isl1$^{mG}$* and Sox2 were used to confirm the SAC identity of each measured cell, as well as the trajectory of the internexin$^+$ dendrite.

## Characterization of SAC homotypic arbor network in soma layers

The homotypic nature of SAC soma-layer contacts was investigated by imaging single *Chat$^{mG}$*-labeled OFF SACs in mice also carrying a single copy of the *Megf10$^{lacZ}$* allele (*Figure 2I,J*). Anti-βgal staining was used to reveal the full SAC population, including arbors. *En-face* images were captured

in Z-stacks spanning the INL and IPL; slices corresponding to each layer were separately Z-projected for display in *Figure 2* and *Figure 2—figure supplement 3*. To quantify the frequency of SAC-SAC contacts, we used Z-stacks from P1 tissue to examine the trajectory and termination site of each dendritic tip in three dimensions. The fraction of *Chat^mG*-labeled dendrites terminating on the βgal-positive soma or arbor of a neighboring SAC was quantified. To be counted, the putative contact needed to be confirmed in a single Z-stack slice; where necessary, 3D reconstructions and orthogonal views were used to confirm contact.

We also performed the same analysis on Z-stacks in which one channel had been flipped about the horizontal and vertical axes. This served as a negative control to measure the frequency with which GFP and βgal arbors interact by chance, given their density and geometry in the P1 retina. Sample sizes are given in main text and in *Figure 2—figure supplement 3*.

## Generation and analysis of 'solitary' SACs
### Reduction of SAC density using Ptf1a^flox mice
*Ptf1a^flox* mutant mice (*Krah et al., 2015*) were crossed into the *Six3-Cre* background to generate Ptf1a-cKO mice. *Six3-Cre* is expressed by retinal progenitors starting at E9.5 in a high-central-to-low-peripheral gradient (*Furuta et al., 2000*; *Figure 3A*). In central retina, where Cre is expressed in all progenitors, amacrine cells were completely absent but bipolar cells, RGCs, Müller glia, and photoreceptors remained (*Figure 3B*; *Figure 3—figure supplement 1*; data not shown). In peripheral retina, where Cre recombination was incomplete, amacrine cells derived only from Cre-negative progenitors (*Figure 3C*). Because the number of Cre-expressing progenitors in peripheral retina still vastly exceeded the number that escaped Cre, amacrine cell density in Ptf1a-cKO peripheral retina was markedly reduced compared to littermate controls (*Figure 3A,B*; *Figure 3—figure supplement 1*).

### Quantification of dendrite phenotypes in solitary and touching SACs
To visualize SACs and quantify their arbor targeting frequencies in Ptf1a-cKO mice, we bred *Megf10^lacZ* into the *Ptf1a^flox* background. All Ptf1a-cKO and littermate control mice in these experiments carried one copy of the *Megf10^lacZ* allele. SAC morphology was revealed with anti-βgal. Sox2 was used to confirm the SAC identity of all cells included in the experiment. SACs were scored as 'solitary' or 'touching' based on whether their dendrites contacted neighboring SACs in the same or adjacent sections. If this could not be determined (e.g. because the adjacent section was missing or damaged), the cell was excluded from further analysis. Because SACs were only present in Ptf1a-cKO peripheral retina, analysis of littermate control SACs was also limited to peripheral retina. In Ptf1a-cKO mice, SACs were more frequently found in the INL than the GCL and it is possible that the INL SACs were a mixed population of ONs and OFFs. Therefore, we did not distinguish between SAC subtypes for the analyses.

IPL projections of βgal-stained cells were examined, and cells were assigned to one of three categories: 1) no arbors projecting to the IPL; 2) Arbors enter the IPL but fail to stratify; 3) Arbors enter the IPL and ramify in a laminar pattern. Examples of the first category of solitary SACs are shown in *Figure 3F*, left, and *Figure 3—figure supplement 1*. Examples of the second category are shown in *Figure 3F*, right, and *Figure 3—figure supplement 1*. The third category is exemplified by all touching SACs shown (*Figure 3E*; *Figure 3—figure supplement 1*). Each cell in the dataset was also scored on an independent criterion: whether it projected to the soma layer (e.g. *Figure 3D,F*, white arrows).

For each animal in the experiment, the following was calculated and plotted in *Figure 3G*: (1) Percentage of SACs with projections to the soma layers; (2) percentage of SACs projecting to the IPL (i.e. the cells assigned to categories 2 and 3 above); (3) percentage of SACs with stratified IPL dendrites (i.e. the cells in category 3). Sample sizes: $n$ = 3 wild-type littermates (28, 62, 32 cells analyzed in each animal); $n$ = 4 Ptf1a-cKO animals (11, 35, 13, 12 solitary and 27, 44, 22, 23 touching SACs analyzed in each animal). Statistics: one-way ANOVA with Tukey's post-hoc test.

## Quantification of SAC projection phenotypes in *Chat^mG* mice
Single SACs labeled in *Chat^mG* and *Chat^mG;Megf10^-/-* mice were morphologically assessed in cross-sections. GFP signal was amplified with anti-GFP antibody staining. All GFP⁺ SACs on any given slide

were imaged and analyzed, to avoid cell selection bias, with the exceptions of: 1) cells severed by the cryosectioning process; 2) cells with arbors that could not clearly be distinguished from those of their neighbors; 3) cells in the far retinal periphery, where sections were oblique to retinal layers, obscuring IPL strata. In experiments analyzing *Megf10* mutants, littermates were always used as controls to avoid complications arising from the fact that the precise state of retinal development at the time of birth might vary from litter to litter.

A cell was scored as innervating the IPL if it ramified branched dendrites within the neuropil. Dendrites that entered the neuropil but did not branch or stratify (e.g. *Figure 6D*) were not sufficient. A cell was scored as projecting to the soma layer if arbors emanating from the cell soma or primary dendrite terminated or arborized in the INL (for OFF SACs) or GCL (for ON SACs). The arbor was required to be $\sim \geq 1$ cell radius in length (i.e. small fine arbors were not counted). One other important exception that was not counted: We observed that many SACs at young ages had single unbranched arbors extending $\sim 180°$ away from the IPL (e.g. *Figure 2J,K* – all four cells have such arbors, even the ones that do not project towards neighboring SAC somata). These processes were not counted for two reasons. First, their trajectory was such that they were unlikely to join the soma-layer dendrite network or contact neighboring somata. Second, these 180° arbors were sometimes still present in P5 SACs (*Figure 2—figure supplement 2*) and therefore they did not appear to be subject to the same developmental regulation as tangentially-directed arbors (*Figure 2L*). This observation suggests they are fundamentally different, and likely serve a different (as yet uncharacterized) purpose. No obvious difference in their frequency was observed between wild-type and *Megf10* mutants.

To produce graphs in *Figures 2L*, *6E* and *8F*, the fraction of cells making ectopic projections – either to the soma layer or to inappropriate IPL sublayers – was calculated for each genotype and each time point. To determine whether a GFP$^+$ IPL arbor was located in normal or abnormal IPL strata, *Megf10:*βgal was used as a counterstain. *Chat$^{Cre}$* was rarely expressed in OFF SACs at P0, making it difficult to obtain large sample sizes at this age. For this reason, and because soma-layer projection frequency did not appear to differ much between P0 and P1, the data from each time point was pooled for analysis of *Megf10* litters.

Sample sizes for *Figure 2L*: P0, *n* = 25 OFF, 63 ON; P1, *n* = 51 OFF, 79 ON; P2, *n* = 46 OFF, 55 ON; P3, *n* = 33 OFF, 49 ON; P5, *n* = 15 OFF, 26 ON; P7, *n* = 23 OFF, 34 ON. Data were from four litters of mice, each of which was assessed at no less than two of these time points.

Sample sizes for *Megf10; Chat$^{mG}$* experiments (*Figure 6E*; *Figure 8F*): *Megf10* heterozygous littermate controls: P0/1, *n* = 11 OFF, 25 ON; P2, *n* = 25 OFF, 23 ON; P3, *n* = 17 OFF, 22, ON; P5, *n* = 16 OFF, 16 ON. *Megf10* mutants: P0/1, *n* = 6 OFF, 25 ON; P2, *n* = 14 OFF, 20 ON; P3, *n* = 34 OFF, 41 ON; P5, *n* = 48 OFF, 54 ON. Data were from two litters of mice.

For the adult data reported in *Figure 8F,* a different procedure was used; see 'Quantification of Mosaic Spacing Phenotypes' section below.

## Analysis of Chat-Megf10-cKO conditional mutants

### Characterization of timing of MEGF10 deletion

For initial characterization of when MEGF10 protein is eliminated by the *Chat$^{Cre}$* driver line, the following experiment was performed: *Chat$^{Cre}$; Megf10$^{flox}$* mice were intercrossed with *Chat$^{Cre}$; Megf10$^{lacZ}$* carriers to generate *Chat$^{Cre}$; Megf10$^{flox/lacZ}$* (Chat-Megf10-cKO) experimental animals and littermate controls (*Chat$^{Cre}$; Megf10$^{flox/+}$*). These animals also carried a *Rosa26* mGFP Cre reporter allele. Animals were sacrificed at P1, P3, and P5; retinas were cross-sectioned and immunostained for anti-MEGF10 (*Figure 7—figure supplement 1*). Comparisons were made across animals from the same litter to assess how MEGF10 immunoreactivity changed over time. Two litters were analyzed in this way, each yielding the same conclusion: MEGF10 immunoreactivity was largely eliminated by P5 in Chat-Megf10-cKO mice (*Figure 7—figure supplement 1*). At P3, overall MEGF10 levels were reduced, but most SACs still expressed detectable protein (*Figure 7—figure supplement 1*). The cells that lost MEGF10 immunoreactivity by P3 were not necessarily the same cells that recombined the mGFP reporter at the *Rosa26* locus (*Figure 7F,G*). At P1, only a very small number of cells (<5 per retina) could be identified that lacked MEGF10 immunoreactivity; most of these were ON SACs although a few recombined OFF SACs were identified (*Figure 7G*). We conclude that a small fraction of SACs loses MEGF10 protein prior to P3, while the majority lose

MEGF10 between P3 and P5. Further, ON SACs are somewhat more likely to lose MEGF10 before P3 than OFF SACs.

## Assessment of morphological and IPL projection phenotypes

To ask if loss of MEGF10 prior to P3 affects dendritic targeting, *Chat*$^{mG}$-labeled single SACs were identified in retinal cross-sections from Chat-Megf10-cKO and *Chat*$^{Cre}$; *Megf10*$^{flox/+}$ control mice, as described above. Analysis was performed at P1 and P3; data in *Figure 7H* is from P3 only. All mGFP$^+$ SACs were first scored as to whether they expressed MEGF10 protein (see *Figure 7F,G*). Subsequently, each cell was scored for soma-layer projection as described above for wild-type and *Megf10*$^{-/-}$ animals. This scoring was done blind to the cell's MEGF10 expression status. The fraction of cells classified as either 'soma-projecting' or 'IPL-only' was calculated for MEGF10$^+$ SACs, MEGF10$^-$ SACs, and littermate control SACs (*Figure 7H*). Sample sizes: *n* = 26 OFF, 18 ON cells from controls; 24 OFF, 19 ON MEGF10$^+$ cells from Chat-Megf10-cKO; 9 OFF, 17 ON MEGF10$^-$ cells from Chat-Megf10-cKO.

To assess SAC stratification at maturity, cross-sections from P17 Chat-Megf10-cKO and littermate controls were stained for anti-ChAT. Four mutants and three littermate controls, from two litters, were examined.

## Analysis of Six3-Megf10-cKO conditional mutants

### Characterization of Cre recombination patterns

Breeders carrying the relevant alleles were interbred to generate *Six3-Cre; Megf10*$^{flox/lacZ}$ (*Six3 Megf10*$^{cKO}$) mice and littermate controls (*Six3-Cre; Megf10*$^{+/lacZ}$ or *Cre*$^-$ *Megf10*$^{flox/lacZ}$). As noted above in *Ptf1a* section, Cre is expressed very early (~E9.5) in *Six3-Cre* retina, but expression is incomplete, with some parts of peripheral retina spared from Cre activity (*Furuta et al., 2000*). Therefore, all mice used for these experiments also carried the *Rosa26*$^{GFPf}$ Cre reporter, to reveal retinal regions that either lacked MEGF10 (GFP$^+$ cells) or were spared from MEGF10 deletion (GFP$^-$ cells). Anti-MEGF10 staining confirmed that the GFP Cre reporter is a reliable marker of MEGF10 expression status (*Figure 7—figure supplement 1*).

### Assessment of morphological phenotypes

For quantification of INL projection frequency at P2, *Six3 Megf10*$^{cKO}$ and littermate control whole-mount retinas were stained for βgal, Sox2, and anti-GFP. This staining marked SACs (Sox2 and βgal), revealed their dendritic morphology (βgal), and defined their MEGF10 expression status (GFP). Confocal stacks were acquired through the INL, extending to the IPL (which was clearly discernable due to dense βgal and GFP expression). The INL was defined as the region above this in the image stack, containing Sox2$^+$ neurons. Cells that projected into the INL were clearly discernable due to their multipolar morphology with numerous dendritic protrusions (e.g. *Figure 7B*). Cells that did not project to the INL had a round morphology with only minor lateral branches less than one cell radius in length (*Figure 7C*). Each βgal-labeled SAC was scored as to whether it expressed GFP, and whether it projected lateral arbors into the INL. If the cell had only INL branches directed toward the IPL through the stack Z-plane, it was not counted as INL-projecting. Scoring was done in separate sessions so that the scorer was blind to GFP expression status when determining INL projections. Sample sizes: *n* = 117 SACs from two control mice; *n* = 302 GFP$^+$ SACs and 149 GFP– SACs from 2 Six3-Megf10-cKO mice.

To assess SAC stratification in cross-sections, P2, P4, or P17 Six3-Megf10-cKO and littermate control retinas were sectioned and stained for anti-βgal (P2) or anti-ChAT (P17). The number of animals examined was: P2, four mutants, two controls; P4, two mutants, three controls; P17, two mutants, two controls.

## Quantification of area covered by SAC dendritic arbors

### Neonatal individual SAC arbor territory

P0 *Chat*$^{mG}$ retinas were imaged in whole-mount preparations stained with anti-Sox2 and anti-GFP antibodies to identify single GFP$^+$ SACs. To avoid cell selection biases, all labeled SACs with arbors that were clearly distinguishable from their neighbors were imaged and analyzed, except for far-peripheral cells that may have been damaged during mounting. At least four animals were imaged

for each genotype. Z stacks were acquired through the GCL, IPL, and INL to encompass all arbors of a single cell. Images were imported into ImageJ, z-projected into a single plane, and polygons were drawn connecting the dendritic tips, nearest neighbor to nearest neighbor, until the dendritic field was captured. Area of this polygon was calculated using ImageJ. Sample sizes: OFF SACs, $n$ = 16 wild-type and 16 $Megf10^{-/-}$; ON SACs, $n$ = 31 wild-type and 34 $Megf10^{-/-}$. Statistics: two-tailed $t$-tests.

## Adult individual SAC arbor territory

Individual SACs were labeled by injection of $Chat^{Cre}$ mice with 'Brainbow' Adeno-associated virus (AAV) driving fluorophore expression in a Cre-dependent manner (*Cai et al., 2013*). The two Brainbow AAV9 viruses, encoding farnesylated fluorescent proteins that are targeted to the plasma membrane (University of Pennsylvania Vector Core), were mixed to $1.5 \times 10^{12}$ genome copies per mL. Adult mice (P40-50) were anesthetized with ketamine-xylazine by intraperitoneal injection. Proprara-caine hydrochloride (0.5%) ophthalmic solution (Akorn, Lake Forest, IL) was applied to the eye to provide local anesthesia. A 30 1/2G needle was used to make a small opening near the ora serrata, and 1 μl of virus was injected with a 33G blunt-ended Hamilton syringe intravitreally. Tissue was collected 3 weeks after the virus injection.

Retinas were stained in whole-mount with anti-GFP, anti-mCherry, and anti-mKate antibodies to reveal SACs. OFF SACs were not labeled in large numbers, so analysis was restricted to more abundantly labeled ON SACs. Imaging, image processing, and quantification were as for P0, except that only SACs in central and mid-peripheral retina were used to avoid confounding effects of eccentricity on arbor size. Sample sizes: $n$ = 10 wild-type and 16 $Megf10$ mutant SACs.

## ChAT arbor plexus retinal coverage

*En-face* images of adult (P46) ON or OFF SAC plexus were obtained from confocal Z-stacks (0.4 μm Z resolution) by performing maximum-intensity Z-projections of 2–4 optical slices encompassing the relevant layer. Using ImageJ, these images were then thresholded, converted to binary, and the percentage of the field of view covered by ChAT-positive arbors was calculated. All image stacks were obtained from central or mid-peripheral retina. Sample size: 9 fields of view from 2 $Megf10$ mutants and two heterozygous littermate controls were used to calculate average coverage for each genotype. Percent change is reported in the Results; total retinal coverage was as follows: Control ON, 65.9 ± 1.3%; control OFF, 70.6 ± 3.6%; mutant ON 56.9 ± 3.8%; mutant OFF, 60.0 ± 3.7% (mean ±S. D).

## Hb9-GFP stratification

P1-P2 retinas carrying $Megf10^{lacZ}$ and Hb9-GFP were co-stained for βgal and GFP. RGCs with dendrites that co-fasciculated with βgal-positive IPL strata were counted. Cells that projected to βgal-positive regions, but also filled non-SAC-projecting IPL regions, were not counted as co-fasciculated. To judge co-fasciculation, we used two criteria: 1) inspection of dendrite anatomy across the confocal stack; 2) fluorescence profiles of GFP and βgal channels across IPL (see next section below). Examples of cells falling into each category are provided in *Figure 1* and *Figure 1—figure supplement 2*. See Results for sample sizes.

## Quantitative assessment of IPL stratification level

Images of retinal cross sections were processed in ImageJ. A vertical ROI (12.5 μm wide) was drawn to perpendicularly bisect the IPL strata, from the edge of the INL to the edge of the GCL. IPL stratification levels were reported as percentage of IPL width. Intensity was calculated for each pixel along the length of the ROI as an average across its width. Background (minimum pixel value) was subtracted; then, all pixel intensity values were normalized to the maximum value of that ROI. Location of fluorescent peaks was calculated as the pixel with maximum intensity; if multiple pixels had the same intensity the peak was defined as the center of the plateau. The procedure was typically performed on single confocal optical sections, but for some P1-2 cells, which have much smaller arbors, it was necessary to use a maximum-intensity projection of a small number of slices in order to fully capture dendrite morphology.

For BC5-BC7 arbor distance measurements (*Figure 9F*), distances as percentage of total IPL width were compared by one-way ANOVA/Tukey's post-hoc test. $n$ = 14 measurements from two control mice; $n$ = 7 normal IPLs, 11 SAC clumps, 11 SAC gaps from 3 *Megf10*$^{-/-}$ mice.

## Generation of Megf10-ΔICD constructs

The MEGF10-ΔICD-GFP construct was reported previously (*Kay et al., 2012*), which was originally made from pUbC-MEGF10-GFP (Addgene #40207). It encodes a version of MEGF10 in which the cytoplasmic domain is truncated after the 9th amino acid and replaced by GFP. Inclusion of those nine amino acids was necessary to achieve plasma membrane localization. For this study, it was subcloned into the pEGFPN3 plasmid, containing the CMV promoter, to make pCMV-MEGF10-Δ ICD-GFP.

To make the MEGF10-ΔICD-Flag construct, Megf10 (truncated after the 9th intracellular domain amino acid as above) was PCR amplified from pUbC-MEGF10-GFP vector using M10flag_Fwd forward primer and Cyto9_flag_Rev1 reverse primer. Resulting PCR products were digested with NotI and AscI restriction enzymes and ligation cloned into pEGFPN3 vector linearized with corresponding restriction enzymes.

## Assay for interaction of MEGF10-ΔICD constructs

### Co-immunoprecipitation

HEK293T cells were grown to 80% confluency. Cells were then transfected using a linear polyethylenimine (PEI) transfection reagent: DNA, PEI, and Opti-MEM were mixed in a 1:3:30 ratio and incubated for 10 min at room temperature then applied to confluent cells. Cells were harvested 48 hr post-transfection. Cells were lysed with NP-40 lysis buffer (1% NP-40, 150 mM NaCl, 50 mM Tris-Cl, and 1X proteinase inhibitor) by pipetting. Lysate was centrifuged at 14,000 x g at 4°C for 15 min. to remove insoluble material. The soluble protein fraction was quantified with Bio-Rad DC assay. For immunoprecipitation, 500 μl (1 μg/ μl) protein in NP-40 buffer lysis buffer was incubated overnight at 4°C with antibody (1 μl of chicken anti-GFP or 2 μl of mouse anti-Flag). Protein G Dynabeads (10 μl) were added to mixture for 1 hr at 4°C while rotating. Beads were sequestered by magnet and flow-through was removed. Beads were washed with 500 μl lysis buffer (3x) on ice then eluted with 30 μl 2X Laemmli containing 5% β-mercaptoethanol.

### Western blot

Samples were prepared in 2X Laemmli sample buffer, heated at 95°C for 10 min, and loaded onto SDS-acrylamide gel (running gel: 8% acrylamide/bis Tris-HCl with 0.1% SDS pH 8.8; stacking gel: 5% acrylamide pH 6.8; cross linked with TEMED and APS). Precision Plus Protein Dual Color Standards (BioRad) were used as a molecular weight marker. The gel was run on a BioRad mini gel running apparatus with SDS-PAGE running buffer (25 mM Tris, 192 mM glycine, 0.1% SDS). Electrophoresis was carried out at 50 V through the stacking gel then adjusted to 120 V until the dye front reached the lower end of the gel. BioRad Immobilon-FL PVDF membrane and Whatman filter paper were used with the BioRad mini cassette for transfer. Samples were transferred in 25 mM Tris, 192 mM glycine, 20% methanol at 100 V for 90 min. Membranes were blocked with PBS/Odyssey blocking buffer and stained with chicken anti-GFP 1:20000, mouse anti FLAG 1:20,000 overnight at 4°C with shaking. After washing with PBST for four times, membranes were stained with 1:20,000 secondary antibodies for 1 hr at room temperature. The membranes were washed with PBST four times and then rinsed with PBS and water. Finally, the membranes were imaged with LI-COR Odyssey using the Image Studio software.

## Quantification of mosaic spacing phenotypes and their effects on SAC IPL projections

### Regularity index

Regularity of SAC cell body distribution in Six3-Megf10-cKO, Chat-Megf10-cKO, and littermate control mice was calculated as previously described (*Kay et al., 2012*). The Voronoi domain regularity index (VDRI) was used as a measure of regularity. It is calculated by first assigning a Voronoi domain to each cell in an array (*Figure 9—figure supplement 1*), and then calculating the mean and standard deviation of the domain areas. The VDRI is defined as the mean area divided by the standard

deviation. Arrays that are less regularly distributed will have a lower VDRI because their domain sizes are more variable (and hence have a higher standard deviation).

P17 whole-mount retinas were stained with an antibody to ChAT and imaged *en face*. One eye was processed from each animal used in the experiment. For each eye, three confocal image stacks were obtained using a 20x objective (636.5 $\mu m^2$ field of view). Images of INL SACs were analyzed using Fiji software. The location of each SAC in the field of view was marked; this information was used to count the number of SACs (*Figure 9B*) as well as define Voronoi domains belonging to each cell, using Fiji functions. The area of each Voronoi domain (excluding edges) was calculated in Fiji.

For statistical analysis of regularity effects across genotypes, we first calculated the per-animal average cell density and VDRI from the three acquired images. Differences between genotypes were then evaluated using one-way ANOVA and Fisher's PLSD. Previously published *Megf10* null and simulation data was also included for comparison (*Kay et al., 2012*). The simulations define the VDRI that would be expected for a randomly-arranged array of cells matched in size and density to real SACs. Data collection and analysis was virtually the same as in the previous study, allowing us to include these data in our statistical comparisons.

## Effects of soma position upon IPL errors: single-cell analysis

To ask if soma position correlates with IPL errors, we first defined the ectopic projection status of each OFF SAC in a set of Z-stacks acquired from ChAT-immunostained retinal whole-mounts. Sample sizes: $n$ = 515 cells from two control (*Megf10$^{+/-}$*) mice; $n$ = 584 cells from 2 *Megf10* mutant mice. The Z-stacks encompassed, at different levels of the stack, SAC somata in the INL and their ramified arbors in the IPL. In *Megf10* mutants, the OFF ectopic IPL arbor network and the typical OFF DS circuit sublayer were identified at different stack levels (*Figure 8C,E*). ChAT$^+$ arbors arising from individual OFF SAC somata were traced through the stack to identify those that joined into the ectopic network. The fraction of SACs that did so was then calculated and plotted in *Figure 8F*. For *Figure 8G*, we further examined these stacks to look for SACs that made ectopic projections at the INL level.

Next, we defined the severity of mosaic spacing perturbations in the local neighborhood of each SAC. Because SAC position is random in *Megf10* mutants, SACs might be more crowded or more isolated from their neighbors than in controls; or, by chance, they might be located at a fairly normal distance from their neighbors. The size of a cell's Voronoi domain is influenced by the distance of all nearest neighbors (*Figure 9—figure supplement 1*), and therefore serves as a convenient measure of local cell density. For simplicity we refer to Voronoi domains as 'territory size' in *Figure 9—figure supplement 1*. The effect of local cell density upon IPL projection errors was determined by plotting the ectopic error rate for each 100 $\mu m^2$ territory size bin (*Figure 9—figure supplement 1*). Sample size per bin, in order from smallest (<200 $\mu m^2$) to largest (>1100 $\mu m^2$): $n$ = 32, 65, 89, 102, 91, 80, 30, 39, 24, 34.

## Multielectrode array recordings
### Isolation of retina, recording, and spike sorting

Two wild-type and two *Megf10$^{-/-}$* animals were used for multielectrode array (MEA) recordings. Immediately following euthanasia, retinas were isolated under infrared (IR,>900 nm) illumination with the assistance of IR-to-visual converters. This preserved the photosensitivity of the retina during the dissection. Dissections were performed in sodium bicarbonate-buffered Ames' solution (Sigma, St. Louis, MO) equilibrated with 5% $CO_2$ +95% $O_2$ to pH 7.4 and maintained at 32–34° C. Hemisection of the eye was performed along the ora serrata by first making a small incision, following which the vitreous was removed and the retina was isolated from the pigment epithelium and eye cup. A piece of dorsal retina (1–2 $mm^2$) was dissected and placed RGC-side down on the planar MEA.

The MEA consisted of 519 electrodes with 30 $\mu m$ inter-electrode spacing, covering a hexagonal region with 450 $\mu m$ on a side (*Field et al., 2010*). The voltage on each electrode was digitized at 20 kHz and stored for post-hoc analysis. Details of recording methods and spike sorting have been described previously (*Field et al., 2007*). Spikes were identified using a threshold of four times the voltage standard deviation on each electrode. Principal component analysis applied to the ensemble of spike waveforms measured on each electrode provided a subspace for clustering spikes according to their shape. A Gaussian mixture model was used to cluster the spikes originating from individual

RGCs. The clusters were manually inspected for each identified ooDSGC to ensure the spike wave-forms were well isolated from other simultaneously recorded RGCs and all spikes were captured within each cluster. When a single cluster of spikes was captured by more than one Gaussian or when a single Gaussian included spikes from more than one cluster, the clustering was manually adjusted to generate a new set of initial conditions for re-fitting the mixture of Gaussians. Spike clusters with >10% estimated contamination based on refractory period violations, or spike rates < 1 Hz, were excluded from further analysis.

## Visual stimulation and RGC responses

Visual stimuli were focused on the photoreceptor outer segment, from an OLED display (Emagin, Inc.) with 60.35 Hz refresh rate. The mean intensity of the stimulus was 7000 photoisomerizations per rod per s, or 5000 photoisomerizations per cone per s for a cone containing all M-opsin. These estimates do not account for the effect of pigment self-screening. To measure the direction tuning of ooDSGCs as a function of contrast, a positive contrast bar (1200 μm wide) was presented on a gray background (*Figure 12B*). On each presentation, the bar moved in one of 12 equally spaced directions at 400 μm/s and was presented at one of the following (Weber) contrasts: 5%, 10%, 20%, 40%, 80%, 150% and 300%. Responses to a total of 8 trials were collected for every condition; stimulus conditions were presented pseudo randomly. Spike times were binned at 1 ms resolution for all subsequent analyses.

To distinguish DSGCs from other RGCs recorded on the MEA, square-wave drifting gratings were used. These gratings drifted in one of 12 different and equally spaced directions and at two different speeds (225 μm/s and 900 μm/s; spatial period 400 μm/cycle). DSGCs were identified based on their direction selectivity index (DSI) defined as:

$DSI = \frac{|\sum \vec{v_i}|}{\sum n_i}$ calculated from responses to drifting gratings and moving bars. Here, $n_i$ is the number of spikes elicited to stimulus movement along the direction $i$ defined by the vector $\vec{v_i}$.

The distribution of DSIs across all recorded RGCs was bimodal, with DSGCs forming the high mode (*Figure 12A*). Based on these distributions, a DSI of 0.25 reliably identified DSGCs in wild-type and *Megf10*$^{-/-}$ retinas. ooDSGCs were isolated from ON DSGCs by their distinct ON and OFF responses to a bar entering and exiting the receptive field (*Figure 12B*). The total ooDSGC sample size obtained by this procedure was $n = 80$ from the two wild-type and $n = 74$ from the two *Megf10*$^{-/-}$ retinas. The paired Kolmogorov-Smirnov (KS) test was used to compare cumulative probability distributions from these two populations.

## Analysis of ooDSGC response

### Measurement of direction tuning width

First, the direction tuning curve for each ooDSGC was obtained by calculating the number of spikes elicited across all trials for each direction of bar movement. Due to the circular nature of the data, the direction tuning curve was treated as circular normal distribution, also called von Mises distribution (*Oesch et al., 2005*), and the tuning width was measured as the circular standard deviation ($\sigma_{circ}$), defined by

$\sigma_{circ} = \sqrt{-2\ln(R)}$ where $R$ is the second moment of the von Mises distribution:

$$f(\theta, \mu) = \frac{1}{2\pi I_0(\kappa)} e^{\kappa \cos(\theta - \mu)}$$

This yielded a nonparametric estimate of the tuning curve width.

### Measurement of direction tuning strength

To measure the strength of tuning, the difference between spike counts to motion in the preferred and null directions was normalized by the sum of these responses. The tuning curves were sampled at 30 degree intervals. To estimate the response in the preferred (null) direction, which could fall between sampled directions, a cosine-weighted average of the two strongest (weakest) responses was calculated. This yielded the following equation for measuring tuning strength:

$$\text{Tuning strength} = \frac{\sum\limits_{i=1}^{2} r_i \cos(|\theta_{PD} - \theta_i|) - \sum\limits_{j=1}^{2} r_j \cos(|\theta_{ND} - \theta_j|)}{\sum\limits_{i=1}^{2} r_i \cos(|\theta_{PD} - \theta_i|) + \sum\limits_{j=1}^{2} r_j \cos(|\theta_{ND} - \theta_j|)}$$

where the summation $\sum_{i=1}^{2}$ is performed over the responses $r_i$ weighted by the cosine terms for the two nearest neighbor movement directions $\theta_i$ around the preferred direction $\theta_{PD}$ and the null direction $\theta_{ND}$. This resulting index for tuning strength varied between zero and unity.

## Measurement of direction tuning similarity between ON and OFF responses

To separately analyze the ON and OFF responses of ooDSGCs, we first defined temporal windows for each ooDSGC that distinctly separate the ON and OFF responses. This was done by passing high-contrast moving bars (150% and 300% contrast) through the receptive field. In the resulting spike rasters, ON and OFF response phases were clearly discernible (*Figure 12B,F*). The boundary for separating the ON and OFF responses was set halfway between the peak ON and OFF spike rate locations *Figure 12F,G*). Once the temporal boundary was defined, the preferred direction was calculated independently for the ON and OFF responses for each ooDSGC. The same ON-OFF temporal boundaries were used for all contrasts shown in *Figure 12—figure supplement 1*. The difference between the preferred directions, $\Delta\phi$, quantified the angular difference between the ON and OFF preferred directions (*Figure 12F*).

## Analysis of ooDSGC subtypes

ooDSGC subtype classification was performed using the K-means clustering algorithm. This was done by first assigning a set of four initial seed values corresponding to the four cardinal directions of ooDSGCs (*Oyster and Barlow, 1967*). Next, the angular difference between the seed values (for first iteration) or the cluster means (for later iterations), and the preferred directions of each ooDSGCs was calculated. The cluster for which the angular difference was minimum was the cluster to which an ooDSGC was assigned. This yielded the four subpopulations of ooDSGCs described in *Figure 12—figure supplement 1*.

## Quantification and statistical analysis

Statistical analysis was performed using GraphPad Prism software (anatomy/development studies) or using custom JAVA-based software and MATLAB software (physiology studies). This software is available, together with the primary data it was written to analyze, at a public repository (*Roy and Field, 2017*; https://github.com/Field-Lab/megf10-dstuning; copy archived at https://github.com/elifesciences-publications/megf10-dstuning). Statistical tests used for each experiment are given in the appropriate Materials and methods section above, and/or in the figure legends. Sample sizes for each experiment are given in the appropriate Methods section above or else in the Results. p-Values ($\alpha$ = 0.05) are given in figure legends, or in the Results if no figure is shown. Error bars are defined in figure legends. Exact *p*-values are reported unless the value was less than $1.0 \times 10^{-7}$.

## Acknowledgements

For financial support we thank the National Eye Institute (EY024694 to JNK; EY026344 to TAR; EY024567 to GDF; EY5722 to Duke University); Pew Charitable Trusts, E Matilda Ziegler Foundation, McKnight Endowment Fund for Neuroscience, Alfred P Sloan foundation (JNK); the Whitehall Foundation (GDF); and Research to Prevent Blindness Unrestricted Grant (Duke University). We thank Sidney Kuo and Greg Schwartz for suggesting use of the *Gjd2-GFP* mouse line; members of the Kay lab and Cagla Eroglu for their comments on the manuscript; Megan Stogsdill and Ari Pereira for mouse colony support; and X Duan and J Sanes (Harvard) for *Kcng4^Cre^* mice.

# Additional information

## Funding

| Funder | Grant reference number | Author |
| --- | --- | --- |
| National Eye Institute | EY024694 | Jeremy N Kay |
| Pew Charitable Trusts | | Jeremy N Kay |
| E. Matilda Ziegler Foundation for the Blind | | Jeremy N Kay |
| McKnight Endowment Fund for Neuroscience | | Jeremy N Kay |
| Alfred P. Sloan Foundation | | Jeremy N Kay |
| Whitehall Foundation | | Greg D Field |
| Research to Prevent Blindness | Unrestricted grant to Duke University | Thomas A Ray<br>Suva Roy<br>Christopher Kozlowski<br>Jingjing Wang<br>Jon Cafaro<br>Greg D Field<br>Jeremy N Kay |
| National Eye Institute | EY026344 | Thomas A Ray |
| National Eye Institute | EY024567 | Greg D Field |
| National Eye Institute | EY5722 to Duke University | Thomas A Ray<br>Suva Roy<br>Christopher Kozlowski<br>Jingjing Wang<br>Jon Cafaro<br>Samuel W Hulbert<br>Greg D Field<br>Jeremy N Kay |

The funders had no role in study design, data collection and interpretation, or the decision to submit the work for publication.

## Author contributions

Thomas A Ray, Conceptualization, Data curation, Formal analysis, Validation, Investigation, Visualization, Methodology, Writing—original draft, Writing—review and editing; Suva Roy, Conceptualization, Data curation, Software, Formal analysis, Validation, Investigation, Visualization, Methodology, Writing—review and editing; Christopher Kozlowski, Conceptualization, Formal analysis, Validation, Investigation, Visualization, Methodology, Writing—review and editing; Jingjing Wang, Conceptualization, Resources, Formal analysis, Validation, Investigation, Visualization, Methodology, Writing—review and editing; Jon Cafaro, Conceptualization, Data curation, Software, Formal analysis, Investigation, Methodology, Writing—review and editing; Samuel W Hulbert, Conceptualization, Validation, Investigation, Methodology; Christopher V Wright, Resources, Writing—review and editing; Greg D Field, Conceptualization, Data curation, Software, Formal analysis, Supervision, Funding acquisition, Validation, Visualization, Methodology, Project administration, Writing—review and editing; Jeremy N Kay, Conceptualization, Data curation, Formal analysis, Supervision, Funding acquisition, Validation, Investigation, Visualization, Methodology, Writing—original draft, Project administration, Writing—review and editing

## Author ORCIDs

Samuel W Hulbert (iD) http://orcid.org/0000-0003-0369-0150

Christopher V Wright (iD) https://orcid.org/0000-0002-9260-4009

Jeremy N Kay (iD) http://orcid.org/0000-0001-6145-1604

## Ethics

Animal experimentation: All animal experimental procedures were reviewed and approved by the Institutional Animal Care and Use Committee of Duke University (protocol A005-16-01).

## Decision letter and Author response

Decision letter https://doi.org/10.7554/eLife.34241.033
Author response https://doi.org/10.7554/eLife.34241.034

## Additional files

### Supplementary files

• Transparent reporting form
DOI: https://doi.org/10.7554/eLife.34241.031

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
