## [Decision Letter]

Thank you for submitting your manuscript "Formation of retinal direction-selective circuitry initiated by starburst amacrine cell homotypic contact” to *eLife*. Three experts reviewed your manuscript, and their assessments, together with my own, form the basis of this letter. As you will see, all of the reviewers were impressed with the importance and novelty of your work.

I am including the three reviews at the end of this letter, as there are a variety of specific and useful suggestions in them. In the written reviews and in the subsequent discussion among the reviewers, the general points that emerged are that (1) additional data and/or data quantification would improve the manuscript at various points, (2) some of the interpretations seem to be too strong and should be more nuanced, (3) there is a massive amount of data and analysis for the reader to digest and anything that you can do to help the reader in this regard would be welcome and will increase the impact of the study, and (4) we feel that the manuscript would benefit from a clear focus on the importance of homotypic neurite interactions mediated by megf10 in (the rate of) sublayer formation, rather than stretching to highlight more speculative concepts.

Reviewer #1:

The authors use the development of starburst amacrine cells in the mouse retina as a model for understanding how the laminar organization of the retina arises during development. This is an excellent model system since this cell type is thought to be one of the earliest organizers of the inner-plexiform layer (IPL), a structure within the retina in which specific cell types arborize to form synaptic connections with appropriate partners.

The approaches and analyses used in this paper are quite powerful. The authors use a wide array of transgenic mice to establish several novel findings regarding the development of SACs. For example, they take advantage of a ChAT-Cre mouse, which has weak expression early in development, or Islet1-Cre mice which label SACs, crossed with a membrane-GFP reporter to obtain images of individual SACs during many stages of development, including migration, during which they have both trialing edge and leading edge processes (around E16), prior to innervating the IPL when they make transient contacts with nearby SACs (around P0), and finally then innervation (P2). Not only are these images stunning, but also, they are quite informative, leading the authors to hypothesize that those transient contacts with neighboring SACs are critical for normal lamination. To further motivate this hypothesis, the authors show that the protein internexin – which establishes polarity -- is oriented toward other SACs while migrating and then turn toward the IPL when they innervate the IPL.

To test this hypothesis the authors again use several approaches. First, they prevent expression of a transcription factor involved of differentiation of amacrine cells in central retina and sparsely in peripheral retina to establish areas of retina where some SACs are isolated, meaning they are so sparse that they cannot make homotypic contacts with other SACs. These SACs maintain an immature phenotype and do not innervate the IPL while maintaining processes in the cellular layers, something never seen in normal mice. They then do a variety of manipulations of the membrane protein Megf10-in some cases you have a SAC with normal levels of Megf10 surrounded by mutant SACs lacking Mef10 or vice versa – a mutant SAC surrounded by normal SACs. In both cases, if a SAC is not making signaling with its neighbors via Megf10, then it has the same phenotype as a "isolated SAC".

(Note authors also do nice control experiments to show that the lamination phenotypes cannot be explained by the disruption of SAC mosaic that Kay previously described in Megf10 mutant mice).

The authors then go on to determine the impact of this early disruption of the lamination on the adult. Surprisingly, there appears to be some recovery, in that many of the mutant or isolated SACs do form lamination in the correct place but in addition, they have ectopic branches sometimes that venture into the INL. In one of the most amazing aspects of this study, the authors show that the pre- and postsynaptic targets of the SACs actually change their lamination – essentially keeping their relative associations with the SAC processes, even when those processes are in the wrong location.

Finally, the authors show the impact of this rearrangements on function and find that there is a broadening of direction selective tuning. Interestingly, they find that one of the phenotypes is that ON vs. OFF direction preference are more weakly aligned in Megf10 mutant mice than in WT.

This study is extremely well done – as noted already multiple approaches are used for all experiments. The authors do an excellent job quantifying their results and they are quite conservative in their interpretations. Every "i" is dotted and every "t" is crossed. This is surely going to be a definitive study not only on starburst amacrine cell development but also the role that this cell plays in organizing the entire direction selective circuit. The authors are to be commended.

1) For all figures based on bar plots and SEM error bars (example, Figure 8B), the authors need to present the variance some way. Either the authors need to show individual data points, or they need to use standard deviation for error bars. Example Figure8B

2) Subsection “Timing of DS circuit IPL sublayer formation”: Authors need to clarify how they determined "percent stratified". For example, they state only "30% of ooDSGCs co-fasciculated with SAC arbors". Is this percent of dendrites or percent of cells? What qualified as "co-fasciculated"?

3) Figure 2J-L – purple color should labelled "β-gal" rather than "SAC" for consistency with other figures (e.g. Figure 5).

Reviewer #2

Circuits in the retina are organized such that pre and postsynaptic cells elaborate and confine their processes and synaptic connections within specific sublayers of the neuropil. The sequence of early developmental steps giving rise to these lamination patterns and the molecular cues that are involved are not well understood. Ray et al., carried out an extensive set of experiments capitalizing on genetic tools, imaging and electrophysiology to fill this knowledge gap. They focused on a well characterized retinal circuit, involving inhibitory starburst (SAC) amacrine cells and their postsynaptic partners, the direction-selective (DS) ganglion cells. The authors propose that homotypic interactions direct the processes of SACs at the end stages of their migration to ramify in the IPL. They show convincingly that Megf10-mediated interactions are needed for timely SAC lamination, and if disrupted, leads to broader tuning of direction-selective responses in the DS ganglion cells. Overall, the data is of high quality, quantification and statistical analyses are provided and many findings are supported by the data presented. However, in some places, the conclusions are over-stated; such claims need to be substantiated by further analysis.

1) A major conclusion is that SAC processes make homotypic contact with neighboring SAC cell bodies – this contact is what drives elaboration of SAC processes in the IPL.

The evidence for homotypic contact is not strong. It is difficult to gain a good sense of what is happening based on the images, especially from the frozen sections. Figure 1J (an en face view) provides a bit more support but further analysis is necessary. At the minimum, using en face image stacks of the labeled SACs, the authors should perform a control by flipping one confocal channel and then re-measuring overlap between the channels, and demonstrate that the overlap is above chance. To confirm that there is contact, electron microscopy (EM) is necessary. This is because the diffraction limit of light does not enable one to conclude from two labeled structures alone that there is 'contact'. If the authors perform the overlap measurements with the control, then I think it is fine to conclude that there is an association that is not random, and these are presumed to be contacts.

2) '[…]deletion of Megf10 causes a profound impairment of IPL-directed SAC dendrite growth, preventing timely sublayer formation'.

It is clear that the processes of SACs, especially the OFF SCAs, in the Megf10 KO have not reached or contribute to a singular plexus in the IPL by P3 (although they do). But at P3 in the mutant, the majority of the processes appear directed towards the IPL (Figure 5B). It would be good to clarify whether the authors mean that Megf10 is important for the rate at which the SAC process lamination occurs, or they also mean, the direction of process outgrowth.

3) SAC-BC repulsion regulates bipolar cell axon lamination.

This isn't so clear to me because it is difficult to come to this conclusion based on the images. The small displacements in BC5+7 may simply be because where SACs occupy space, BC5+7 cannot elaborate their axons because the SAC process network is very dense. Where there is a gap in the SAC plexus, these BC axon terminals do come closer together, and this could be because space is now available rather than repulsion is absent.

Results section:

1) Throughout the text, the authors make statements about the behavior of the SACs, which are not strongly supported by the images. Here are some examples:

i) Subsection “Homotypic contact is required for SAC IPL innervation and dendrite lamination”: "Misprojecting DACs are still in contact with numerous other amacrine[…]" and legend for Figure 3—figure supplement 1G: "SACs that made errors had extensive interactions with GAD65+ amacrine cells". It not possible to infer 'contact' or 'interactions' from these data. Please explain.

ii)Subsection “SAC IPL errors induce laminar targeting errors by their DS circuit partners”: 'No changes were seen in Syt2-labeled BC6, suggesting a specific effect on the bipolar cells that make extensive contact with SACs.' The BC6 axons are much further away from the SAC ON plexus compared to the BC7 axon terminals. Thus, I am not sure one can readily conclude that 'DS-circuit bipolar cells respond to SAC attractive cues'.

2) The authors have really put in an enormous effort to obtain a comprehensive study. There is a lot of data and the supplements that complement the Figures are important. But I wonder if the authors would consider streamlining some parts because it is really a bit overwhelming for the reader to try and assimilate all the information relevant to each major conclusion by going back and forth from the main figures to the supplements. For example, the various effects on bipolar cells in the Megf10-/- retina; it would be helpful to at least provide the summary in a single figure for the bipolar cell types analyzed (BC2,3a,5,6,7).

3) The mistargeting of BC2 in Megf10-/- to where there are ectopically located SAC processes is intriguing and clear that there is a bipolar axon terminal lamination defect associated with abnormally placed SAC processes (but for a BC type that is not part of the SAC-DS circuit?). However, to establish a causal link to SACs, it would help to show that lamination of other retinal neurons that are associated with BC2 bipolar cells (if that is known) are unperturbed in their lamination (i.e. are the BC2's following their targets?).

4) Figure 2—figure supplement 2: There should be quantitation for the E16 observations to substantiate SACs projecting to each other. The neuroblastic layer is dense with nuclei – where the processes appear to 'project', they seem to be just coursing around the nuclei.

5) Are SACs not expected to be labeled in the Gad1-GFP line?

6) It would be helpful to indicate in all the figures whether we are looking at a single image plane or a z-projection of several optical planes. This will help the reader gain a sense of just how 'close' the labeled structures are. What was the z-dimension step size for the confocal images on the wholemounts (not sure I found this in the Materials and methods section).

7) Where possible, labeling the borders of the IPL would be helpful- presumably these are defined always by labeling of cell nuclei?

8) OFF SACs show more dramatic ectopic (somatic) processes than ON SCs in the Megf10 mutant. But this defect does not seem to have a physiological correlate – or does it?

9) Gad1-GFP line should be listed in the transgenics used.

10) Has the expression of cre in the Chat-cre line been documented somewhere? This is important with respect to separating the early and late deletion of Megf10.

11) Figure 8—figure supplement 1: Please provide a more detailed explanation of how the 2D cross-correlation analysis was performed.

12) Although it does appear that Meg10 plays two roles- one in somal mosaic positioning and the other in lamination of the processes of SACs, it isn't clear that timing alone separates these roles. If mosaic arrangements are due to repulsive SAC-SAC interactions, then fasciculation of the SAC processes must be an adhesion-based, perhaps even 'attractive' mechanism. Is it known if Megfs have other downstream pathways that are not 'repulsive'?

13) Could the authors elaborate a bit more in the Discussion section on how the lamination of the OFF and ON SAC plexuses become better confined to a single lamina (with exceptions) by adulthood?

Reviewer #3

This study by the Kay lab investigates the role of Megf10 in the development of starburst amacrine cells (SACs). In a previous study by Kay and colleagues (Kay et al., 2012), the mosaic spacing of SACs is impaired in Megf10 knockout mice. In this manuscript, Ray and colleagues examine the early developmental stages of SACs starting at E16. They found that Megf10 is required for the timely stratification of SAC neurites at the inner plexiform layer (IPL). In the absence of Megf10, SACs fail to eliminate ectopic neurites and unable to stratify at the correct IPL depth in the first few postnatal days (P0-P2). This developmental deficit leads to mistargeting and gaps of SAC neurites in the IPL in the mature retina. Furthermore, using elegant genetic tools, the authors demonstrate that Megf10 function requires homotypic interaction of megf10 expressed in neighboring SACs. Overall, this study advances the current understanding of sublayer formation in the retina. It also provides a missing link between the altered developmental process of SACs and the deficits in the mature retina of Megf10 mutants. My major concerns are about some of the interpretations of the data and analysis.

1) One developmental feature that is prominently mentioned in this study is a transient homotypic network of SAC dendrites in the inner nuclear layer (INL) between E16 and P1. The authors describe the neurites outside the IPL to be selectively targeted to the neighboring SAC somas and neurites in the INL. However, an alternative scenario is that SACs during this period have random growth of neurites to both INL and IPL. After P1, the INL neurites are selectively pruned, and the IPL neurites further grow and stratify in the correct IPL sublayer. I did not find sufficient evidence supporting the presence and functional significance of a homotypic network of SAC arbors at the INL. In particular,

- Subsection “Early SAC projections target neighboring SAC somata”: The authors describe SAC neurites in the INL as "soma-directed" neurites. It is unclear to me whether these neurites are "directed" at somas, or they are just randomly distributed in 3D. There is no quantification that shows "selective" somatic targeting over random growth. In fact, in both Figure 2E (E16) and Figure 2H (P0), SACs send more neurites to IPL than to INL.

- Subsection “Early SAC projections target neighboring SAC somata”: The authors claim that "P1 INL arbors selectively contacted somata or arbors of SAC neighbors". I have difficulty recognizing this pattern as "selective" since the INL at this stage is densely covered by SAC somas and ectopic neurites. The INL neurites may always end next to neighboring somas or neurites just because of the high density.

- Subsection “Early SAC projections target neighboring SAC somata”: The authors describe that On SACs send fine soma-directed branches from their IPL arbors (Figure 2—Figure supplement 2). Again, how do the authors distinguish "soma-directed" projection from random contacts due to imperfect stratification?

2) Subsection “Timing of DS circuit IPL sublayer formation”: The authors mention that SACs are stratified within the expected IPL sublayer, but ooDSGC dendrites are unstratified. This needs to be supported by quantifications. In Figure 1—figure supplement 1, panel C, both SACs and the ooDSGC appear to be unstratified in a similar way.

3) Subsection “Homotypic contact is required for SAC IPL innervation and dendrite lamination”: "the projections were typically more elaborate than those observed in wild-type retina". Please provide a quantitative analysis to support this claim.

4) Subsection “Requirement for MEGF10 in SAC IPL innervation and sublayer formation”: "this phenotype is not due to aberrant SAC migration.…" However, from Figure 5A and Figure 5B, Off SAC somas in MEGF10 mutants are farther away from IPL than those in the wild type. And there are significantly less On SACs in the mutant. This is consistent with delayed migration of SACs, which might also account for the delayed stratification phenotype in the mutants.

5) Subsection “Requirement for MEGF10 in SAC IPL innervation and sublayer formation”: "Rather, the absence of SAC sublayers was due to innervation of the soma layers instead of the IPL". I do not see a clear causal relationship here.

6) Related to point 1 above, subsection “Requirement for MEGF10 in SAC IPL innervation and sublayer formation”: To support the statement that P1 SACs in Megf10 mutants "preferentially" project neurites to the soma layer, the authors need to show that each SAC in Megf10 mutants sends more neurites in the INL than IPL.

7) Subsection” SAC errors persist to adulthood in *Megf10* mutants”: The authors describe the laminar disruptions in mature retinas of Megf10 mutants occur "sporadically and at apparently arbitrary retinal locations." It is important to determine the percentage of IPL that is disrupted or shows ectopic stratification in Megf10 mutants. It is surprising that close to 100% SACs in Megf10 mutant mice show ectopic IPL projection in the adult (Figure 8F), but the laminar disruption shown in this manuscript (e.g. Figure 7A,B, Figure 8A and Figure9D) and the previous paper by Kay et al., 2012 seems to occur for a small fraction of the IPL length.

8) The authors use Six3-Megf10 cko and Chat-Megf10 cko to argue for separate time windows for laminar formation and somatic spacing. However, the data indicate that these two processes occur in overlapping temporal windows. First, irregular soma spacing with gaps in MEGF10 null retinas is very obvious in Figure 5A, indicating that mosaic spacing is already affected at P0. Second, earlier deletion of megf10 in Six3-cko mice causes more sever deficit in somatic spacing than in Chat-megf10 cko mice, indicating that somatic mosaics formation requires early megf10 signaling.

9) Please explain the rationale of the cross-correlation analysis in Figure 8—figure supplement 1. Why is there a strong correlation between the locations of SAC somas and the corresponding locations in the IPL layer? Given the morphology of SACs, it is unexpected to see such a correlation. In addition, statistical test needs to be performed for Figure 8—figure supplement 1C to claim a difference between mutant and control groups.

- Subsection “Early SAC projections target neighboring SAC somata”: Please clarify the statement "At P0-1, other IPL layers do not yet exist.…" Do the authors mean that there is no neurites in the IPL other than those of SACs?

What are the dashed lines of different depths in Figure 5E? How do the authors determine the border of IPL and INL given that SACs are the only amacrine cell type in early development? Delayed migration towards IPL and/or aberrant neurite growth can obscure the classification of IPL borders.

---

## [Author Response]

Reviewer #1:The authors use the development of starburst amacrine cells in the mouse retina as a model for understanding how the laminar organization of the retina arises during development. This is an excellent model system since this cell type is thought to be one of the earliest organizers of the inner-plexiform layer (IPL), a structure within the retina in which specific cell types arborize to form synaptic connections with appropriate partners.[…]This study is extremely well done – as noted already multiple approaches are used for all experiments. The authors do an excellent job quantifying their results and they are quite conservative in their interpretations. Every "i" is dotted and every "t" is crossed. This is surely going to be a definitive study not only on starburst amacrine cell development but also the role that this cell plays in organizing the entire direction selective circuit. The authors are to be commended.

We thank the reviewer for these kind words!

1) For all figures based on bar plots and SEM error bars (example, Figure 8B), the authors need to present the variance some way. Either the authors need to show individual data points, or they need to use standard deviation for error bars.

We have added individual data points to these graphs as requested. See Figures 5C, Figure 8B, Figure 9B and Figure 11E.

2) Subsection “Timing of DS circuit IPL sublayer formation”: Authors need to clarify how they determined "percent stratified". For example, they state only "30% of ooDSGCs co-fasciculated with SAC arbors". Is this percent of dendrites or percent of cells? What qualified as "co-fasciculated"?

We scored the fraction of ooDSGC cells, not dendrites, that were co-fasciculated; this has been clarified in subsection “Timing of DS circuit IPL sublayer formation” and subsection “Hb9-GFP stratification”. We have also added further details on how ooDSGCs were scored as co-fasciculated (same subsection “Hb9-GFP stratification”). In response to reviewer 3, we added an additional step to the scoring procedure: We now have examined fluorescence profile plots across the IPL for the SAC and ooDSGC channels, in addition to examining anatomy in Z-stacks. These are shown in Figure 1E.

The paragraph in the Materials and methods section now reads:

“P1-P2 retinas carrying *Megf10^lacZ^* and *Hb9-GFP* were co-stained for βgal and GFP. RGCs with dendrites that co-fasciculated with βgal-positive IPL strata were counted. […] Examples of cells falling into each category are provided in Figure 1 and Figure 1—figure supplement 2.” (subsection “Hb9-GFP stratification”).

The fluorescence profile plots were not always dispositive for scoring stratification, because at P1-2 a profile line could not always be drawn to avoid SAC cell bodies within or abutting the IPL (which would give a false “SAC band”). For this reason, we relied on expert judgment combining both visual inspection and profile plot data. For transparency about our scoring system we provide examples of “stratified” and “diffuse” ooDSGCs in Figure 1—figure supplement 2.

3) Figure 2J-L – purple color should labelled "β-gal" rather than "SAC" for consistency with other figures (e.g. Figure 5).

This has been fixed.

Reviewer #2Circuits in the retina are organized such that pre and postsynaptic cells elaborate and confine their processes and synaptic connections within specific sublayers of the neuropil. The sequence of early developmental steps giving rise to these lamination patterns and the molecular cues that are involved are not well understood. Ray et al., carried out an extensive set of experiments capitalizing on genetic tools, imaging and electrophysiology to fill this knowledge gap. They focused on a well characterized retinal circuit, involving inhibitory starburst (SAC) amacrine cells and their postsynaptic partners, the direction-selective (DS) ganglion cells. The authors propose that homotypic interactions direct the processes of SACs at the end stages of their migration to ramify in the IPL. They show convincingly that Megf10-mediated interactions are needed for timely SAC lamination, and if disrupted, leads to broader tuning of direction-selective responses in the DS ganglion cells. Overall, the data is of high quality, quantification and statistical analyses are provided and many findings are supported by the data presented. However, in some places, the conclusions are over-stated; such claims need to be substantiated by further analysis.1) A major conclusion is that SAC processes make homotypic contact with neighboring SAC cell bodies – this contact is what drives elaboration of SAC processes in the IPL.The evidence for homotypic contact is not strong. It is difficult to gain a good sense of what is happening based on the images, especially from the frozen sections. Figure 1J (an en face view) provides a bit more support but further analysis is necessary. At the minimum, using en face image stacks of the labeled SACs, the authors should perform a control by flipping one confocal channel and then re-measuring overlap between the channels, and demonstrate that the overlap is above chance.

We agree with the reviewer that claims concerning the homotypic nature of the SAC INL network required further evidence. We have added a new figure (Figure 2—figure supplement 3) to address this concern. This figure includes the channel-flipping analysis – we thank the reviewer for this suggestion. We found that the “random” rate of contact is about 15% (Figure 2—figure supplement 3). This is significantly lower than the 88.8% contact rate we observed in the real data; therefore, we conclude that the interaction between GFP^+^ individual SACs and the broader βGal^+^ SAC arbor network is likely to be selective and specific. Relevant Materials and methods section updates have been added to subsection “Characterization of SAC homotypic arbor network in soma layers”.

Also included in this new figure (Figure 2—figure supplement 3) is a more detailed documentation of the contacts sites. We provide an example of a putative contact between an individual GFP+ SAC arbor and an adjacent GFP-negative βGal+ arbor, showing in single optical planes, and in orthogonal views of 3D stacks, that the putative contact is apparent from all three perspectives. We now make clear in the Methods (subsection “Characterization of SAC homotypic arbor network in soma layers”) that this kind of analysis was performed during our quantification to confirm any contacts that were uncertain based on inspection of the individual Z slices.

For the main figure (Figure 2H,I), we chose to show a maximum-intensity projection of a larger Z volume encompassing the entire INL arbor. This is because the INL arbors often traversed many Z planes; therefore, only a maximum intensity projection gives the reader a real sense of the overall arbor morphology. However, as we now make clear in subsection “Characterization of SAC homotypic arbor network in soma layers” we did not use these Z-projections for quantification. Instead, we only counted putative contacts when they were confirmed in a single Z-plane, and/or with orthogonal views of the stack as illustrated in Figure 2—figure supplement 3.

To confirm that there is contact, electron microscopy (EM) is necessary. This is because the diffraction limit of light does not enable one to conclude from two labeled structures alone that there is 'contact'. If the authors perform the overlap measurements with the control, then I think it is fine to conclude that there is an association that is not random, and these are presumed to be contacts.

We agree that EM would be required to definitively demonstrate contacts, but we also agree with the reviewer’s comment that contacts are often inferred from light microscopy as long as appropriate anatomical analysis/controls are performed. We would argue that our revised manuscript, with 3D rotations and the cell flipping controls, now allows us to claim that homotypic contacts exist, at least to the resolution provided by light microscopy.

2) '[…]deletion of Megf10 causes a profound impairment of IPL-directed SAC dendrite growth, preventing timely sublayer formation'.It is clear that the processes of SACs, especially the OFF SCAs, in the Megf10 KO have not reached or contribute to a singular plexus in the IPL by P3 (although they do). But at P3 in the mutant, the majority of the processes appear directed towards the IPL (Figure 5B). It would be good to clarify whether the authors mean that Megf10 is important for the rate at which the SAC process lamination occurs, or they also mean, the direction of process outgrowth.

The reviewer is right that we used sloppy and confusing wording here in our original submission. We have removed all references to “soma-directed” and “IPL-directed” arbors and replaced them with descriptions of where SACs are ramifying/elaborating their arbors (i.e. within the soma layer or within the IPL). For example, in place of the sentence quoted by the reviewer, we now write, “These studies (i.e. deletion of Megf10) revealed a severe deficit in IPL dendrite arborization” (subsection “Requirement for MEGF10 in SAC IPL innervation and sublayer formation”).

3) SAC-BC repulsion regulates bipolar cell axon lamination.This isn't so clear to me because it is difficult to come to this conclusion based on the images. The small displacements in BC5+7 may simply be because where SACs occupy space, BC5+7 cannot elaborate their axons because the SAC process network is very dense. Where there is a gap in the SAC plexus, these BC axon terminals do come closer together, and this could be because space is now available rather than repulsion is absent.

The reviewer’s model is entirely plausible. We did not mean to argue against this possibility; it is one of many ways in which repulsion can be implemented. The point we sought to make in describing our observations is this: If SAC arbors occupy an IPL location, BC3a/BC5/BC7 bipolar cell terminals are excluded from that location. This stands in stark contrast to the behavior of ooDSGC dendrites: If SACs occupy an IPL location, ooDSGCs will occupy it too. Because some cells can enter while others cannot, we feel that describing this as “repulsion” is reasonable; the SAC barrier is not so impenetrable as to exclude every arbor.

Nevertheless, we take the reviewer’s point that caution is needed in how we describe this finding. In the Results section and the new model figure (Figure 11F), we do not use the word “repulsion” but instead simply describe where the bipolar cell axons go when confronted with a SAC ectopia or gap. We have revised the relevant portions of the Discussion section starting at subsection “SACs as a scaffold for DS circuit assembly”) to soften our claims, referring to this as “possible repulsion” and “arbor exclusion.” We also draw a contrast with ooDSGCs to make the point that there is some selectivity to the putative repulsion (Results section).

1) Throughout the text, the authors make statements about the behavior of the SACs, which are not strongly supported by the images. Here are some examples:i) Subsection “Homotypic contact is required for SAC IPL innervation and dendrite lamination”: "Misprojecting DACs are still in contact with numerous other amacrine[…]" and legend for Figure 3—figure supplement 1G: "SACs that made errors had extensive interactions with GAD65+ amacrine cells". It not possible to infer 'contact' or 'interactions' from these data. Please explain.

We apologize for imprecise wording here. What we should have said was: There is a strong likelihood of contact/interactions between SAC arbors and GAD65+ amacrine arbors in the IPL. To support this claim, we have gone back to examine the original confocal Z-stacks from the P15 *Ptf1a^cKO^* experiment. We find that the images in the figure cited by the reviewer (now Figure 3—figure supplement 2F) are entirely representative of the vast majority of our images from this experiment: GAD65+ arbors completely filled the IPL in all Z planes examined. Thus, when SACs projected into the IPL, they were highly likely to interact with if not contact GAD65+ arbors, as illustrated in Figure 3—figure supplement 2F.

Based on this analysis, and mindful of the reviewer’s wise suggestion to be careful in how these images are described, we now write in the Results section: “Misprojecting SACs were still closely opposed to numerous other amacrine cells, and their arbors were intermingled in the IPL, strongly suggesting that generic amacrine interactions are not sufficient to ensure normal dendrite targeting.”

The legend to Figure 3—figure supplement 2F has also been updated to read: “SACs that made errors likely interacted in the IPL with GAD65^+^ amacrine cell arbors (F, arrows) because these arbors completely filled the IPL in the region innervated by the SAC dendrites.”

ii)Subsection “SAC IPL errors induce laminar targeting errors by their DS circuit partners”: 'No changes were seen in Syt2-labeled BC6, suggesting a specific effect on the bipolar cells that make extensive contact with SACs.' The BC6 axons are much further away from the SAC ON plexus compared to the BC7 axon terminals. Thus, I am not sure one can readily conclude that 'DS-circuit bipolar cells respond to SAC attractive cues'.

Because the analysis of BC6 was “data not shown,” we have chosen to remove the sentence rather than perform a more comprehensive analysis. We now make claims only about the behavior of BC2, BC3a, BC5, and BC7.It is important to note that BC6 (unlike the four bipolar types we did analyze) is not currently thought to be a monosynaptic partner of SACs or ooDSGCs, so removing this data does not alter our conclusion that DS circuit bipolar cells change their IPL lamination to follow SAC errors.

2) The authors have really put in an enormous effort to obtain a comprehensive study. There is a lot of data and the supplements that complement the Figures are important. But I wonder if the authors would consider streamlining some parts because it is really a bit overwhelming for the reader to try and assimilate all the information relevant to each major conclusion by going back and forth from the main figures to the supplements. For example, the various effects on bipolar cells in the Megf10-/- retina; it would be helpful to at least provide the summary in a single figure for the bipolar cell types analyzed (BC2,3a,5,6,7).

We thank the reviewer for this helpful suggestion. We have included bipolar cell model figures as the reviewer requested (for clarity, we made separate model figures to put next to the primary data in the main and supplementary figures). To the larger point about streamlining: As we noted in the beginning of this document, we have added many new model figures, broken up large figures, and shortened the Results section to make the story easier to follow. One beneficial effect of our figure reorganization is that we made room to move certain key panels from the supplementary to the main figures. This should aid the reader in finding the data they are looking for.

3) The mistargeting of BC2 in Megf10-/- to where there are ectopically located SAC processes is intriguing and clear that there is a bipolar axon terminal lamination defect associated with abnormally placed SAC processes (but for a BC type that is not part of the SAC-DS circuit?).

BC2 is part of the SAC-DS circuit according to Duan et al., (2014) and Kim et al., (2014). Serial block-face EM reconstructions in Ding et al., (2016) show that, among the bipolar cell types, BC2 and BC3a make the most synapses onto OFF SACs, along with BC1.

However, to establish a causal link to SACs, it would help to show that lamination of other retinal neurons that are associated with BC2 bipolar cells (if that is known) are unperturbed in their lamination (i.e. are the BC2's following their targets?).

To our knowledge specific non-DS circuit targets of BC2 (and BC3a) have not been definitively identified. So additional experiments here are not possible. However, the reviewer’s point is well taken: It is possible that starbursts affect BC2 (and the other BC cell types) indirectly, by moving an intermediary cell type to a new laminar location. Such a mechanism would be entirely consistent with the “SAC as early scaffold” model, but it would imply that the direct bipolar cell interaction happens with the intermediary cell type instead of with SACs themselves. We now acknowledge this possibility in the Discussion section: “SACs might achieve their scaffolding functions directly, by providing guidance cues to their partners; or they may do so indirectly, by patterning the IPL projections of an intermediary cell type that in turn guides later-arriving projections.”

We also softened our claim concerning SAC-BC interactions in the Results section, referring now to SAC “guidance strategies” rather than “guidance cues”.

4) Figure 2—figure supplement 2: There should be quantitation for the E16 observations to substantiate SACs projecting to each other. The neuroblastic layer is dense with nuclei – where the processes appear to 'project', they seem to be just coursing around the nuclei.

As requested we have performed quantitative analysis of the internexin+ primary dendrites; more on this below. But first we should address a confusion that may have arisen for the reviewer due to imprecise language in our previous submission. We did not mean to claim that all SACs project directly towards each other; indeed, it would be impossible to conclude this from the internexin staining pattern. Rather we are making a point about the angle of primary dendrite orientation: If a SAC projects in the tangential plane, within the INBL (or INL/GCL at later stages), it is making a projection in the direction of neighboring somata rather than towards the IPL. By projecting in the tangential plane, E16 SACs make it possible to contact each other’s somata or soma-layer arbors. By contrast, P1-2 primary dendrites are oriented towards IPL and therefore quite unlikely to establish such contacts (although previously established contacts do persist through this period).

We have revised the Results to make clear that the internexin analysis is aimed at determining whether SACs orient their primary dendrites towards the IPL, or in the tangential plane where they might contact neighboring somata. To make this determination, we measured the angle of internexin+ primary dendrites at E16, and at P1 for comparison. This analysis, now presented in Figure 2F, shows that E16 SACs have a broad range of primary dendrite projection angles, including many that project tangentially within the INBL. By contrast, P1 SACs project almost exclusively towards the IPL. A schematic illustrating how the analysis was done is provided in Figure 2—figure supplement 1E. The Materials and methods section have also been updated to describe this analysis.

5) Are SACs not expected to be labeled in the Gad1-GFP line?

This is an important question; we thank the reviewer for spurring us to address it. Empirically, we found that SACs are labeled only rarely in the Gad1-GFP line. A picture is now provided in Figure 5—figure supplement 1 showing lack of overlap between GFP and the SAC marker *Sox2*.

6) It would be helpful to indicate in all the figures whether we are looking at a single image plane or a z-projection of several optical planes. This will help the reader gain a sense of just how 'close' the labeled structures are. What was the z-dimension step size for the confocal images on the wholemounts (not sure I found this in the Materials and methods section).

This is an important question; we thank the reviewer for spurring us to address it. Empirically, we found that SACs are labeled only rarely in the Gad1-GFP line. A picture is now provided in Figure 5—figure supplement 1 showing lack of overlap between GFP and the SAC marker *Sox2*.

7) Where possible, labeling the borders of the IPL would be helpful- presumably these are defined always by labeling of cell nuclei?

We have added vertical white lines or dashed horizontal lines to mark the IPL width/borders in many of the figures (e.g. Figure 1, Figure 4, Figure 5, Figure 7, Figure 8 and Figure 9). IPL borders were defined in various ways depending on the experiment: (1) labeling of cell nuclei; (2) tdTomato fluorescence from unrecombined cells in mTmG mice, which fills the IPL; (3) immunofluorescence against GAD65, which also fills the IPL; (4) autofluorescence signal intensity differences between soma layers and IPL. These details have been added to subsection “Image acquisition and processing”.

8) OFF SACs show more dramatic ectopic (somatic) processes than ON SCs in the Megf10 mutant. But this defect does not seem to have a physiological correlate – or does it?

This is a very interesting question. To ask if the effects on DS tuning width and strength were similar for the OFF and ON DS responses, we have now performed additional data analysis. We found that the magnitude of the mutant effect size was the same for both ON and OFF.

However, we are choosing not to include the analysis in the paper, for two reasons. First, it turns out that the necessary analysis was fairly complicated and would require its own multipanel figure to explain in a clear and satisfying way. We are reluctant to burden the reader with an entire new multi-panel figure that is mostly peripheral to the paper’s main point. Second, our analysis uncovered some interesting differences between the ON and OFF tuning curves for wild-type ooDSGCs. These are interesting enough that they merit a more complete unpacking in the context of a separate study; they really should not be buried in the supplementary figures of this paper.

The fact that the reviewer raised this point suggests to us that it is worth addressing in the paper. Given the above considerations we elected to make this “data not shown” and mention it in the legend to Figure 12. We write: “Mutant ooDSGC population is tuned more broadly (D, right shift of red curve) and more weakly (E, left shift of red curve) than WT. Similar results were obtained when ON and OFF responses were considered separately (not shown).”

We are extremely confident in the outcome of our analysis, so we think this is a reasonable way to balance general readability concerns with the specific interests of a subset of our readers.

9) Gad1-GFP line should be listed in the transgenics used.

We regret this oversight; it has been fixed (Key Resources Table; subsection “Animals”.)

10) Has the expression of cre in the Chat-cre line been documented somewhere? This is important with respect to separating the early and late deletion of Megf10.

We regret this oversight; it has been fixed (Key Resources Table; subsection “Animals”.)

11) Figure 8—figure supplement 1: Please provide a more detailed explanation of how the 2D cross-correlation analysis was performed.

For the sake of clarity and streamlining the manuscript, we have decided to remove this analysis. It is only useful for making a very abstract and theoretical point, which is somewhat peripheral to the manuscript’s main point. Moreover, it is very complicated to explain. For further details please see response to reviewer #3 (comment 3) below.

Given that this has been removed, we have now revised our conclusions about the interactions between mosaic phenotypes and IPL errors. We now write: “While mosaic spacing errors do not account for the *Megf10* mutant ectopic IPL phenotype, we cannot exclude the possibility that the placement of IPL arbor gaps might be at least partly explained by soma position.” (subsection “Formation of SAC IPL sublayers”).

12) Although it does appear that Meg10 plays two roles- one in somal mosaic positioning and the other in lamination of the processes of SACs, it isn't clear that timing alone separates these roles.

We did not mean to imply that timing is the only thing that separates the two MEGF10 roles. No doubt the mosaic positioning effect is mediated by its own cellular mechanisms; these would constitute additional factors, aside from timing, that separate the two MEGF10 functions. We apologize for being unclear on this point. Instead we meant to say that, since timing can separate the two roles, this demonstrates that they are indeed separate. During the earliest stages of SAC-SAC interactions (i.e. prior to P3), it is clear that MEGF10 influences both mosaic spacing and laminar targeting. If this were all we knew about MEGF10, it would be entirely possible that the laminar targeting and mosaic spacing phenotypes are really one and the same, entirely inseparable. The late deletion experiment indicates that is not the case – one phenotype can be induced without the other. This is the key point we sought to make. We have added a new model figure summarizing the results of the conditional mutant experiments, which should help clarify this point (Figure 9C).

If mosaic arrangements are due to repulsive SAC-SAC interactions, then fasciculation of the SAC processes must be an adhesion-based, perhaps even 'attractive' mechanism. Is it known if Megfs have other downstream pathways that are not 'repulsive'?

We agree with the reviewer that fasciculation – a key step in sublayer formation – must be adhesion/attraction based. In fact, we make this point in subsection “Formation of SAC IPL sublayers”. As we now try to suggest in the revised version of this paragraph, our data indicates that MEGF10 is not acting in a repulsive fashion, at least when it comes to IPL innervation. However, it is not the attractive/adhesion cue either, as shown by the fact that layers ultimately form in its absence. Instead, it seems to be a signal for a change in growth mode. Whether MEGF10 acts in this way during mosaic formation remains to be determined in future work. The revised text is quoted below in answer to the next query.

13) Could the authors elaborate a bit more in the Discussion section on how the lamination of the OFF and ON SAC plexuses become better confined to a single lamina (with exceptions) by adulthood?

The honest answer here is that we do not really know how the sublayers ultimately form in *Megf10* mutants. In subsection “SAC errors persist to adulthood in *Megf10* mutants” we point out that there are compensatory mechanisms that ultimately drive SACs to innervate the IPL and fasciculate there into sublayers (albeit imperfectly).

This comment from reviewer 2, together with the previous one, spurred us to elaborate in the Discussion section on one possible compensatory mechanism. We point out that there likely exist attractive/adhesive cues that promote homotypic SAC fasciculation – a key step in layer formation (subsection “Formation of SAC IPL sublayers”). This system remains intact in mutants, and therefore could be one of the compensatory mechanisms. The revised paragraph reads:

“It is generally assumed that sublayer formation has two basic molecular requirements: (1) Attractive/adhesive molecules that mediate co-fasciculation of stratified arbors; and (2) repulsive cues that prevent straying of arbors into other sublayers (Lefebvre et al., 2015; Sanes and Yamagata, 2009). Our MEGF10 studies suggest an additional, earlier requirement for cell-cell interactions that occur prior to neuropil innervation. The purpose of this surprisingly early SACSAC interaction, we propose, is to ensure that SACs grow dendrites at the right time and place to co-fasciculate with their SAC neighbors. The molecular basis of this homotypic co-fasciculation – clearly another essential player in sublayer formation – remains to be determined. MEGF10 is probably not involved; the cofasciculation system appears intact in *Megf10* mutants given that sublayers do eventually form. Perhaps this system is part of the mechanism that compensates for loss of MEGF10 to ultimately generate the sublayers.”.

Reviewer #3This study by the Kay lab investigates the role of Megf10 in the development of starburst amacrine cells (SACs). In a previous study by Kay and colleagues (Kay et al., 2012), the mosaic spacing of SACs is impaired in Megf10 knockout mice. In this manuscript, Ray and colleagues examine the early developmental stages of SACs starting at E16. They found that Megf10 is required for the timely stratification of SAC neurites at the inner plexiform layer (IPL). In the absence of Megf10, SACs fail to eliminate ectopic neurites and unable to stratify at the correct IPL depth in the first few postnatal days (P0-P2). This developmental deficit leads to mistargeting and gaps of SAC neurites in the IPL in the mature retina. Furthermore, using elegant genetic tools, the authors demonstrate that Megf10 function requires homotypic interaction of megf10 expressed in neighboring SACs. Overall, this study advances the current understanding of sublayer formation in the retina. It also provides a missing link between the altered developmental process of SACs and the deficits in the mature retina of Megf10 mutants. My major concerns are about some of the interpretations of the data and analysis.

We thank the reviewer for this comment; we are glad they agree that our work advances the field on several fronts.

1) One developmental feature that is prominently mentioned in this study is a transient homotypic network of SAC dendrites in the inner nuclear layer (INL) between E16 and P1. The authors describe the neurites outside the IPL to be selectively targeted to the neighboring SAC somas and neurites in the INL. However, an alternative scenario is that SACs during this period have random growth of neurites to both INL and IPL. After P1, the INL neurites are selectively pruned, and the IPL neurites further grow and stratify in the correct IPL sublayer. I did not find sufficient evidence supporting the presence and functional significance of a homotypic network of SAC arbors at the INL.

In addressing the question of whether a network of SAC INL arbors is present, we would first point out that our data show definitively that SACs make extensive transient projections to the soma layers (particularly the INL). The extent of these projections is quite surprising, since before now it had generally assumed that retinal neurons project stable arbors only within the plexiform layers, even during early development. In the initial submission we showed that most (~88%) of these INL SAC arbors contact a neighboring SAC soma or arbor. This finding led us to claim that the INL network is homotypic. The reviewer raises the quite reasonable question of whether this rate of apparent contact might arise even if “*SACs during this period have random growth of neurites*.” We have now performed a flipped-channel analysis to investigate this question (Figure 2—figure supplement 3). We found that the “random” rate of contact (in the flipped condition) is about 15% (Figure 2—figure supplement 3). This is significantly lower than the 88% contact rate we observed in the real data; therefore, we conclude that the interaction between GFP+ individual SACs and the broader βGal+ SAC arbor network is likely to be selective and specific.

We believe this finding should go a long way towards alleviating the reviewer’s concern, raised in the “alternative scenario,” that INL projections are random. The remaining aspects of the alternative scenario are not necessarily in conflict with what we propose is happening in embryonic retina. The reviewer’s alternative scenario mainly concerns *how* the soma-layer network arises. For instance, the alternative scenario posits that E16 SACs project in random directions. It is entirely possible, given our new data (Figure 2F), that this is in fact the case, but those dendrites within the INBL that contact neighboring SACs become stabilized, leading to homotypic network formation. We have not investigated in detail how the network arises, beyond showing that E16 SACs project into the soma layer such that they could plausibly contact their neighbors. Nevertheless, showing that this homotypic soma-layer network exists is a novel anatomical discovery; it motivates the rest of the study and, we hope, will change how the field thinks about early dendrite differentiation in retinal interneurons. How exactly the network arises is certainly an important question, but we feel it is best left for future studies.

In particular,- Subsection “Early SAC projections target neighboring SAC somata”: The authors describe SAC neurites in the INL as "soma-directed" neurites. It is unclear to me whether these neurites are "directed" at somas, or they are just randomly distributed in 3D. There is no quantification that shows "selective" somatic targeting over random growth. In fact, in both Figure 2E (E16) and 2H (P0), SACs send more neurites to IPL than to INL.

As noted in our response to reviewer 2, the choice of “somadirected” and “IPL-directed” as descriptors of SAC arbors was a poor one, and confusing. While a tangential projection within the soma layer might be considered “soma-directed,” the reviewer is right that we cannot establish the precise guidance parameters for any given dendrite; nor is this necessary to make the point that we wish to make. What we intended to communicate in the initial submission, and what we believe we now communicate in this revision, was the following: E16 SACs direct primary dendrites tangentially, within the soma layer, while postnatal SACs do not. By projecting tangentially within the INBL, embryonic SACs adopt a trajectory where they might contact their neighbors, thereby giving rise to the soma-layer contacts that we observed at E16 and that persist until P2-3 (Figure 2D,G).

To clarify this point we have replaced the term “soma-directed” with “soma-layer projections,” and have made other edits to subsection “Early SAC projections target neighboring SAC somata” that are consistent with the model described above. Further, we now add quantitative data supporting the existence of tangential soma-layer projections at E16 but not P1 (Figure 2F), and we show that the result of the E16 events is establishment of a homotypic INL network, not a random one (Figure 2—figure supplement 3). While there may be a developmental stage where arbors are distributed randomly in 3D, as the reviewer suggests, the ultimate outcome is a homotypic network, perhaps through selective stabilization of certain arbors. It will be interesting to investigate this possibility in future studies.

The reviewer is correct that many E16 primary dendrites are also directed towards the IPL. This is not in conflict with our model, as long as arbors are also projecting tangentially such that they could establish a soma-layer network. Our newly added data shows that this is the case – tangential and IPL-directed projections account for a similar fraction of total primary dendrites analyzed, with tangential being slightly more frequent (Figure 2F).

The key revised portion of the Results section reads:

“The majority of INBL SACs engaged in these soma-layer contacts, such that a GFP^+^ arbor network connected them (Figure 2G). Analysis of primary dendrite orientation indicated that soma-layer contacts likely arose due to projections targeted within this layer: Unlike mature SACs, which exclusively project their primary dendrites towards the IPL, many E16 SACs projected tangentially through the INBL – i.e., towards neighboring somata (Figure 2E,F).”.

- Subsection “Early SAC projections target neighboring SAC somata”: the authors claim that "P1 INL arbors selectively contacted somata or arbors of SAC neighbors". I have difficulty recognizing this pattern as "selective" since the INL at this stage is densely covered by SAC somas and ectopic neurites. The INL neurites may always end next to neighboring somas or neurites just because of the high density.

As noted above, we have addressed this concern by flipping one channel of the confocal stacks to obtain an expected rate of interaction if arbors were randomly distributed (Figure 2—figure supplement 3). The “random” rate of contact (in the flipped condition) is about 15%, significantly lower than the ~88% contact rate we observed in the real data.

- Subsection “Early SAC projections target neighboring SAC somata”: The authors describe that On SACs send fine soma-directed branches from their IPL arbors (Figure 2—figure supplement 2). Again, how do the authors distinguish "soma-directed" projection from random contacts due to imperfect stratification?

In the figure panels cited by the reviewer (now Figure 2—figure supplement 3E,F) we provide specific examples of cases where ON SACs branched to contact SAC somata located either within the IPL, or at the IPL/GCL border. We have not performed the same quantitative specificity analysis as we did for OFF SACs. But we argue that the anatomy shown in Figure 2—figure supplement 3E,F is consistent with ON SACs behaving similarly to OFF SACs. The figure shows: (1) the existence of branching morphology that brings dendrites into the vicinity of neighboring homotypic somata; and (2) the existence of actual dendro-somatic SAC-SAC contact. Thus, ON SACs, like OFF SACs, create the possibility for soma contact by projecting into the soma layer, and this possibility is realized, at least in some cases. We propose that the most likely interpretation of these ON-OFF similarities is that both SAC subtypes are engaging in homotypic interactions.

However, because we have not performed the specificity analysis, we softened the language used to describe the ON SAC anatomy. We have removed the word “selective” when describing ON SAC interactions, and we now write, “ON SACs also made soma layer projections between P0-P3 that contacted neighboring SAC somata” (subsection “Early SAC projections target neighboring SAC somata”).

2) Subsection “Timing of DS circuit IPL sublayer formation”: The authors mention that SACs are stratified within the expected IPL sublayer, but ooDSGC dendrites are unstratified. This needs to be supported by quantifications. In Figure 1—figure supplement 1, panel C, both SACs and the ooDSGC appear to be unstratified in a similar way.

We thank the reviewer for a suggestion that undoubtedly strengthens the paper. We have now performed fluorescence profile plots across the IPL for P1 and P2 dual-channel SAC & ooDSGC images, similar to what we had previously included for mature retina (old Figure 9; new Figure 10 and Figure 11). Representative examples are now shown in Figure 1E. These plots show clear sublaminar peaks for SAC dendrites at both P1 and P2, while ooDSGC dendrites are distributed more broadly across the IPL (Figure 1E). Importantly, the representative profile chosen for P1 is from the cell that the reviewer questions in the comment above; the profile clearly shows stratification of SACs but not ooDSGCs. While seeking to improve the clarity of this figure and its accompanying supplementary figure (Figure 1—figure supplement 2), we also chose a better representative image to include in panel Figure 1D. The previous panel 1D cell is now in the supplement.

3) Subsection “Homotypic contact is required for SAC IPL innervation and dendrite lamination”: "the projections were typically more elaborate than those observed in wild-type retina". Please provide a quantitative analysis to support this claim.

This is a good suggestion, but unfortunately, we have not been able to devise a satisfactory way to do it. We attempted to measure arbor coverage area and other dendrite anatomical parameters in whole-mount, *en face* views of solitary and touching SACs, but the βGal labeling was too dim to permit such an analysis. Measuring these parameters in crosssection is difficult, because arbors may be severed at the sectioning border. We felt that any quantitative measure we might obtain from these samples would therefore show only a relative difference between the two groups, rather than providing an absolute measure (e.g. arbor area, total arbor length, or Sholl branch complexity) that could be compared productively across experiments and across labs. Because this relative difference in arbor complexity is already quite readily apparent from a qualitative inspection of the micrographs (Figure 3D, arrow vs. Figure 3F; Figure 3—figure supplement 2), we argue that quantification would add little value. Indeed, as shown in these figures, many solitary SACs have a dendrite morphology completely outside the range of anything observed for controls. Quantifying dendritic material in the INL would fail to accurately describe this qualitative morphological change.

Furthermore, we already showed quantifications of the number of solitary SACs projecting to the soma layer – this was increased relative to both controls (Figure 3G). So, when we assert that solitary SACs over-innervate the soma layer, we have both qualitative and quantitative data to back up this claim.

To aid the reader in assessing the validity of our qualitative claim, we created a supplementary figure (Figure 3—figure supplement 2A-D) devoted to providing multiple examples of solitary SAC dendrite morphology. The text has also been revised (legend to Figure 3—figure supplement 2) to clarify that the example images are indeed representative.

4) Subsection “Requirement for MEGF10 in SAC IPL innervation and sublayer formation”: "this phenotype is not due to aberrant SAC migration.……" However, from Figure 5A and Figure 5B, Off SAC somas in MEGF10 mutants are farther away from IPL than those in the wild type.

We agree with the reviewer that this is too strong a statement; we have clarified by specifying that we are referring here specifically to “radial migration” and that it occurred “similar to littermate controls.” However, based on the evidence provided, we feel comfortable making our revised assertion that “Loss of IPL innervation was not due to aberrant SAC radial migration.” (subsection “Requirement for MEGF10 in SAC IPL innervation and sublayer formation”). While it is true, as the reviewer points out, that P1 OFF SACs are clearly farther from the IPL in mutants than in controls, we find no evidence for a major difference at P0. This can be observed in Figure 5A, where INL SACs are positioned adjacent to the IPL in both mutants and littermate controls. It is also clear from our counts of INL SACs at P0 (reported in the text in subsection “Requirement for MEGF10 in SAC IPL innervation and sublayer formation”), which showed no difference in the number of SACs arriving in the INL by this age. Finally, one can appreciate in cross-sections (Figure 5A; data not shown) that mutant SACs do not arrest their migration in the ONBL; they successfully reach inner retina by P0, in normal numbers. Together these data are consistent with a model whereby SACs complete their radial migration more or less normally, arriving at the IPL, but then moving away from it (possibly as a result of failing to innervate it properly) between P0 and P1.

We have not measured the soma-IPL distance at P0, so it is possible that there may be some subtle effects of *Megf10* on final position after migration. For this reason, we agree with the reviewer that we should not say radial migration was “normal.” But the issue here is whether the migration defect (if any) is sufficient to explain the lack of IPL innervation. If the SACs are farther from the IPL at P0, it is not by much; indeed, as we now point out in the text (subsection “Requirement for MEGF10 in SAC IPL innervation and sublayer formation”), most SACs are able to send at least some dendrites into the IPL, suggesting that they migrated to a location from which IPL innervation was feasible. However, once they enter the IPL, these dendrites do not ramify and stratify as control dendrites do. These observations strongly suggest that any migration/radial positioning defects are unlikely to be the primary cause of IPL innervation defects in *Megf10* mutants. The relevant section of the manuscript now reads:

“Loss of IPL innervation was not due to aberrant SAC radial migration, because, at P0, mutant SACs had reached the inner retina in normal numbers (wild-type, 2600 ± 287 SACs/mm^2^; mutant, 3153 ± 145 SACs/mm^2^; *p* = 0.144, 2-tailed *t-*test; *n* = 3 each group), and were positioned adjacent to the IPL, similar to littermate controls (Figure 5A). Furthermore, most mutant SACs sent at least some arbors into the IPL at P0-1 (Figure 5A,C; Figure 6C), suggesting that they migrated to a location from which IPL innervation was feasible (Figure 5A-C; Figure 6C). However, the mutant SAC arbors that reached the IPL appeared undifferentiated, with a lack of space-filling branches (Figure 5C,E).” (subsection “Requirement for MEGF10 in SAC IPL innervation and sublayer formation”).

In addition to this change, we also point out in the legend to Figure 5 that “mutant somata abut the IPL at P0, suggesting their radial migration was similar to controls. By P1, OFF somata have moved apically.”

And there are significantly less On SACs in the mutant. This is consistent with delayed migration of SACs, which might also account for the delayed stratification phenotype in the mutants.

The reviewer makes an interesting observation concerning the ratio of OFF to ON SACs. Chow et al. (2015) showed that displaced amacrine cells cross the IPL using a migratory strategy that is distinct from radial migration: At the completion of their radial migration to the IPL, cells flatten in the tangential plane and extend bipolar arbors within the IPL. We observed SACs with this morphology (Figure 2—figure supplement 3) suggesting that ON SACs likely cross the IPL in this way.

Inspired by the reviewer’s suggestion, we did a pilot study on a small number of sections from a small number of P0 animals (5 fields of view from 2 *Megf10* mutants and 2 littermate controls). We found that, while the total number of SACs in the INL and GCL was unchanged between mutant and control, the INL/GCL ratio was indeed slightly higher in the mutants, as the reviewer observed (67% INL in mutants, 54% INL in controls). The fact that total SAC number was similar suggests that radial migration is completed largely normally, consistent with data reported in the manuscript (for further details see our response above to comment 3H). However, the altered INL/GCL ratio suggests there may be a requirement for MEGF10 in the IPL crossing behavior discovered by Chow et al.

Proving that this is in fact the case would require substantial further experiments, which we feel are outside the scope of this study. Even if the phenotype were confirmed, it would not change our conclusion that migratory phenotypes cannot explain the loss of IPL innervation (subsection “Requirement for MEGF10 in SAC IPL innervation and sublayer formation”). This is because IPL crossing happens after radial migration is complete; moreover, it happens concurrently with initial IPL innervation rather than prior to it (Chow et al., 2015). If something has gone wrong with crossing, the problem likely occurred during or after initial interaction with the IPL and would not have prevented such interactions.

The reviewer suggests that delayed crossing might be responsible for delayed stratification in *Megf10* mutants. This is a reasonable hypothesis. However, given our finding that IPL innervation is delayed in mutants, it is equally reasonable to propose the opposite hypothesis: If mutant SACs are unable to properly send arbors into the IPL, this might delay the onset of crossing behavior. Since both possibilities are reasonable, additional work will be required to distinguish between them (if it turns out that MEGF10 mutants do in fact have an IPL crossing phenotype). It will be quite interesting to carry out future studies to resolve this issue. However, we feel that such work is clearly separate from this study, and not necessary to support our conclusions.

To take into account the possibility that IPL crossing might not be normal in *Megf10* mutants, we have revised our migration claim to specifically address radial migration, rather than migration generally (subsection “Requirement for MEGF10 in SAC IPL innervation and sublayer formation” and the legend to Figure 5).

5) Subsection “Requirement for MEGF10 in SAC IPL innervation and sublayer formation”: "Rather, the absence of SAC sublayers was due to innervation of the soma layers instead of the IPL". I do not see a clear causal relationship here.

Thank you for pointing out our imprecise wording. We did not mean to imply that the soma layer overinnervation causes absence of sublayers; this sentence has been removed. The absence of sublayers is attributable to the IPL innervation deficits, as we now make clear in the portion of the Results section that replaces this sentence (subsection “Requirement for MEGF10 in SAC IPL innervation and sublayer formation”).

6) Related to point 1 above, Subsection “Requirement for MEGF10 in SAC IPL innervation and sublayer formation”: To support the statement that P1 SACs in Megf10 mutants "preferentially" project neurites to the soma layer, the authors need to show that each SAC in Megf10 mutants sends more neurites in the INL than IPL.

Perhaps “preferentially” was too strong of a word choice; we have removed it. The section describing *Megf10^–/–^* phenotypes (subsection “Requirement for MEGF10 in SAC IPL innervation and sublayer formation”) has been extensively re-written to avoid over-interpretation. We now focus on pointing out that mutant SACs have two phenotypes: underinnervation of the IPL (Figure 5) and the overinnervation of the soma layers (Figure 6).

To address the reviewer’s concern, we have also added new figure panels and analysis to support the assertion that P1 SACs overinnervate the INL:

1) We now show in the main figures, using *Megf10:*βGal, that the mutant P1 INL arbor network is more extensive than the control network. This was previously in the supplementary figures, but we have moved it to a much more prominent main figure location (Figure 6C) because of the importance of this data.

2) We highlight (subsection “SAC dendrite targeting requires transcellular MEGF10 signaling”) an important finding that was present in the earlier submission but did not receive sufficient emphasis: At P1, the number of SACs projecting to the INL is the same in mutants and controls (Figure 6E). If the same number of SACs participate in the INL network, but the network is denser in mutants (Figure 6C), it stands to reason that the typical mutant SAC must have a larger/more elaborate INL arbor than the typical control SAC.

3) In support of this inference, we now provide high-magnification *en-face* images of representative control and mutant SACs, documenting more elaborate INL arbors in mutants (Figure 6B).

These new data bolster the lower-magnification views included in our original submission (Figure 5A,B; Figure 6A), which clearly show at P1, and especially at P3, that the amount of dendritic material in the mutant INL is dramatically higher than in controls. Indeed, between P0 and P3, the amount of dendritic material in the mutant INL is stable or even increases, while it decreases in controls (Figure 5A-B). While we cannot claim (nor did we intend to claim in the original submission) that each individual SAC sends more neurites to the INL than the IPL, we feel our data strongly supports the following claim: At the population level, mutant SACs overinnervate the INL during the period when their IPL arbors are underdeveloped. The relevant portion of the Results section has been extensively revised to reflect this point.

7) Subsection” SAC errors persist to adulthood in Megf10 mutants”: The authors describe the laminar disruptions in mature retinas of Megf10 mutants occur "sporadically and at apparently arbitrary retinal locations." It is important to determine the percentage of IPL that is disrupted or shows ectopic stratification in Megf10 mutants.

We agree with the reviewer about the importance of documenting the extent of *Megf10* IPL phenotypes. We showed in the initial submission that ~80% of OFF SACs participate in the ectopic network (Figure 8F), highlighting the prevalence of these errors. At the reviewer’s request we now extend our quantitative analysis to the IPL gap phenotype: We measured the area covered by the SAC dendritic plexus in ON and OFF sublayers and found that *Megf10* mutant SACs cover ~15% less retinal area than control SACs (15.0 ± 0.9% decrease for OFF, 13.7 ± 4.0% decrease for ON, mean ± SD). These data are now included in subsection “SAC errors persist to adulthood in *Megf10* mutants”, and the methodology is described in subsection “Characterization of SAC homotypic arbor network in soma layers”.

Retinal coverage measurements were obtained by segmenting *en-face* images of the ChAT+ SAC plexus and calculating the field-of-view fraction occupied by SAC arbors. We attempted to use this approach to quantify *en-face* images of the ectopic OFF SAC network, but we could not obtain accurate measurements due to the presence of cell bodies (either out of focus or partially in focus) within the images at this IPL level. Even though we could not perform the exact analysis suggested by the reviewer, we would argue that our measurements of the fraction of SACs that make errors (Figure 8F; Figure 9—figure supplement 1D) provides a reasonable quantitative description of error prevalence.

It is surprising that close to 100% SACs in Megf10 mutant mice show ectopic IPL projection in the adult (Figure 8F), but the laminar disruption shown in this manuscript (e.g. Figure 7A,B, Figure 8A and Figure9D) and the previous paper by Kay et al., 2012 seems to occur for a small fraction of the IPL length.

Based on the figure panels cited by the reviewer, their confusion at the extent of the ectopic SAC projection phenotype is likely due to looking primarily at cross-sections. We should have done a better job drawing the reader’s attention to the *en-face* panels (e.g. Figure 8E; Figure 10C,D). This vantage point provides the best view of the overall anatomy of the SAC projections; it is hard to appreciate how extensive the IPL errors are without looking at these panels. This is because the amount of ectopic arbor present in a cross-sectional view is highly dependent on how the sectioning angle falls relative to the ectopic arbor fascicles. For example, a perpendicular cut through a long fascicle will leave only a tiny cross-sectional footprint. We have now rewritten the relevant portion of the Results section to draw attention to the whole-mount panels:

“The second type of SAC error in mature *Megf10*^–/–^ IPL was dendrite mistargeting to ectopic IPL strata (Figure 8A,B,E). Both ON and OFF SACs were affected; in each case ectopic arbors were mostly found in IPL regions inappropriately close to the soma layers (Figure 8A,B). *En-face* images of mutant IPL revealed that ectopic OFF arbors formed a patchy but extensive fascicle network connecting many of the cells (Figure 8E,F; 78.5 ± 3.5% of SACs participated in the network). (subsection “SAC errors persist to adulthood in *Megf10* mutants”).

8) The authors use Six3-Megf10 cko and Chat-Megf10 cko to argue for separate time windows for laminar formation and somatic spacing. However, the data indicate that these two processes occur in overlapping temporal windows. First, irregular soma spacing with gaps in MEGF10 null retinas is very obvious in Figure 5A, indicating that mosaic spacing is already affected at P0. Second, earlier deletion of megf10 in Six3-cko mice causes more sever deficit in somatic spacing than in Chat-megf10 cko mice, indicating that somatic mosaics formation requires early megf10 signaling.

The reviewer is absolutely correct: The time windows are “distinct,” but certainly not separate. As we wrote in the original submission, “… MEGF10 appears to act at distinct, albeit partially overlapping times, to control different aspects of SAC development, each of which are regulated by contact with homotypic neighbors.” (subsection “Formation of SAC IPL sublayers” of original submission; subsection “MEGF10 as the signal mediating SAC homotypic recognition” of new submission). We regret that the manuscript was not written clearly enough for this point to come through in the previous version. To clarify this point we have added a model figure (Figure 9C) summarizing all of the results of our conditional deletion experiments. As shown in the model, MEGF10 is continuously required until at least P3-5 for mosaic spacing, whereas its requirement in laminar targeting has ended by P3. We have also updated the text to more accurately reflect this point. For example, we now write that the two processes have “separable temporal requirements,” rather than “different” temporal requirements as in the first submission (subsection “MEGF10 as the signal mediating SAC homotypic recognition”).

9) Please explain the rationale of the cross-correlation analysis in Figure 8—figure supplement 1. Why is there a strong correlation between the locations of SAC somas and the corresponding locations in the IPL layer? Given the morphology of SACs, it is unexpected to see such a correlation. In addition, statistical test needs to be performed for Figure 8—figure supplement 1C to claim a difference between mutant and control groups.

In the interest of manuscript clarity, we have removed this cross-correlation analysis. The analysis is very abstract, and as the reviewer points out, the connection to anatomy and mechanism is opaque. It is nowhere near as definitive as the other “mosaic vs. IPL error” analysis that we performed (Figure 9—figure supplement 1). Furthermore, the rhetorical point supported by this analysis is rather peripheral to the overall thrust of the manuscript. Finally, the analysis requires a lot of explanation and is hard for the reader to grasp. For all these reasons, we decided to remove this from the paper.

Given that this has been removed, we have now revised our conclusions about the interactions between mosaic phenotypes and IPL errors. We now write: “While mosaic spacing errors do not account for the *Megf10* mutant ectopic IPL phenotype, we cannot exclude the possibility that the placement of IPL arbor gaps might be at least partly explained by soma position.” (Discussion section).

- Subsection “Early SAC projections target neighboring SAC somata”: Please clarify the statement "At P0-1, other IPL layers do not yet exist.……" Do the authors mean that there are no neurites in the IPL other than those of SACs?

We apologize for the lack of clarity. What we meant to say was that neurites are present in the IPL but not stratified. We revised this sentence to read: “Because SACs stratify early – before any other cell type investigated to date (Figure 1; Kay and Sanes, 2013; Stacy and Wong, 2003) – they are unlikely to form strata by following pre-existing laminar cues.” (subsection “Early SAC projections target neighboring SAC somata”).

What are the dashed lines of different depths in Figure 5E? How do the authors determine the border of IPL and INL given that SACs are the only amacrine cell type in early development? Delayed migration towards IPL and/or aberrant neurite growth can obscure the classification of IPL borders.

The dashed lines denote the borders of the IPL in each image. We had neglected to note this in the figure legend, but this has now been corrected (this panel is now Figure 6D). As we mentioned above in response to reviewer 2: IPL borders were defined in various ways depending on the experiment: (1) labeling of cell nuclei; (2) tdTomato fluorescence from unrecombined cells in mTmG mice, which fills the IPL; (3) immunofluorescence against GAD65, which also fills the IPL; (4) autofluorescence signal intensity differences between soma layers and IPL. These details have been added to subsection “Image acquisition and processing”.